# GAME-RL:SYNTHESIZING MULTIMODAL VERIFIABLE GAME DATA TO BOOST VLMS GENERAL REASONING

**Jingqi Tong**[1,2*], **Jixin Tang**[1*], **Hangcheng Li**[1*], **Yurong Mou**[1*], **Ming Zhang**[1], **Jun Zhao**[1†]
**Yanbo Wen**[1],**Fan Song**[1],**Jiahao Zhan**[1],**Yuyang Lu**[1],**Chaoran Tao**[1],**Zhiyuan Guo**[1],**Jizhou Yu**[1]
**Tianhao Cheng**[1],**Zhiheng Xi**[1],**Changhao Jiang**[1],**Zhangyue Yin**[1],**Yining Zheng**[1],**Weifeng Ge**[1]
**Guanhua Chen**[5], **Tao Gui**[1,2,3], **Xipeng Qiu**[1,2,3†], **Qi Zhang**[1,3,4†], **Xuanjing Huang**[1,3,4]
[1]Fudan University [2]Shanghai Innovation Institute
[3]Shanghai Key Laboratory of Multimodal Embodied AI
[4]Shanghai Artificial Intelligence Laboratory [5]Southern University of Science and Technology
`jqtong25@m.fudan.edu.cn, {zhaoj19, xpqiu, qz}@fudan.edu.cn`

## ABSTRACT

Vision-language reinforcement learning (RL) has primarily focused on narrow domains (e.g. geometry or chart reasoning). This leaves broader training scenarios and resources underexplored, limiting the exploration and learning of Vision Language Models (VLMs) through RL. We find video games inherently provide rich visual elements and mechanics that are easy to verify. To fully leverage the multimodal and verifiable rewards in video games, we propose Game-RL, constructing diverse game tasks for RL training to boost VLMs general reasoning ability. To obtain training data, we propose Code2Logic, a novel approach that adapts game code to synthesize reasoning data with unlimited examples and controllable difficulty gradation, thus obtaining the GameQA dataset of 30 games and 158 verifiable tasks. Remarkably, RL training solely on GameQA enables multiple VLMs to generalize across 7 diverse out-of-domain vision-language benchmarks, demonstrating the value of Game-RL for enhancing VLMs general reasoning. Furthermore, game data provides improvements comparable to general multimodal reasoning datasets (e.g. geometry/chart). More importantly, scaling up game diversity or game data volume consistently improves VLMs' generalizable reasoning capabilities. Our findings highlight scaling reinforcement learning in game environments as a promising direction for enhancing generalizable multimodal reasoning in foundation models. All code, dataset and model weights are released at `https://github.com/tongjingqi/Game-RL`.

## 1 INTRODUCTION

Vision-Language Models (VLMs) have achieved impressive progress in basic tasks such as image description and vision question answering. However, they still struggle with diverse and complex tasks that require multi-step reasoning in real-world scenarios (Lu et al., 2023; Zhang et al., 2024a). One key reason is that vision-language RL primarily focused on narrow domains, mainly including geometry (Peng et al., 2024; Zhang et al., 2024c) and chart reasoning (He et al., 2024), as shown in Table 1. This leaves broader training scenarios underexplored, limiting the exploration and learning of VLMs through RL (Lu et al., 2023; Liu et al., 2025; Jiang et al., 2024a; Zhao et al., 2024; Wang et al., 2025).

We recognize that video games have three inherent advantages: First, they have rich visual elements and scenes with texts. Second, their mechanics are easy to verify, so we can synthesize reliable tasks with verifiable results. Third, their environments are fully controllable and easy to modify, providing convenience for difficulty control. Although existing works (Zhang et al., 2024a; Paglieri et al., 2024; Zhang et al., 2025; Li et al., 2024) found game a good arena to evaluate VLMs' reasoning ability,

---

[*]Equal Contribution
[†]Corresponding Authors

Table 1: Our GameQA dataset extends RL training scenarios for VLMs to the domain of video games, providing diverse verifiable game tasks along with controllable difficulty. Comparison of existing visual reasoning datasets. Size means sum of the number of VQA pairs in train set and test set. Ration. Annot. means has annotated reasoning process. 3D Scene means some games of it are 3D scene. Detailed introduction can be found in Section 6.

| Reasoning Domain | Related Work | Train Set | Game Count | Ration. Annot. | 3D Scene | Adjustable Difficulty |
|---|---|---|---|---|---|---|
| Math | MAVIS (Zhang et al. (2024c)) | ✔ | N/A | ✔ | ✘ | ✘ |
| | MultiMath (Peng et al. (2024)) | ✔ | N/A | ✔ | ✘ | ✘ |
| | Geo170k (Gao et al. (2023)) | ✔ | N/A | ✔ | ✘ | ✔ |
| | MathV360K (Jiang et al. (2024b)) | ✔ | N/A | ✔ | ✘ | ✔ |
| Game | ING-VP (Zhang et al. (2024a)) | ✘ | 6 | ✘ | ✘ | ✔ |
| | BALROG (Paglieri et al. (2024)) | ✘ | 6 | ✘ | ✘ | ✘ |
| | VideoGameBench (Zhang et al. (2025)) | ✘ | 23 | ✘ | ✔ | ✘ |
| | VCbench (Li et al. (2024)) | ✘ | 10 | ✘ | ✘ | ✔ |
| | GameQA (Ours) | ✔ | 30 | ✔ | ✔ | ✔ |

they did not apply training on it. One possible reason is they did not transform game data into visual question and answer (VQA) format to adapt to training, as shown in Table 1.

To fully use the multimodal and verifiable rewards in game, we propose Game-RL, constructing game scenarios for RL training to boost VLMs general reasoning ability. Then we propose Code2Logic, a novel approach that adapts game code to synthesize game reasoning task data, as shown in Figure 1. Code2Logic establishes mapping from game code to reasoning logic. Using Code2Logic, we have constructed the GameQA, which is a visual question-answer (VQA) dataset to train and evaluate the reasoning capabilities of VLMs. GameQA includes 30 games, 158 tasks and 140K questions with controllable difficulty gradation, covering 4 cognitive categories, as shown in Figures 2 and 3.

We then apply Group Relative Policy Optimization (GRPO) (Guo et al., 2025) to train multiple VLMs exclusively on GameQA. Remarkably, despite training solely on game tasks, these models demonstrate out-of-domain generalization, achieving consistent performance improvements across 7 diverse vision-language reasoning benchmarks (e.g., Qwen2.5-VL-7B improves by 2.65%). Furthermore, we find that training on GameQA yields improvements comparable to training on general multimodal reasoning datasets such as MAVIS (Zhang et al., 2024c) and MultiMath (Peng et al., 2024). More importantly, scaling up either game diversity or data volume consistently enhances VLMs' generalizable reasoning capabilities.

Our main contributions are as follows:

- We propose Game-RL, constructing game tasks for RL training to boost VLMs' general reasoning ability. To obtain training data, we propose Code2Logic, a novel approach that adapts game code to synthesize diverse verifiable game task data with unlimited samples and controllable difficulty.

- Using Code2Logic, we develop the GameQA dataset, which includes 30 games, 158 tasks and 140K questions with controllable difficulty gradation.

- We show that VLMs trained solely on GameQA through RL achieve out-of-domain generalization across 7 diverse vision reasoning benchmarks, demonstrating the value of Game-RL for enhancing VLMs' general reasoning. Our findings highlight scaling reinforcement learning in game environments as a promising direction for enhancing models' generalizable reasoning.

## 2  CODE2LOGIC: SYNTHESIZING VERIFIABLE GAME TASK DATA

We propose Code2Logic to synthesize verifiable game reasoning tasks data via adapting game code. Code2Logic comprises three core steps: game code construction, task template design, and data

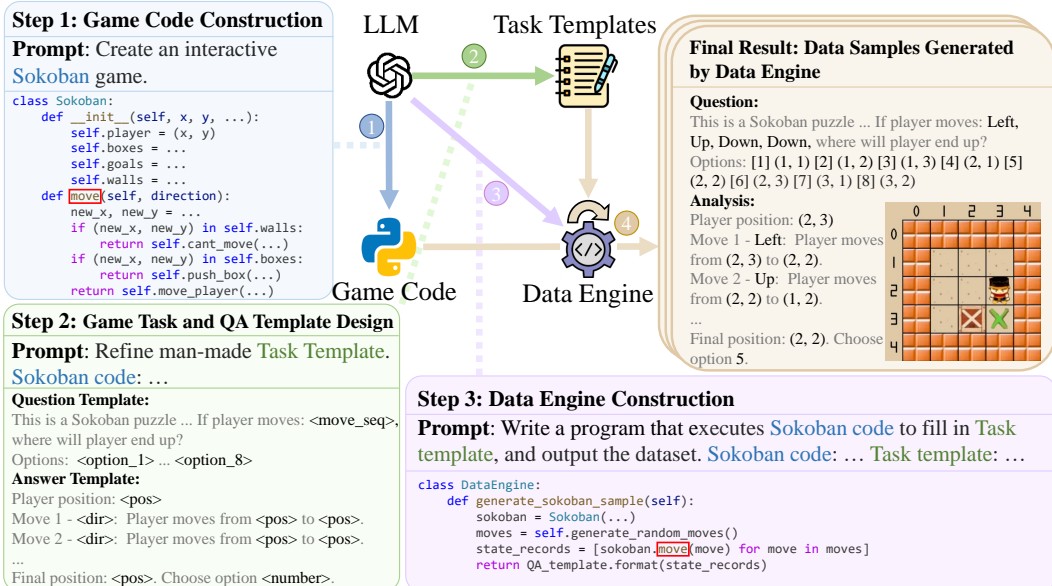

Figure 1: Overview of Code2Logic approach. The process involves three main steps: (1) Using LLMs to construct game code. (2) LLM-assisted design of the task templates including question and analysis templates based on the generated game code. Each task template condenses one type of reasoning pattern in the game. (3) Using LLMs to construct a data engine that directly reuses the core game code from the first step, including functions like `move`. (4) After these main steps, the data engine is executed to fill in the task templates developed in Step 2 and generate data samples, as illustrated in the "Final Result" section.

engine construction for dataset generation, as shown in Figure 1. Code2Logic establishes a mapping from game code to reasoning logic.

## 2.1 GAME CODE CONSTRUCTION

The first step is to construct the code of the target video game. Game code defines the state space and contains some core functions encoding the transition rules of the game. These functions can be reused and adapted for data engine construction.

Taking Sokoban as an example, it is simple enough to be easily built with one-line prompt using LLMs such as Claude 3.5 and GPT-4o. Sokoban game state is only composed of wall, player, box and target, while action only includes moving in four directions. The prompt used for game code generation and pseudocode of the game is illustrated in Figure 1 (Step 1). Meanwhile, the `move` function in Sokoban's game code can inspire the "State Prediction" questions, such as "Predict where will player end up after these steps", and this function can be reused for data engine construction.

## 2.2 GAME TASK AND QA TEMPLATE DESIGN

The second step is to design game task and question-answer templates (QA Templates) according to each video game. Game task designing base on the visual element and action space of video game generated in the first step. Taking Sokoban as an example, we can design a task about predicting the position of player after moves according to a game state screen, so we can propose a question and an answer. The question is "If the player moves left, up, down, down, where is the player?" and the answer is "The position of player is (2,2)." Then we can generalize these questions and answers into a QA Template, as shown in Figure 1 (Step 2). Meanwhile, we can fill contents into a QA Template to get many instances. Detailed QA Templates of Sokoban is provided in Appendix G.1.

In GameQA dataset, each task is defined as a group of samples from the same game that share the same QA template. We categorize the tasks into three types of "Target Perception Task", "State Prediction Task" and "Strategy Optimization Task", with descriptions detailed in Appendix E.1.

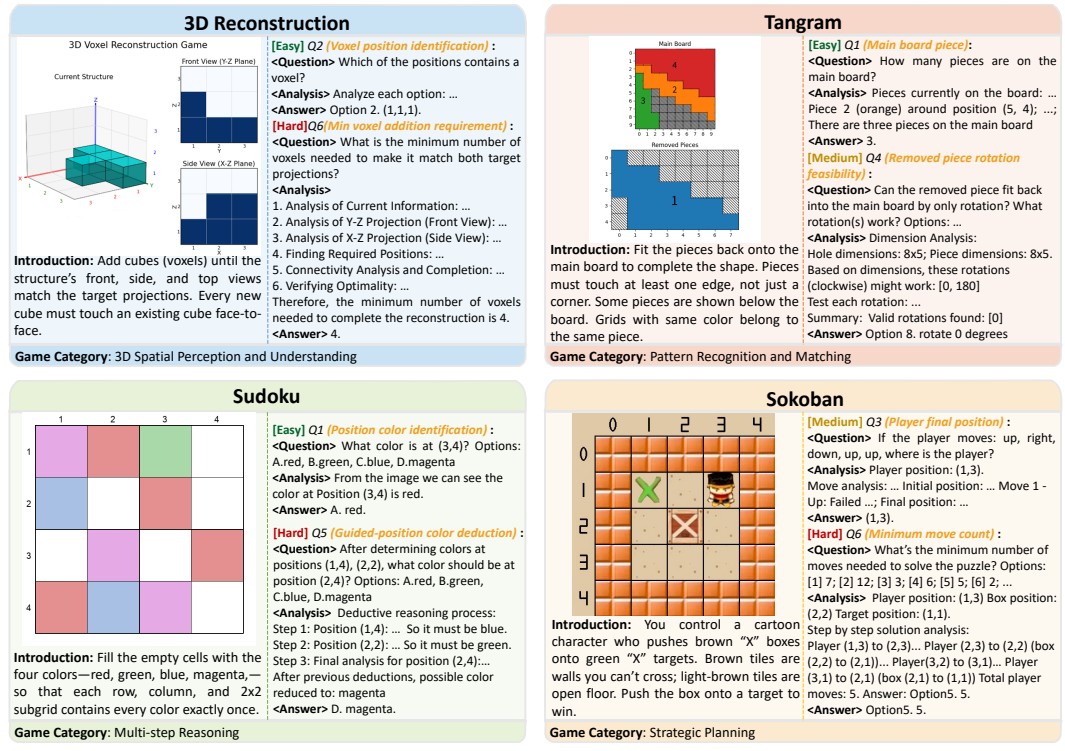

Figure 2: Four game examples from GameQA: 3D Reconstruction, Tangram, Sudoku, and Sokoban, each representing distinct cognitive categories. Each game displays two VQA examples consisting of: (a) current game state visualization, (b) a targeted question, and (c) step-by-step reasoning with the answer. GameQA transforms complex game-playing tasks into this structured VQA format. See Appendix J for more VQA examples of some representative games.

For a task, we use LLMs to assist the design and refinement of the QA template, an example prompt for such refinement is illustrated in Figure 1 (Step 2). Additionally, LLMs can also design tasks, with example prompt shown in Appendix G.2. In summary, designing a QA template is equivalent to extracting one type of reasoning pattern from the video game code obtained in the first step. The third step then further instantiates these reasoning task templates into a dataset.

## 2.3 DATA ENGINE CONSTRUCTION FOR DATA SAMPLES GENERATION

The third step is data engine construction for samples generation. The data engine is a program that, when executed, automatically generates task instances in batch according to task templates. LLMs are used to assist in constructing the data engine's code based on the game code obtained in the first step. The prompt for guiding LLMs to construct the data engine code is shown in Figure 1 (Step 3). The pseudocode for the data engine is shown in Figure 1 (Step 3). Taking Sokoban's state prediction task as an example, the data engine program is constructed from four core modules:

- **Game environment initialization:** This module randomly generates Sokoban environment by directly reusing the initialization functions from the Step 1's game code. Sokoban environment includes the position of player, boxes, goals and walls.

- **Proposing task instance:** This module proposes a state prediction task instance by generating a multi-step random movement sequence.

- **Solving task instance:** This module generates a solution using a corresponding algorithm. For state prediction task, the algorithm simulates each action by reusing the movement logic from game code of Step 1, which handles all movement situations including collisions and box pushing.

- **QA data construction:** This module fills the task templates. The movement sequence will be filled into the question template to become a question instance. The state transition trajectory will be filled into the answer template to become an answer instance.

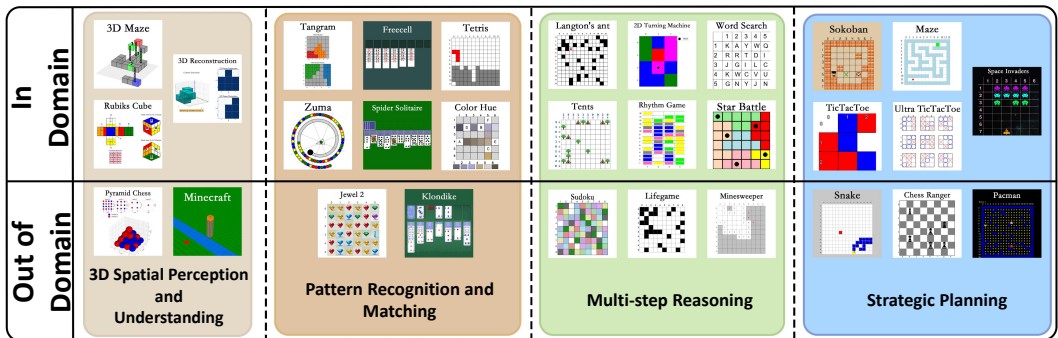

Figure 3: Overview of the GameQA dataset. The 30 games in GameQA are classified into four categories based on the core abilities required to solve game tasks. Appendix F.2 provides definitions of these four game categories. The games chose as out-of-domain are not used for training; instead, they are used to test the generalization performance of the model trained on the in-domain games.

## 2.4 DATA QUALITY ASSESSMENT AND DATA AUGMENTATION

We implemented manual quality verifications at each step of our method.

**Game code verification:** We verify the correctness of the game code generated by the models by manually running the game program. If errors are found, including vision issues in the game display or code logic errors, they are fed back to the LLM for regeneration. For complex game features that LLMs cannot generate correctly, we retrieved relevant open-source code and provided it to the LLM. Games created with external code are shown in Appendix E.4.

**Data engine validation:** During the data engine development process, LLMs generate an initial version based on the prompt. This version is then manually tested. If the generated reasoning questions and answers do not conform to the templates from the second step or contain errors, the LLM will be instructed to regenerate the data engine.

**Data augmentation and filtering:** Once the data engine is constructed, data samples can be easily generated in batch, yet the answer processes often exhibit repetitive patterns. We therefore employed LLM paraphrasing to perform data augmentation, detailed in Appendix F.3. Additionally, we applied data filtering to ensure the augmented samples were correct, of appropriate length, and without excessive textual repetition. Details of our data quality assurance process is shown in Appendix F.4. Time spent on these main steps for each game in GameQA is detailed in Appendix E.3.

## 3 THE GAMEQA DATASET

The GameQA dataset is a visual-language question answering dataset that transforms game-playing tasks into a Visual Question Answering (VQA) format, as shown in Figure 2. GameQA includes 30 games, 158 distinct tasks and approximately 140K questions in total.

**Dataset composition:** The dataset has 30 different games classified into four categories based on the core abilities required to solve game tasks: "3D Spatial Perception and Understanding", "Pattern Recognition and Matching", "Multi-step Reasoning", and "Strategic Planning", as detailed in Appendix F.2. The criteria on choosing these 30 games are listed in Appendix E.2. As illustrated in Figure 3, the 30 games are divided into two sets: 20 In-Domain games for model training and 10 Out-of-Domain games for testing generalization performance. The Out-of-Domain set was held out during training.

Table 2: GameQA train and test set statistics summary. Full table in Appendix F.1.

| Statistic Category | Train Set | Test Set |
|---|---|---|
| Total Games | 20 | 30 |
| In-Domain | 20 | 20 |
| Out-of-Domain | 0 | 10 |
| Total Tasks | 102 | 158 |
| Total Questions | 126,760 | 15,047 |
| Multiple Choice | 86,520 | 10,518 |
| Avg. Choices | 7.10 | 7.05 |
| Fill-in-the-Blank | 40,240 | 4,529 |
| Unique Images | 74,620 | 8,620 |

**All tasks are verifiable:** All questions in the dataset are either multiple-choice or fill-in-the-blank. The multiple-choice questions typically have 7-8 options, while the fill-in-the-blank questions require simple answers such as numbers or coordinates, with detailed statistics shown in Table 2. The specific verification method in our training and quantified verification correctness are in Section 4.2.

**Difficulty levels:** Each task is assigned with one of three difficulty levels of questions ("QA Level"). And data samples are divided into 3 difficulty levels based on image complexity ("Plot Level"), namely game state or grid size, controlled by code parameters, providing three perception and reasoning difficulty levels for the tasks. Examples demonstrating these difficulty levels are in Appendix J. The training and evaluation result about difficulty are shown in Tables 13 and 7.

## 4 GAME-RL: RL ON GAME TASKS

### 4.1 RL ALGORITHM

We use the Group Relative Policy Optimization (GRPO) (Shao et al., 2024) as our algorithm. We use the standard formulation from DeepSeek, with the loss function shown below.

$$\mathcal{J}_{\text{GRPO}}(\theta) = \mathbb{E}[q \sim P(Q), \{o_i\}_{i=1}^{G} \sim \pi_{\theta_{old}}(O|q)] \frac{1}{G} \sum_{i=1}^{G} \frac{1}{|o_i|} \sum_{t=1}^{|o_i|}$$

$$\left\{ \min \left[ \frac{\pi_\theta(o_{i,t}|q, o_{i,<t})}{\pi_{\theta_{old}}(o_{i,t}|q, o_{i,<t})} \hat{A}_{i,t}, \text{clip} \left( \frac{\pi_\theta(o_{i,t}|q, o_{i,<t})}{\pi_{\theta_{old}}(o_{i,t}|q, o_{i,<t})}, 1 - \epsilon, 1 + \epsilon \right) \hat{A}_{i,t} \right] - \beta \mathbb{D}_{KL}[\pi_\theta || \pi_{\text{ref}}] \right\}$$

### 4.2 REWARD DESIGN

**LLM as a judge.** Although the GameQA tasks are verifiable, the answer of a question can have different expression forms (e.g., (2, 3) vs. "x=2, y=3"), making traditional rule-based judge methods (e.g., text matching) suffer from insufficient accuracy. We therefore use Qwen2.5-32B-Instruct-AWQ (Qwen et al., 2025) (the AWQ-quantized version) as an evaluator model (Zheng et al., 2023). The evaluator model solely determines if the final answer is semantically equivalent to the ground truth, with details and prompt shown in H.2. Manual check on 300 randomly sampled cases confirms 100% verification accuracy, validating the evaluator reliably reflects answer correctness.

**Outcome reward design of RL.** The reward signal used for training is solely based on the correctness of the model's **final answer**. A reward of 1 is assigned if the evaluator model determines that the final answer semantically matches the ground truth, and a reward of 0 is assigned otherwise.

## 5 EXPERIMENT

### 5.1 EXPERIMENT SETTINGS

The detailed description of data preparation, training processes, and evaluation method can be found in Appendix C.

**Hyperparameters of GRPO training.** We rollout 12 samples per questions. The model was trained for one epoch. The learning rate was 2e-7, with a 5% warm-up. The clipping value $\epsilon$ was set to 0.2 and the KL-divergence coefficient $\beta$ was set to 0.04.

**Benchmarks.** We evaluated the models on a set of vision-language reasoning benchmarks consisting of our GameQA test set and public general vision reasoning benchmarks.

- **GameQA benchmark.** This test set includes around 500 question-answer pairs for each of 30 games, totaling 15,047 samples.
- **General benchmarks.** We use the MMMU validation set (Yue et al., 2024a) for testing general multimodal understanding, and include MMMU-Pro (Yue et al., 2024b), which features 10-option multiple-choice questions. To assess mathematical reasoning in visual contexts, we use Math-Vista (Lu et al., 2024) (testmini, 1,000 samples), MathVerse (Zhang et al., 2024b) (testmini, 3,940

Table 3: Evaluation results on general vision benchmarks, which shows game-only training enhances general reasoning capability of VLMs. We fine-tune three VLMs (Qwen2.5-VL-7B (Bai et al., 2025), InternVL2.5-8B (Chen et al., 2024) and InternVL3-8B (Chen et al., 2025)) on 5K GameQA samples using GRPO, resulting in the models Game-RL-Qwen2.5-VL-7B, Game-RL-InternVL2.5-8B and Game-RL-InternVL3-8B, respectively. Performance improvement compared to the vanilla model is denoted by (↑). Best performance per section is in bold. Evaluation details: Appendix C.4.

| Models | Avg. (↑) | MathVista | MathVerse | MMBench | MMMU | CharXiv | MathVision | MMMU-Pro |
|---|---|---|---|---|---|---|---|---|
| Baseline | | | | | | | | |
| Random | 14.84 | 17.90 | 12.74 | 26.37 | 24.67 | 0.00 | 10.03 | 12.19 |
| Proprietary Multimodal Large Language Models | | | | | | | | |
| GPT-4o | 56.9 | 63.8 | 50.2 | 86.0 | 69.1 | 47.1 | 30.4 | 51.9 |
| Claude-3.5-Sonnet | 59.5 | 67.7 | 56.8 | 78.5 | 68.3 | 60.2 | 33.3 | 51.5 |
| Gemini-2.5-Pro | **73.1** | **77.7** | **65.9** | **88.3** | **79.7** | **62.9** | **66.0** | **71.2** |
| Open-Source Multimodal Large Language Models | | | | | | | | |
| Qwen2.5-VL-32B | 60.70 | **77.40** | **60.41** | 88.13 | 63.83 | 47.50 | 33.80 | 53.83 |
| Ovis2-34B | 57.91 | 71.50 | 53.71 | **88.73** | 60.91 | **49.10** | 35.93 | 45.48 |
| InternVL2.5-38B | 55.62 | 68.60 | 48.38 | 86.93 | 57.53 | 41.50 | 39.40 | 46.98 |
| LLaVA-OV-72B | 49.51 | 58.60 | 46.85 | 83.13 | 52.74 | 35.20 | 33.87 | 36.18 |
| Qwen2.5-VL-72B | **60.95** | 75.50 | 56.87 | 86.80 | **65.34** | 48.10 | **40.60** | 53.45 |
| InternVL2.5-78B | 57.96 | 70.20 | 52.34 | 88.33 | 61.84 | 42.70 | 39.93 | 50.38 |
| Qwen2.5-VL-7B | 50.00 | 66.62 | 45.10 | 84.05 | 49.78 | 37.92 | 30.17 | 36.32 |
| Game-RL-Qwen2.5-VL-7B | **52.65** (+2.65) | **68.48** | **48.60** | **85.00** | **51.96** | **42.08** | **32.48** | **39.99** |
| InternVL2.5-8B | 45.80 | 57.43 | 35.85 | 81.90 | 47.92 | 31.68 | 28.62 | 37.20 |
| Game-RL-InternVL2.5-8B | **48.40** (+2.60) | **62.38** | **38.72** | **82.18** | **48.91** | **35.66** | **31.87** | **39.08** |
| InternVL3-8B | 54.15 | 68.72 | 49.76 | 85.98 | 56.85 | 38.92 | 35.24 | 43.59 |
| Game-RL-InternVL3-8B | **56.05** (+1.90) | **73.24** | **51.40** | **86.36** | **57.82** | **40.75** | **38.05** | **45.10** |

Table 4: Training on GameQA leads to generalization competitive to using outstanding general multimodal reasoning datasets. Evaluation details: Appendix C.4.

| Models | Avg. (↑) | Out of Domain Games | | | | Avg. (↑) | General Vision Benchmarks | | | | | | |
|---|---|---|---|---|---|---|---|---|---|---|---|---|---|
| | | 3D Spatial Perc. & Under. | Pattern Recog. & Matching | Multi-step Reasoning | Strategic Planning | | Math Vista | Math Verse | MMBench | MMMU | CharXiv | Math Vision | MMMU-Pro |
| Qwen2.5-VL-7B | 27.09 | 23.60 | **29.20** | 26.21 | 29.34 | 49.94 | 66.80 | 45.08 | 83.67 | 49.01 | 37.70 | 30.80 | 36.49 |
| + MAVIS-8K (GRPO) | 27.61 (+0.52) | 26.80 | 28.25 | 28.42 | 26.98 | 51.53 (+1.59) | 67.90 | 46.16 | 83.62 | 50.45 | 39.20 | 34.98 | 38.42 |
| + Multimodal-Open-R1-8K (GRPO) | 28.33 (+1.24) | 24.87 | 27.86 | 29.93 | 30.64 | 51.86 (+1.92) | 67.63 | 48.09 | 83.78 | 49.78 | 40.20 | **34.89** | 38.65 |
| + MultiMath-8K (GRPO) | 28.38 (+1.29) | **28.10** | 28.45 | 27.45 | 29.53 | 52.81 (+2.87) | **69.36** | 47.99 | **84.13** | 53.44 | 40.83 | 33.92 | **39.91** |
| + GameQA-5K (GRPO) | 29.87 (+2.78) | 27.00 | 28.52 | 31.49 | 32.46 | 52.31 (+2.37) | 68.70 | **48.72** | 83.16 | 50.21 | **41.40** | 34.27 | 39.74 |
| + GameQA-5K, MultiMath-8K (GRPO) | **30.93** (+3.84) | 28.20 | 28.28 | **34.26** | 32.99 | **53.23** (+3.29) | 69.20 | 48.02 | **84.73** | 53.21 | 42.10 | 34.47 | **40.89** |

samples), and MathVision (Wang et al., 2024a) (open subset, 3,040 questions). For general visual understanding, we adopt MMBench (Liu et al., 2023) (validation set), and for chart-based reasoning, we use CharXiv (Wang et al., 2024b).

## 5.2 MAIN RESULTS

By analyzing the models' performance on in-domain games, unseen out-of-domain game tasks, and general vision-language benchmarks, we find that:

**Game-only training enhances general reasoning capability of VLMs.** Training on the GameQA dataset significantly improves performance on the in-domain games and exhibits strong generalization to the out-of-domain games (Table 5). These improvements further lead to performance gains on the interactive game environments from the ING-VP benchmark (Zhang et al., 2024a) (Table 6). Remarkably, all the three VLMs trained solely on game data show **robust generalization to broader general vision benchmarks**, achieving consistent performance gains across all seven diverse general vision reasoning benchmarks (Table 3). These results suggest that the models have successfully learned **transferable visual understanding and reasoning abilities** from the GameQA dataset.

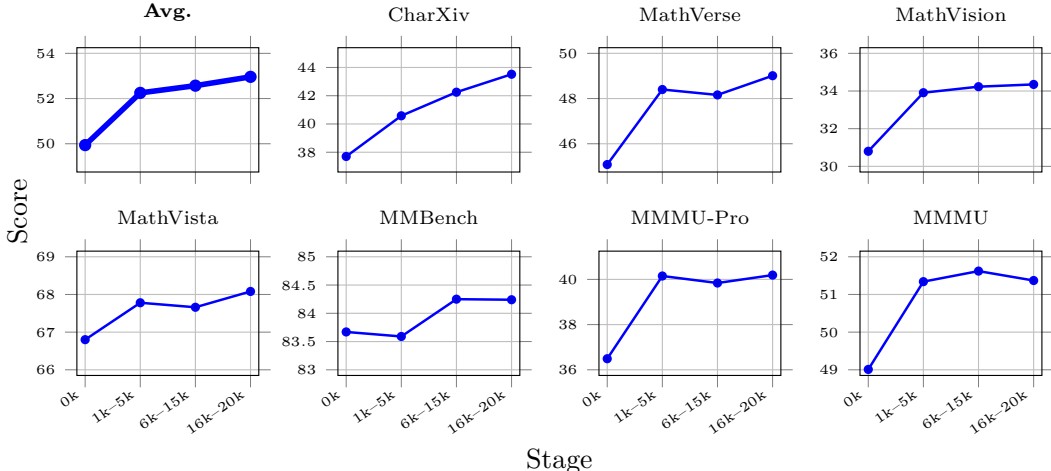

Figure 4: Scaling effect of game data volume on general vision benchmarks. We train Qwen2.5-VL-7B-Instruct on a total of 20k samples. For every 1000 samples trained, the model is evaluated. We divide results into 3 stages and average the scores per stage to show the upward trend more clearly.

> **Takeaway 1**  Training solely on GameQA enhances VLMs' general reasoning abilities, with robust generalization to unseen games (Table 5), interactive game environments (Table 6), and 7 diverse general vision reasoning benchmarks (Table 3). These results demonstrate the models have learned transferable visual understanding and reasoning abilities after game-data training.

**GameQA enables out-of-domain improvements competitive with using general multimodal reasoning datasets.** To better understand the value of GameQA, we compare it with outstanding general visual reasoning datasets, including MAVIS (Zhang et al., 2024c), Multimodal-Open-R1 (lmms lab, 2025) and MultiMath (Peng et al., 2024). MAVIS includes various geometry and function problems. Multimodal-Open-R1 is a geometry-centered dataset. MultiMath is a comprehensive and diverse multimodal math dataset. Based on Qwen2.5-VL-7B, we apply the same GRPO training on 5k GameQA samples, 8k MAVIS samples, 8k Multimodal-Open-R1 samples, 8k MultiMath samples, respectively, to conduct comparative training. The results (Table 4) show that despite using fewer training data (5k vs. 8k) that are also out-of-domain for the general benchmarks, the GameQA-trained model is competitive compared to the models trained on geometry or function data, where the general benchmarks would be considered in-domain. These results demonstrate that GameQA is a high-quality data source which enables strong out-of-domain generalization, suggesting the high potential of games in enhancing general reasoning abilities in models.

> **Takeaway 2**  Training on GameQA leads to performance gains comparable to using general multimodal reasoning datasets such as geometry datasets (Table 4). This highlights the effectiveness of GameQA as a high-quality data source, suggesting games may serve as valuable scenarios and resources to boost general reasoning abilities.

**GameQA enhances model performance in mixed training.** We train Qwen2.5-VL-7B using GRPO on a mixture of 5k GameQA and 8k MultiMath samples, as shown in Table 4. The results suggest that **GameQA can bring extra benefits when mixed with other dataset for training.**

## 5.3    SCALING EFFECT OF GAME DATA VOLUME ON GENERALIZATION

We train the Qwen2.5-VL-7B model on a GameQA subset of 20,000 samples from the 20 in-domain games using the GRPO method. As shown in Figure 4, the model's performance score demonstrates an overall upward trend on 7 general vision benchmarks as the amount of training game data increases. This indicates that **scaling up game data volume consistently enhances the VLM's generalizable reasoning abilities.**

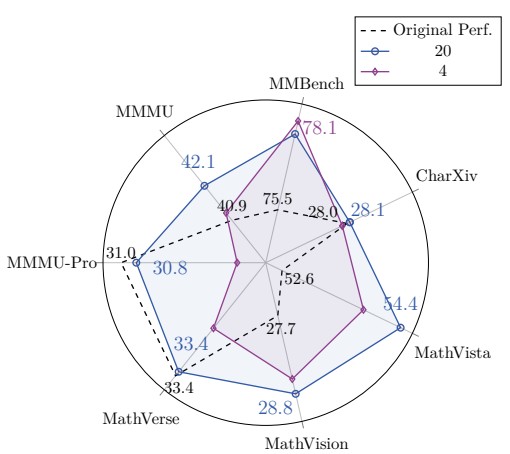
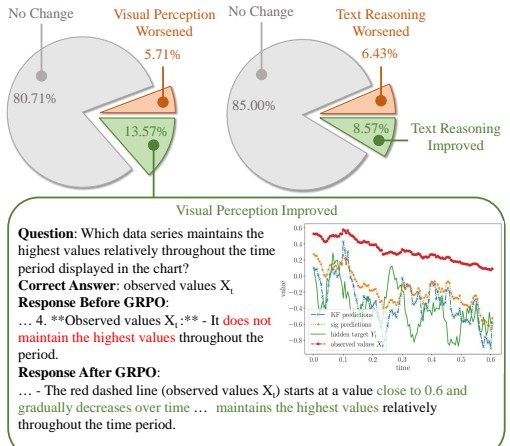

Figure 5: The scaling effect of game diversity. As the VLM is trained on an increasing number of distinct games from GameQA, the performance on general visual benchmarks improves.

Figure 6: Impact of Game-RL on visual perception and text reasoning abilities on general vision benchmarks. Two pie charts show both abilities improve, with a perception case below.

## 5.4 SCALING EFFECT OF GAME DIVERSITY ON GENERALIZATION

With around 5,000 total training samples, we trained the Qwen2.5-VL-3B on GameQA subsets with 4 and 20 distinct games (Table 8). Performance on general benchmarks improves as game diversity increases (Figure 5). This suggests that **scaling game diversity makes better generalization** (Xi et al., 2025), enabling the model to acquire more robust visual understanding and reasoning abilities.

> **Takeaway 3** Scaling up either game diversity (Figure 5) or game data volume (Figure 4) consistently improves VLMs' generalizable reasoning capabilities. These results suggest scaling reinforcement learning in game environments offers a promising pathway toward more generalizable multimodal reasoning in foundation models.

## 5.5 QUALITATIVE ANALYSIS

To confirm that GRPO on GameQA substantially enhances visual perception and text reasoning abilities of models, we manually analyzed 790 cases randomly sampled from the results of InternVL2.5-8B, containing responses before and after GRPO. The results (Figure 6) confirm that after GRPO, the model demonstrates **improved recognition of visual information and performs more precise text reasoning.** More statistics and cases are illustrated in Appendix B.3. In addition, our qualitative analysis of model performances across four game categories reveals common behaviors and challenges, detailed in Appendix I.

> **Takeaway 4** Beyond quantitative metrics, manual qualitative analysis (Figure 6) illustrates that Game-RL strengthens both visual perception and text reasoning capabilities of VLMs.

## 5.6 ADVANCED VLMS PERFORM NOTABLY WORSE THAN HUMANS ON GAMEQA

Both leading open-source and proprietary models achieve average accuracy levels considerably lower than those of human (Table 5). This clear difference highlights the difficulty and high requirements of the GameQA benchmark, requiring not only accurate visual comprehension of game scenes but also the ability to carry out multi-step logical reasoning. Furthermore, our qualitative analysis and case study in Appendix I reveal that even the most advanced models currently struggle to match human-level understanding, particularly in tasks demanding deep reasoning. More experiments are in Appendix D.

Table 5: Evaluation results on GameQA benchmark. In-domain and out-of-domain game category results are shown. The percentage of performance improvements compared to the vanilla model is denoted by (↑). Best performance per section is indicated in bold.

| Models | Avg. (↑) | In Domain Games | | | | Avg. (↑) | Out of Domain Games | | | |
|---|---|---|---|---|---|---|---|---|---|---|
| | | 3D Spatial Perc. & Under. | Pattern Recog. & Matching | Multi-step Reasoning | Strategic Planning | | 3D Spatial Perc. & Under. | Pattern Recog. & Matching | Multi-step Reasoning | Strategic Planning |
| *Baselines* | | | | | | | | | | |
| Human | **84.75** | **85.18** | **80.74** | **84.46** | **88.62** | **81.61** | **79.17** | **73.81** | **81.27** | **92.19** |
| Random | 11.90 | 11.69 | 12.04 | 10.24 | 13.61 | 9.68 | 7.11 | 9.50 | 11.83 | 10.27 |
| *Proprietary Multimodal Large Language Models* | | | | | | | | | | |
| GPT-4o | 40.52 | 32.01 | 34.81 | 50.67 | 44.59 | 43.81 | 48.90 | 36.91 | 48.58 | 40.87 |
| Claude-3.5-Sonnet | 47.69 | 37.41 | 43.16 | 56.09 | 54.11 | 50.34 | 51.30 | 43.62 | 60.42 | 46.01 |
| Claude-4-Sonnet | 46.58 | 31.12 | 39.73 | 66.90 | 48.57 | 55.16 | 45.60 | 56.58 | 63.28 | 55.17 |
| Gemini-2.5-Pro | **58.95** | **46.93** | **52.79** | **74.62** | **61.46** | **67.60** | **57.60** | **77.37** | **77.62** | **57.80** |
| *Open-Source Multimodal Large Language Models* | | | | | | | | | | |
| Ovis2-8B | 24.98 | 19.92 | 24.43 | 27.21 | 28.37 | 26.99 | 29.70 | 20.70 | 34.14 | 23.41 |
| InternVL3-9B | 26.89 | 21.86 | 22.53 | 32.54 | 30.65 | 26.73 | 25.20 | 30.38 | 32.14 | 19.18 |
| LLaMA3.2-11B-Vision | 19.69 | 19.12 | 16.48 | 21.30 | 21.86 | 18.04 | 18.40 | 14.92 | 17.73 | 21.11 |
| Qwen2.5-VL-32B | 34.09 | 28.26 | 30.99 | 40.27 | 36.83 | 35.97 | 32.90 | 33.02 | 44.03 | 33.94 |
| Ovis2-34B | 34.53 | 27.92 | 32.72 | 39.46 | 38.03 | 35.29 | 35.50 | 34.20 | 38.71 | 32.75 |
| InternVL2.5-38B | 30.04 | 23.39 | 25.86 | 36.82 | 34.08 | 32.42 | 30.60 | 33.96 | 39.35 | 25.79 |
| InternVL3-38B | 35.23 | 28.33 | 31.76 | 41.89 | 38.96 | 38.69 | 33.30 | 43.62 | 50.09 | 27.75 |
| LLaVA-OV-72B | 24.87 | 19.92 | 24.88 | 27.72 | 26.95 | 28.32 | 26.80 | 23.52 | 32.87 | 30.11 |
| Qwen2.5-VL-72B | 37.63 | 29.47 | 32.85 | 45.76 | 42.42 | 39.22 | **35.90** | 37.38 | 46.86 | **36.75** |
| InternVL2.5-78B | 32.35 | 27.15 | 28.84 | 39.53 | 33.90 | 35.30 | 32.80 | 37.26 | 42.41 | 28.75 |
| InternVL3-78B | **38.00** | **33.15** | **33.03** | **46.60** | 39.20 | **39.74** | 34.90 | 40.70 | **50.95** | 32.43 |
| InternVL2.5-8B | 22.22 | 20.39 | 17.18 | 25.34 | 25.97 | 20.05 | 20.80 | 22.45 | 18.88 | 18.07 |
| Game-RL-InternVL2.5-8B | **29.44** (+7.22) | 26.74 | 26.05 | 29.51 | 35.44 | **23.87** (+3.82) | **25.00** | 25.12 | 24.91 | 20.45 |
| InternVL3-8B | 26.51 | 22.53 | 21.91 | 30.18 | 31.43 | 27.64 | **29.60** | **27.44** | 29.62 | 23.91 |
| Game-RL-InternVL3-8B | **33.09** (+6.58) | 27.94 | 28.52 | 36.81 | 39.07 | **28.80** (+1.16) | 29.20 | 25.31 | 34.59 | 26.09 |
| Qwen2.5-VL-7B | 25.78 | 22.58 | 21.92 | 25.21 | 33.40 | 27.09 | 23.60 | 29.20 | 26.21 | 29.34 |
| Game-RL-Qwen2.5-VL-7B | **32.12** (+6.34) | 26.80 | 26.88 | 33.34 | 41.45 | **30.51** (+3.42) | 27.10 | 31.56 | 31.24 | 32.13 |

## 6 RELATED WORK

**Multimodal reasoning data construction.** Currently, the data construction methods are mainly divided into two categories: human expert supervision and automated synthesis. Peng et al. (2024) and Lu et al. (2021) collect visual reasoning problems from textbooks, Lu et al. (2023) constructs datasets through labeling by STEM students, but they are limited by the scarcity of high-quality data sources and the high cost of manual annotation. Lu et al. (2021); He et al. (2024); Gao et al. (2023); Shi et al. (2024) uses expert models to generate reasoning processes, but the results are limited by the performance of the expert model. Trinh et al. (2024) and Zhang et al. (2024c) synthesize geometric reasoning data through procedural methods, but these methods are often designed for specific domains and have high transfer costs. Table 1 provides a comparison of existing vision language reasoning datasets.

**Using game data to enhance VLMs' reasoning capabilities.** Game environments provide well-defined rules and mechanics that are easy to verify. However, prior work has primarily used games either to train task-specific agents or to build evaluation benchmarks, without exploring how to use game data to improve VLMs' general reasoning abilities. For instance, Reed et al. (2022) claims to train a generalist agent through pure supervised training, but it is difficult to generalize on out-of-domain game tasks. Meanwhile, Paglieri et al. (2024); Zhang et al. (2024a; 2025); Li et al. (2024) all establish gaming environments for VLMs, but these environments are used exclusively for evaluation purposes rather than training. These limitations indicate that how to effectively use game data to enhance the reasoning ability of visual language models remains a critical problem to be addressed. Table 1 provides a comparison of existing game reasoning benchmarks.

## 7 CONCLUSION

To explore broader training scenarios and resources for vision-language RL, we propose Game-RL to construct game tasks for VLMs' RL training. We also propose the novel Code2Logic approach to synthesize diverse verifiable game reasoning tasks, obtaining the GameQA dataset. Multiple VLMs trained through RL solely on GameQA achieve out-of-domain generalization across diverse general vision reasoning benchmarks. Our findings highlight game environments as a new RL scaling dimension for enhancing AI's generalizable reasoning.

STATEMENTS

**Ethics Statement.** This work adheres to the ICLR Code of Ethics. Our study does not involve human subjects, sensitive personal information, or private data. All datasets used in experiments were either synthesized by our proposed Code2Logic pipeline or publicly available benchmark datasets, and no additional ethical approvals were required. We have carefully considered issues of fairness, reproducibility, and research integrity. To minimize potential harm, we ensured that the synthesized data contains only game-based, rule-driven reasoning tasks without personal or discriminatory content.

**Reproducibility Statement.** We have taken multiple steps to ensure the reproducibility of our results. Section 3 provides the construction details of the GameQA dataset, including task categories, difficulty levels, and statistics. Section 2.1 and Appendix E.1 describe the Code2Logic pipeline in detail, with pseudocode and examples of task templates. Experimental settings, model training details, and evaluation metrics are reported in Appendix C. In addition, all prompts and augmentation procedures are provided in Appendix H.

ACKNOWLEDGEMENTS

This work was partly supported by the National Natural Science Foundation of China under Grant Nos. 62521004 and 62376061, and by the Shanghai Science and Technology Committee under Grant No. 24511103900.

The valuable efforts of the following individuals in the data synthesis and validation processes were of great importance to the development of this project: (Sorted by last name, then first name) Ruifeng Chen, Yingqian Huang, Yutong Ke, Hengxi Lin, Yuanhao Ni, Qingyun Shi, Haitian Wang, Xiaoyong Wang, Yufei You, Juntao Zhang, Weixin Zhang, Yang Zhang.

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

## APPENDIX CONTENTS

## A    LIMITATIONS AND FUTURE WORK

Using reasoning processes to conduct Supervised Fine-Tuning (SFT) has not achieved satisfactory out-of-domain generalization. Therefore, future work could explore methods to better leverage these **reasoning processes** to enhance the model's general capabilities, such as employing reinforcement learning based on process supervision. In addition, GameQA currently involves single-turn game question answering. Future work could focus on developing training and evaluation methods for **multi-turn** interactions in gaming scenarios.

## B    MORE ANALYSIS

### B.1    IMPROVEMENTS ON INTERACTIVE ENVIRONMENTS AFTER GAME-RL

We evaluate the models on the ING-VP benchmark (Zhang et al., 2024a) to assess the improvements in **interactive environments**. ING-VP is an out-of-domain game benchmark which includes six video games. The input is an image of the current game state, and the model outputs a specific move. The game state then changes accordingly, achieving the finish or generating the next input image for the models. Under the original benchmark setup, the model scores 0 across the games both before and after training. We therefore choose an easier setup.

**Reasoning improvements from RL training on GameQA transfer to interactive environments**, as shown from the results of Qwen2.5-VL-7B in Table 6. After Game-RL training, average completion rate improves from 12% to 24%, with substantial gains in Maze (18% to 36%) and Hanoi (26% to 62%). To understand these gains, we further analyze the single-step accuracy (i.e., the proportion of moves that lead closer to finish). For Maze, single-step accuracy increases from 49% to 54%; for Hanoi, from 33% to 43%. These single-step improvements cumulatively leads to the substantial increases in the overall task completion rates.

Table 6: Performance on ING-VP benchmark (task completion rate, %) before and after RL training on GameQA, demonstrating improvements on the sequential decision-making visual game tasks.

| Model | Average | Maze | Sokoban | 8 Queens | Hanoi | 15 Puzzle | Sudoku |
|---|---|---|---|---|---|---|---|
| Qwen2.5-VL-7B | 12 | 18 | 13 | 0 | 26 | 2 | 12 |
| Game-RL-Qwen2.5-VL-7B | 24 (+12) | 36 | 15 | 0 | 62 | 4 | 24 |

### B.2    GAMEQA EVALUATION RESULTS BY DIFFICULTY

The fine-grained difficulty gradation in the GameQA dataset enables a more systematic evaluation of the models. As shown in Table 7, when either QA Level or Plot Level **increases**, the models' accuracy scores generally show a noticeable **decrease**.

### B.3    ERROR TYPE ANALYSIS

We **manually check** the reasoning process of 650 GameQA cases randomly sampled from InternVL2.5-8B before and after GRPO to identify the error types. We categorize the errors as 3 types: Visual Perception Error, Text Reasoning Error and both. The results are presented in Figure 9. Besides, in Figures 7 and 8, we supplement Figure 6 with two examples of text reasoning improvement and another example of visual reasoning improvement after GRPO training.

Table 7: Evaluation results on GameQA benchmark by **difficulty**. Scores are broken down by question difficulty (QA Level) and image complexity (Plot Level) within in-domain and out-of-domain game sets. The percentage of performance improvements compared to the vanilla model is denoted by (↑). Best performance per section is indicated in bold.

| Models | Avg. (↑) | In Domain Games by Difficulty | | | | | | Avg. (↑) | Out of Domain Games by Difficulty | | | | | |
|---|---|---|---|---|---|---|---|---|---|---|---|---|---|---|
| | | QA Level | | | Plot Level | | | | QA Level | | | Plot Level | | |
| | | Easy | Medium | Hard | Easy | Medium | Hard | | Easy | Medium | Hard | Easy | Medium | Hard |
| Proprietary Multimodal Large Language Models | | | | | | | | | | | | | | |
| GPT-4o | 41.56 | 48.10 | 40.63 | 32.95 | 47.69 | 38.60 | 37.80 | 44.03 | 55.63 | 44.45 | 32.05 | 54.49 | 42.98 | 33.71 |
| Claude-3.5-Sonnet | 48.90 | 59.26 | 46.07 | 36.70 | 55.49 | 45.97 | 43.80 | 50.94 | 62.80 | 46.15 | 43.80 | 60.58 | 49.19 | 42.26 |
| Claude-4-Sonnet | 48.80 | 58.36 | 50.14 | 42.23 | 56.45 | 48.99 | 47.81 | 56.03 | 66.71 | 55.91 | 45.45 | 62.80 | 54.31 | 50.37 |
| Gemini-2.5-Pro | **60.62** | **67.43** | **62.36** | **56.21** | **67.48** | **62.42** | **57.72** | **67.68** | **74.47** | **67.56** | **61.10** | **74.23** | **66.43** | **61.90** |
| Open-Source Multimodal Large Language Models | | | | | | | | | | | | | | |
| Ovis2-8B | 25.56 | 33.67 | 22.80 | 20.21 | 28.30 | 26.12 | 23.69 | 27.37 | 32.41 | 28.87 | 20.90 | 33.96 | 25.56 | 22.06 |
| InternVL3-9B | 27.45 | 34.92 | 24.20 | 21.15 | 30.42 | 26.94 | 24.43 | 26.54 | 30.45 | 23.23 | 25.86 | 34.41 | 23.93 | 20.63 |
| LLaMA3.2-11B-Vision | 19.67 | 24.51 | 18.54 | 15.96 | 20.71 | 20.22 | 19.16 | 18.35 | 19.37 | 20.32 | 15.41 | 20.99 | 18.14 | 15.68 |
| Qwen2.5-VL-32B | 34.80 | 41.52 | 33.43 | 27.18 | 38.07 | 34.93 | 31.09 | 36.59 | 47.99 | 33.60 | 28.16 | 46.87 | 32.25 | 29.86 |
| Ovis2-34B | 35.32 | 46.36 | 31.81 | 25.11 | 39.57 | 34.70 | 31.70 | 35.40 | 46.86 | 30.02 | 29.22 | 42.95 | 33.76 | 28.87 |
| InternVL2.5-38B | 30.83 | 40.08 | 28.21 | 22.68 | 35.55 | 30.36 | 27.23 | 32.50 | 38.80 | 29.71 | 28.93 | 41.52 | 30.74 | 24.47 |
| InternVL3-38B | 36.07 | 44.05 | 31.73 | **30.60** | 40.70 | 36.47 | 30.20 | 38.78 | 50.00 | 36.14 | 30.17 | 49.03 | 35.62 | 30.86 |
| LLaVA-OV-72B | 25.49 | 35.50 | 21.21 | 17.57 | 27.81 | 24.66 | 23.87 | 28.98 | 38.80 | 23.83 | 24.20 | 34.07 | 27.00 | 25.46 |
| Qwen2.5-VL-72B | 38.58 | 47.61 | **36.70** | 27.38 | 43.50 | **37.67** | 33.43 | 39.77 | **52.90** | **37.90** | 28.51 | 49.15 | 37.13 | **32.28** |
| InternVL2.5-78B | 33.05 | 41.90 | 30.58 | 24.00 | 36.58 | 32.30 | 30.00 | 35.40 | 43.96 | 34.57 | 27.69 | 45.22 | 32.97 | 27.20 |
| InternVL3-78B | **38.64** | **47.64** | 35.60 | 30.47 | **43.91** | 37.34 | **34.57** | **40.17** | 52.37 | 35.84 | **32.23** | **50.34** | **39.36** | 29.93 |
| InternVL2.5-8B | 22.31 | 27.12 | 21.98 | 18.35 | 24.65 | 22.66 | 21.37 | 19.78 | 18.36 | 21.29 | 19.72 | 24.52 | 17.78 | 16.67 |
| + GameQA (SFT) | **47.33** (+25.02) | **56.59** | **45.36** | **39.79** | **53.73** | **45.50** | **44.76** | **26.10** (+6.32) | **27.84** | 21.95 | **28.39** | **30.43** | **27.37** | **20.07** |
| + GameQA (GRPO) | 29.52 (+7.21) | 36.37 | 26.95 | 25.90 | 34.23 | 28.14 | 27.91 | 23.67 (+3.90) | 24.41 | **22.92** | 23.67 | 28.44 | 22.85 | 19.33 |
| InternVL3-8B | 26.85 | 33.82 | 26.37 | 19.79 | 28.91 | 28.20 | 24.90 | 27.49 | 32.29 | 26.62 | 23.55 | 32.59 | 26.46 | 22.99 |
| + GameQA (SFT) | **51.08** (+24.23) | **63.08** | **47.66** | **43.22** | **58.93** | **48.69** | **48.74** | 27.35 (-0.14) | **37.56** | 22.07 | 22.31 | **35.38** | 24.83 | 21.19 |
| + GameQA (GRPO) | 33.53 (+6.68) | 39.73 | 32.17 | 28.99 | 37.16 | 32.37 | 32.80 | **29.14** (+1.65) | 34.36 | **29.41** | **23.67** | 34.81 | **27.31** | **24.85** |
| Qwen2.5-VL-7B | 26.02 | 36.25 | 23.11 | 17.75 | 28.29 | 25.69 | 25.67 | 27.25 | 31.87 | 25.83 | 24.03 | 33.73 | 25.38 | 22.12 |
| + GameQA (SFT) | **49.23** (+23.21) | **60.19** | **47.47** | **39.13** | **55.56** | **47.65** | **46.60** | 30.33 (+3.08) | **42.00** | 25.35 | 23.55 | 36.86 | **29.29** | 24.29 |
| + GameQA (GRPO) | 32.41 (+6.39) | 42.51 | 30.25 | 23.96 | 35.05 | 32.40 | 31.79 | **30.77** (+3.52) | 38.68 | **25.96** | **27.57** | **38.17** | 28.87 | **24.66** |

# C    EXPERIMENT DETAILS

## C.1    HUMAN AND RANDOM BASELINES

We included two baselines for comparison: human and random.

- **Human Baseline**: We select approximately 20 questions from each of the 30 games, resulting in 623 questions. These are grouped into 30 sets and assigned to STEM undergraduates unfamiliar with the games. Each question is presented in a PowerPoint slide (Figure 10) using `python-pptx`[1], and responses are collected via an online questionnaire.[2]

- **Random Baseline**: This represents the lower performance bound, calculated as the expected score from random guessing on multiple-choice questions, with fill-in-the-blank tasks contributing zero.

## C.2    DATA PREPARATION

### C.2.1    SFT DATA PREPARATION

- **Game data.** First, we use our data engine to generate 5K task instances from 20 in-domain games, obtaining 100K samples in total. Then we perform data augmentation, as shown in Appendix F.3. Finally, we filter these samples as detail in Appendix F.4. A total of 40K samples are selected for SFT.

- **Other datasets.** We prepare the "Geo-Multi" dataset for data quality comparison in SFT experiments. Geo-Multi consists of 40K samples, with 20K randomly sampled from Multi-Math300K (Peng et al., 2024) and another 20K from Geo170K (Gao et al., 2023).

### C.2.2    REINFORCEMENT LEARNING DATA PREPARATION

- **GameQA data.** We sample 5K samples from 20 in-domain games. This sample size is smaller than that used for SFT to balance performance and computational cost, as the GRPO training process is relatively resource-intensive.

---

[1]https://python-pptx.readthedocs.io/en/latest/
[2]https://www.wjx.cn/

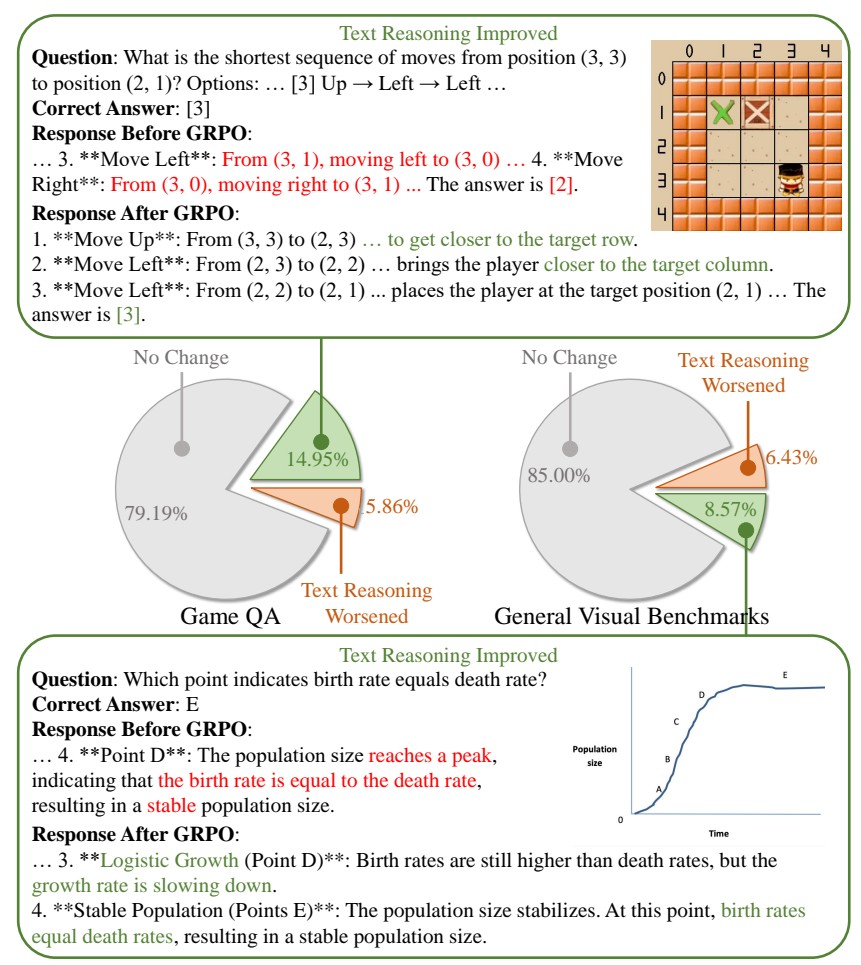

Figure 7: Impact of Game-RL on text reasoning performance, with two improvement examples.

Table 8: Selection of games for the game diversity scaling experiment.

| Game Set | 3D Spatial Perception and Understanding | Pattern Recognition and Matching | Multi-step Reasoning | Strategic Planning |
|---|---|---|---|---|
| 4 Games | 3D Reconstruction | Tangram | Word Search | TicTacToe |
| 20 Games (include 4 above + 16 new) | 3D Maze, Rubik's Cube | Spider Solitaire, Freecell, Tetris, Zuma, Color Hue | Tents, 2D Turing Machine, Langton's Ant, Rhythm Game, Star Battle | Sokoban, Space Invaders, Maze, Ultra TicTacToe |

- **Other datasets.** Details on the general reasoning datasets used for comparative training (Table 5.2) are as follows. We sample 4K samples from the geometry and function split of MAVIS (Zhang et al., 2024c), respectively. For multimodal-open-r1-8k-verified (lmms lab, 2025), we use all the samples. For MultiMath (Peng et al., 2024), we sample 8K samples.

To investigate the impact of game diversity on model generalization, we construct a subset of 4 games in addition to the full 20 in-domain training games. Table 8 details the specific games included.

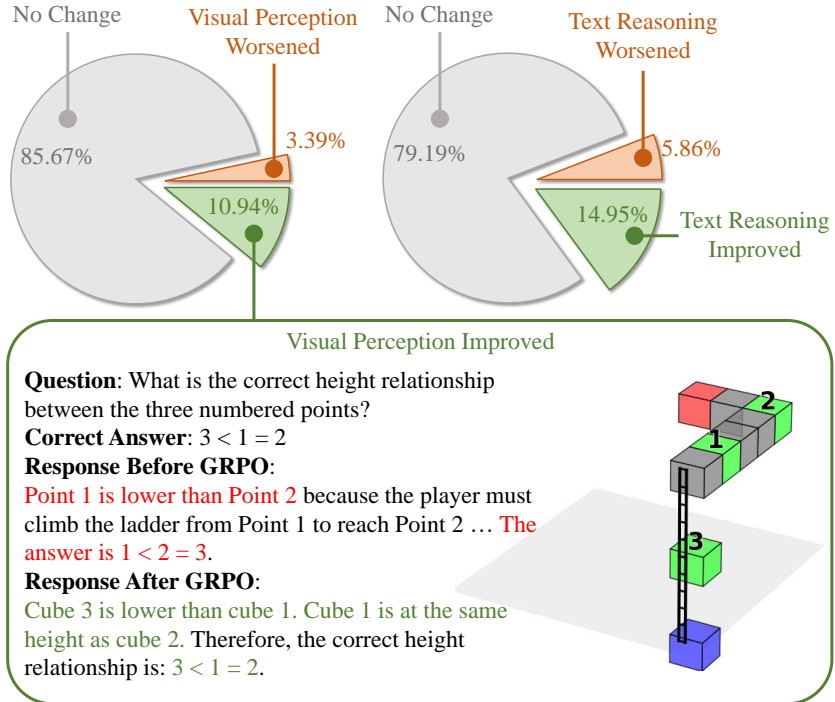

Figure 8: Impact of Game-RL on visual reasoning performance on GameQA.

## C.3 TRAINING AND EVALUATION DETAILS

### C.3.1 MODELS

We trained three VLMs, Qwen2.5-VL-7B (Bai et al., 2025), InternVL2.5-8B (Chen et al., 2024) and InternVL3-8B (Chen et al., 2025) on our data.

### C.3.2 TRAINING HYPERPARAMETERS

**LoRA-based supervised fine-tuning hyperparameters.** In this setup, Low-Rank Adaptation (LoRA) Hu et al. (2021) was applied to all linear layers of the language model. We trained the model for one epoch. The rank was set to 16, with alpha set to 32. The learning rate was 5e-5, including a 3% warm-up period followed by a constant rate.

**GRPO-based reinforcement learning.** In this setup, we conducted full-parameter fine-tuning of the language model while freezing the visual encoder and projection layers. We rollout 12 samples per questions. We trained the model for one epoch. The learning rate was 2e-7, with a 5% warm-up. The clipping value $\epsilon$ was set to 0.2 and the KL-divergence coefficient $\beta$ was set to 0.04. More hyperparameters are listed in Table 9.

### C.3.3 COMPUTE RESOURCES

**LoRA-Based Supervised Fine-Tuning.** For the models in the 7-8 billion parameter range, this LoRA-based SFT training was conducted on a single A800 GPU and the training duration lasted around 15 hours or less for 40k samples (1 epoch).

**GRPO-Based Reinforcement Learning.**

- **7-8 billion parameter model** Training on 5k samples (1 epoch) required approximately 22 hours, utilizing five A800 GPUs, including resources for the deployment of the evaluator model.

- **3 billion parameter model** The training process lasted approximately 18 hours for 5k samples (1 epoch) on four A800 GPUs, also including the evaluator model's deployment.

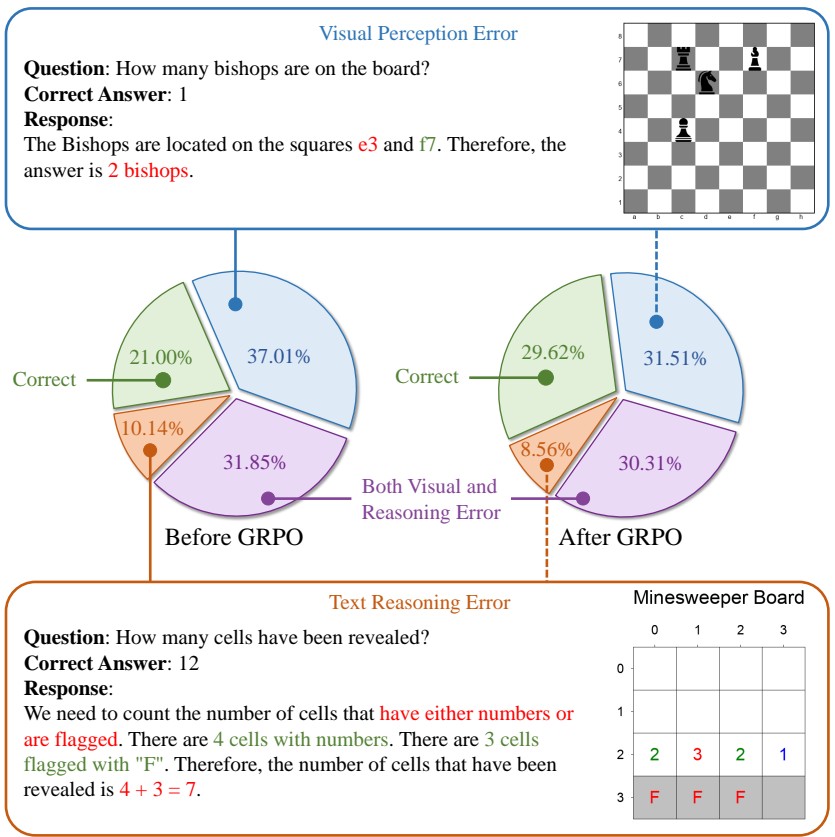

Figure 9: InternVL2.5-8B error types before and after GRPO show that Game-RL increases correct ratio and reduces visual and reasoning errors. Two cases show the two types of error in detail. Solid and dash lines connected to two cases means both are before training.

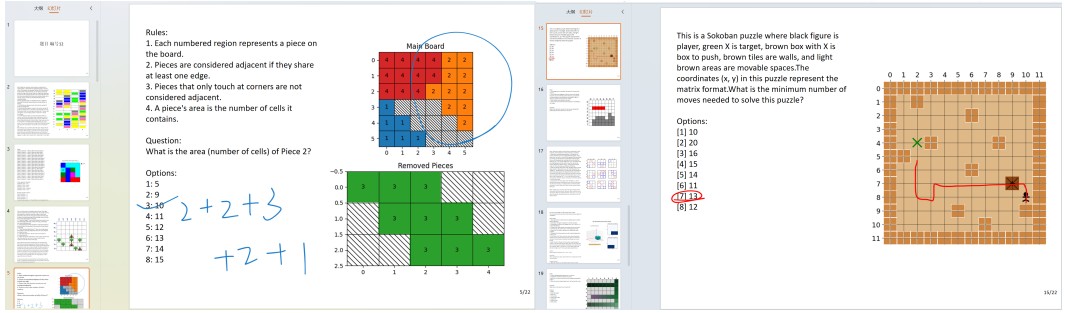

Figure 10: Two example PowerPoint slides demonstrating how the student participants, during the human baseline evaluation, took the tests and might write or draw on the slides.

**GPU usage.** Training a 7-8 billion parameter model with GRPO-based RL required approximately 22 hours, utilizing five A800 GPUs. This GPU allocation included resources for the deployment of the evaluator model. For a 3 billion parameter model, the training process lasted approximately 18 hours on four A800 GPUs, with this count also inclusive of the evaluator model's deployment. For the 32 billion parameter model, the LoRA-based training took approximately 22 hours using eight H20 GPUs with 141GB memory each, while the reward model was deployed on four A100 GPUs.

Table 9: GRPO Hyperparameters. Some hyperparameters are different when training 32B model.

| Hyperparameter | Value |
|---|---|
| Learning Rate (non-32B) | 2e-7 |
| Learning Rate (32B) | 5e-7 |
| Batch Size (non-32B) | 3 |
| Batch Size (32B) | 16 |
| KL-divergence Coefficient ($\beta$) (non-32B) | 0.04 |
| KL-divergence Coefficient ($\beta$) (32B) | 0.001 |
| Number of Generations | 12 |
| Temperature | 1.0 |
| Top-p | 0.85 |
| Top-k | 50 |
| Optimizer | AdamW |
| Warm-up Ratio | 0.05 |
| Weight Decay | 0.1 |

### C.4 INFERENCE AND EVALUATION

Besides trained models, we also evaluated proprietary large-scale models such as GPT-4o (20240806) (OpenAI, 2024) and Claude 3.5 Sonnet (20241022) (Anthropic, 2024), and open-source models that represent the current state of the art, including InternVL3-78B and Qwen2.5-VL-72B.

The inference and evaluation configurations are unified across the original models and our trained models.

- **Inference:** The inference `temperature` parameter is set to 0.2. The prompt template is shown in Appendix H.3.

- **Evaluation:** We evaluate the generated answers using the LLM-as-a-judge approach (Zheng et al., 2023), with Qwen2.5-72B-AWQ acting as the evaluator and the evaluation prompt shown in H.4.

For the main RL experiments (Tables 3 and 5) and the SFT experiments (Appendix D.1), we report the single checkpoint that achieves the highest validation accuracy. The validation set is a 1% hold-out split from the training data.

For the results in Table 3, both the three trained models (Game-RL-Qwen2.5-VL-7B, Game-RL-InternVL2.5-8B, Game-RL-InternVL3-8B) and their corresponding baselines (Qwen2.5-VL-7B, InternVL2.5-8B, InternVL3-8B) are evaluated by averaging over five inference runs with different seeds (13, 65, 117, 15, 67) to ensure the improvements are robust.

For comparative training (Table 4) and scaling effect analyses of game diversity (Figure 5), we evaluate the top three checkpoints (based on validation accuracy) and report their average test performance. Averaging across multiple top-performing checkpoints reduces training noise and yields more robust estimates.

## D MORE EXPERIMENTS

### D.1 SFT EXPERIMENTS

As described in Appendix C, we perform additional supervised fine-tuning (SFT) experiments on the three VLMs. The results of these experiments are presented in Tables 10 and 11, and we find:

**The GRPO demonstrates an advantage over SFT in out-of-domain generalization, though SFT yields strong in-domain gains.** When evaluating model performance, the effects of SFT and GRPO training methods showed clear differences across domains. Specifically, SFT training with the GameQA dataset led to substantial in-domain improvements: the InternVL2.5-8B and Qwen2.5-VL-7B models improved their average accuracy by 24.51% and 22.59% respectively on a test set covering 20 games, reaching final scores of 46.73% and 48.37% (Table 10). However, while SFT

Table 10: Evaluation results on GameQA benchmark. In-domain and out-of-domain game category results are shown. Performance improvement compared to the vanilla model is denoted by (↑). Best performance per section is indicated in bold.

| Models | Avg. (↑) | In Domain Games | | | | Avg. (↑) | Out of Domain Games | | | |
| --- | --- | --- | --- | --- | --- | --- | --- | --- | --- | --- |
| | | 3D Spatial Perc. & Under. | Pattern Recog. & Matching | Multi-step Reasoning | Strategic Planning | | 3D Spatial Perc. & Under. | Pattern Recog. & Matching | Multi-step Reasoning | Strategic Planning |
| InternVL2.5-8B | 22.22 | 20.39 | 17.18 | 25.34 | 25.97 | 20.05 | 20.80 | 22.45 | 18.88 | 18.07 |
| + GameQA (SFT) | 46.73 (+24.51) | 42.38 | 45.55 | 51.56 | 47.44 | 25.81 (+5.76) | 24.40 | 24.69 | 25.90 | 28.25 |
| + GameQA (GRPO) | 29.44 (+7.21) | 26.74 | 26.05 | 29.51 | 35.44 | 23.87 (+3.82) | 25.00 | 25.12 | 24.91 | 20.45 |
| InternVL3-8B | 26.51 | 22.53 | 21.91 | 30.18 | 31.43 | 27.64 | 29.60 | 27.44 | 29.62 | 23.91 |
| + GameQA (SFT) | 50.72 (+24.21) | 46.91 | 45.42 | 55.91 | 54.63 | 26.02 (-1.62) | 24.20 | 14.80 | 32.03 | 33.05 |
| + GameQA (GRPO) | 33.09 (+6.58) | 27.94 | 28.52 | 36.81 | 39.07 | 28.80 (+1.15) | 29.20 | 25.31 | 34.59 | 26.09 |
| Qwen2.5-VL-7B | 25.78 | 22.58 | 21.92 | 25.21 | 33.40 | 27.09 | 23.60 | 29.20 | 26.21 | 29.34 |
| + GameQA (SFT) | 48.37 (+22.59) | 42.05 | 42.82 | 58.66 | 49.96 | 29.27 (+2.18) | 26.80 | 21.16 | 31.79 | 37.32 |
| + GameQA (GRPO) | 32.12 (+6.34) | 26.80 | 26.88 | 33.34 | 41.45 | 30.51 (+3.42) | 27.10 | 31.56 | 31.24 | 32.13 |

Table 11: Evaluation results on general vision benchmarks. Performance improvement compared to the vanilla model is denoted by (↑). Best performance per section is indicated in bold.

| Models | Avg. (↑) | MathVista | MathVerse | MMBench | MMMU | CharXiv | MathVision | MMMU-Pro |
| --- | --- | --- | --- | --- | --- | --- | --- | --- |
| InternVL2.5-8B | 45.89 | 57.50 | 36.04 | 81.93 | 47.96 | 31.70 | 28.87 | 37.25 |
| + GameQA (SFT) | 45.06 (-0.83) | 58.10 | 35.79 | 82.87 | 48.07 | 31.20 | 22.80 | 36.56 |
| + Geo-Multi (SFT) | 43.84 (-2.05) | 53.60 | 36.90 | 82.80 | 43.17 | 30.40 | 27.80 | 32.22 |
| + GameQA (GRPO) | 47.91 (+2.02) | 61.70 | 37.11 | 83.87 | 50.06 | 32.00 | 31.93 | 38.69 |
| InternVL3-8B | 54.48 | 69.10 | 50.10 | 86.00 | 57.88 | 39.10 | 35.33 | 43.84 |
| + GameQA (SFT) | 49.66 (-4.82) | 63.20 | 43.30 | 84.53 | 53.44 | 32.90 | 29.60 | 40.64 |
| + Geo-Multi (SFT) | 48.52 (-1.42) | 62.60 | 40.96 | 82.87 | 48.54 | 37.80 | 29.80 | 37.06 |
| + GameQA (GRPO) | 55.88 (+1.40) | 73.00 | 50.71 | 86.20 | 58.34 | 39.90 | 37.93 | 45.10 |
| Qwen2.5-VL-7B | 49.94 | 66.80 | 45.08 | 83.67 | 49.01 | 37.70 | 30.80 | 36.49 |
| + GameQA (SFT) | 47.26 (-2.72) | 63.00 | 37.31 | 83.07 | 47.49 | 38.60 | 25.73 | 35.62 |
| + Geo-Multi (SFT) | 48.52 (-1.42) | 62.60 | 40.96 | 82.87 | 48.54 | 37.80 | 29.80 | 37.06 |
| + GameQA (GRPO) | 52.27 (+2.33) | 68.20 | 47.97 | 83.53 | 50.53 | 42.70 | 33.07 | 39.89 |

excels at improving in-domain performance, it can sometimes lead to performance degradation on general tasks as seen in 10, a phenomenon known as "catastrophic forgetting". Crucially, for training sets that belong to the mathematics domain, such as Geo-Multi (see Appendix C.2), models trained on them can still exhibit a decline in general capabilities. This may reflect the tendency of the SFT method to overfit the model to specific domain data (Chu et al., 2025).

In contrast, when training on GameQA using the GRPO, models not only successfully avoided performance degradation on general tasks but also generally achieved performance improvements across multiple general visual benchmarks (Table 3). A typical example is the Qwen2.5-VL-7B model, which, after GRPO training, showed enhanced performance on challenging benchmarks such as MathVista, MathVerse, and CharXiv, reaching a level comparable to larger-scale models like InternVL2.5-38B.

**Pure RL outperforms SFT-then-RL in out-of-domain generalization.** To validate our choice of directly applying RL, we compare our approach to a two-stage SFT-then-RL pipeline on Qwen2.5-VL-7B. As shown in Table 12, while the two-stage pipeline yields strong performance on in-domain tasks, it leads to a significant performance drop of -2.5% on general vision benchmarks. This suggests that full-parameter SFT on a narrow, specialized domain like GameQA can cause catastrophic forgetting. In contrast, pure RL better preserves and enhances out-of-domain generalization capabilities, achieving a +2.33% improvement.

## D.2  TRAINING DATA DIFFICULTY EXPERIMENT

To analyze how reasoning difficulty impacts generalization, we categorized tasks into Easy (55-85% baseline accuracy), Medium (30-55%), and Hard (5-30%). We trained Qwen2.5-VL-7B on 5k samples from different difficulty combinations. Table 13 shows that training on a diverse mix of difficulties yields the best generalization. The model trained on 'Easy+Medium+Hard' samples achieves the highest average score (52.74), outperforming models trained on simpler subsets. This confirms that the complex reasoning patterns in GameQA are crucial for improving general reasoning.

Table 12: Comparison of training pipelines on Qwen2.5-VL-7B. SFT is performed on 20k samples, followed by GRPO on 5k. Performance on general benchmarks is shown as a relative change from the baseline.

| Training Stage | In-domain Games | Out-of-domain Games | General Benchmarks (Avg. Change) |
|---|---|---|---|
| Baseline (Before SFT) | 25.89 | 26.92 | 0.0% |
| SFT | 56.21 | 30.74 | -3.9% |
| SFT-then-RL | 58.08 | 31.96 | -2.5% |
| **Pure RL (Our Method)** | **32.12** | **30.51** | **+2.33%** |

Table 13: Impact of training data difficulty on generalization. Results are averaged over the top three checkpoints. The baseline score is 49.94.

| Training Data (5k) | Average | MathVista | MathVerse | MMBench | MMMU | CharXiv | MathVision | MMMU-Pro |
|---|---|---|---|---|---|---|---|---|
| Baseline | 49.94 | 66.80 | 45.08 | 83.67 | 49.01 | 37.70 | 30.80 | 36.49 |
| Easy | 52.40 | 67.87 | 47.93 | 83.91 | 51.58 | 41.77 | 33.73 | 40.03 |
| Medium | 51.95 | 67.30 | 48.10 | 83.64 | 50.53 | 40.83 | 33.96 | 39.30 |
| Hard | 52.07 | 67.57 | 47.53 | 83.58 | 51.03 | 40.97 | 34.58 | 39.26 |
| Easy+Medium | 52.45 | 68.60 | 48.07 | 83.42 | 51.30 | 41.40 | 34.86 | 39.49 |
| **Easy+Medium+Hard** | **52.74** | **68.33** | **48.68** | **83.93** | **52.36** | **40.73** | **34.31** | **40.81** |

## D.3 MODEL SCALE EXPERIMENT

To verify if GameQA's benefits extend to larger models, we conducted experiments on Qwen2.5-VL-32B using LoRA. As shown in Table 14, the 32B model demonstrates greater performance gains than the 7B model under identical LoRA settings across all domains. The average improvement on general benchmarks is +0.78 for the 32B model compared to +0.68 for the 7B model. This trend suggests that GameQA is an effective resource for enhancing reasoning in larger-scale VLMs, and we hypothesize that full-parameter fine-tuning would yield even more substantial gains.

Table 14: Performance gains from GRPO (LoRA, 5k GameQA data) on 7B vs. 32B models. Gains are shown in parentheses.

| Model | Training | General Vision Benchmarks | In-domain Games | Out-of-domain Games |
|---|---|---|---|---|
| Qwen2.5-VL-7B | Baseline | 49.94 | 25.78 | 27.09 |
| | +GameQA (LoRA) | 50.62 (+0.68) | 30.90 (+5.12) | 30.45 (+3.36) |
| Qwen2.5-VL-32B | Baseline | 60.18 | 35.91 | 35.76 |
| | +GameQA (LoRA) | **60.96 (+0.78)** | **43.24 (+7.33)** | **40.77 (+5.01)** |

## D.4 DATA SCALE EXPERIMENT

We extended our data scaling experiment up to 20k samples. While individual checkpoints exhibit fluctuations common in RL, a "binned averaging" analysis (Table 15) reveals a clear upward trend. By averaging performance across training stages, we smooth out short-term noise and observe a stable improvement trajectory. The average performance on general benchmarks steadily increases from +2.31 in the early stage to +3.02 in the late stage, confirming that model performance continues to improve when we scale up more data from GameQA.

## E APPROACH DETAILS

### E.1 TASK CATEGORY

**Target Perception Task** focuses on visual perception and basic state awareness. **State Prediction Task**, building directly on the perceptual capacities, needs predictions about state transitions.

Table 15: Binned averaging analysis of data scaling effect on Qwen2.5-VL-7B. Performance gains over baseline are shown in parentheses.

| Training Stage | Sample Range | General Vision Benchmarks | In-domain Games |
|---|---|---|---|
| Baseline Model | 0k | 49.94 | 25.78 |
| Early Stage Training | 1k–5k | 52.25 (+2.31) | 27.68 (+1.90) |
| Mid Stage Training | 6k–15k | 52.57 (+2.63) | 34.97 (+9.19) |
| Late Stage Training | 16k–20k | 52.96 (+3.02) | 38.69 (+12.91) |

**Strategy Optimization Task** then needs both perceptual and predictive capacities to find optimal solutions. This progressive structure helps organize reasoning skills from simple to complex.

The conceptual outline for each category, using Sokoban as an example, is as follows:

- **Target Perception Task:** Queries static information within the game state. For instance, questions ask about the position or number of boxes, and the answers list the specific positions of each box by directly inspecting the current state.

- **State Prediction Task:** Infers state changes following actions. For instance, questions predict the player's position after a sequence of moves. Answers are derived by analyzing the initial state, simulating the execution of each step, recording the resulting state changes, and thus determining the final player position.

- **Strategy Optimization Task:** Aims to find optimal solutions. For instance, questions ask for the shortest path to push a specific box to its designated target. The answer is derived by first analyzing the initial state to determine the optimal route for moving the box to its target, and then simulating the execution of this optimal action sequence.

### E.2    SELECTION CRITERIA OF 30 GAMES IN GAMEQA

We select these 30 games based on the following criteria.

- **Ability coverage** The games need to cover a diverse range of reasoning abilities, including 3D Spatial Perception and Understanding, Pattern Recognition and Matching, Multi-step Reasoning, and Strategic Planning.

- **Code simplicity** The code should be easy to construct, meaning they are simple enough to be programmed by an LLM, or are open-sourced.

- **Static game** They should be static or can be transformed into a static state, so that problems can be solved from a static image.

### E.3    TIME SPENT ON THE MAIN STEPS OF CODE2LOGIC FOR EACH GAME

Figure 11 illustrates the estimated time spent on implementing the main steps of the Code2Logic approach across all the 30 games in the GameQA dataset. Time cost on a game ranges from a minimum of 4 hours to a maximum of 12 hours, with an average of 7.5 hours per game. This is relatively cost-effective and appears highly acceptable, especially considering that once the data engine code is built, it can generate an unlimited number of data samples at scale.

Furthermore, we anticipate **the required manual effort and time cost will progressively decrease** due to two key factors:

- **Advancements in code agent capabilities:** We used models like GPT-4o and Claude 3.5 in our work. With the emergence of more powerful code agents (e.g., Claude Code) and general models (e.g., GPT 5.2, Claude 4.6), the synthesis difficulty and human efforts will progressively decrease.

- **Improved annotator experience and efficiency:** In our project, the data annotators were STEM undergraduates with limited experience, each handling only two games on average. By employing more experienced annotators (e.g., STEM graduates) and increasing their familiarity through repeated tasks, time cost per game can be significantly reduced.

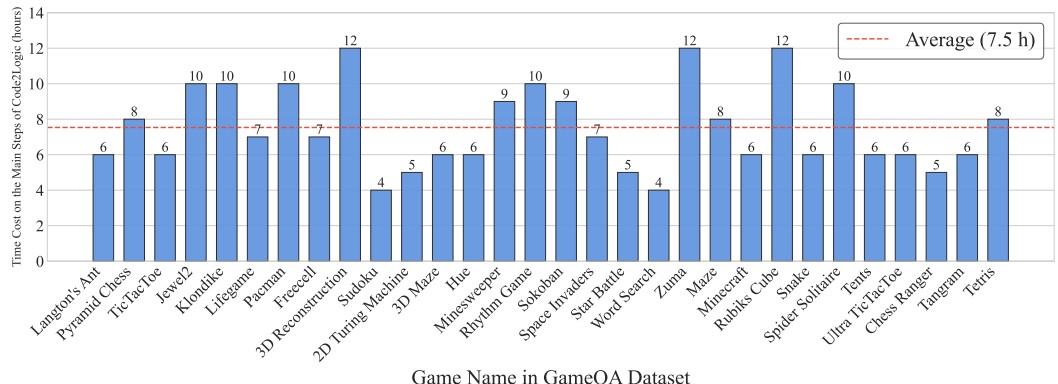

Figure 11: Estimated time (in hours) spent on implementing the main steps of Code2Logic across different games in the GameQA dataset, with an overall average of 7.5 hours per game.

### E.4 GAMES GENERATED USING EXTERNAL OPEN-SOURCE CODE

This section lists the games whose code is generated based on open-source code, as referenced in Section 2.1.

**Spider Solitaire** (Open-source code URL: `https://github.com/rdasxy/spider_solitaire`)

Based on an original Python implementation of Spider Solitaire, our code reused its core rules and GUI. And we simplified the game to a single suit (Spades) to reduce complexity, enriched initial setups through LLM-implemented random deals, and adapted the original game rules into detailed instructions.

**Klondike** (Open-source code URL: `https://github.com/milorb/klondike`)

Based on the original open-source Klondike Solitaire project built on `Pygame`[3], we adapted the code using an LLM into an automated dataset generation tool. It reused the core game engine and introduced a method for generating diverse random initializations.

**Space Invaders** (Open-source code URL: `https://github.com/leerob/space-invaders`)

Based on the original Space Invaders game built with `Pygame`, we utilized its core elements and visual assets to generate static game scenes. Using an LLM, we converted the dynamic game into static scene snapshots for dataset generation.

## F MORE INFORMATION ABOUT THE GAMEQA DATASET

### F.1 THE GAMEQA DATASET STATISTICS

The full GameQA train and test set statistics table is shown in Table 16.

### F.2 DEFINITION OF EACH CATEGORY

**3D Spatial Perception and Reasoning.** These games involve the ability to perceive, plan, and reason in 3D space to complete tasks such as navigation and spatial transformation. For example, in 3D Reconstruction, the task is to reconstruct the stacking arrangement of 3D cubes from a side view. Solving this task requires 3D spatial reasoning to establish the relationship between the 2D view and the 3D view.

---

[3]https://www.pygame.org/news

**Pattern Recognition and Matching.** These games require capability on discerning and matching visual patterns related to object shapes, colors, combinations, and other regularities. For example, in Tangram, the task is to identify which piece can fill the empty space. Solving this task requires recognizing the shape of void and matching with given pieces.

**Multi-step Reasoning.** These games feature multi-step reasoning and iteratively applying rules to reach the solution. For example, in Sudoku, the task is to infer which color should fill the empty space to ensure that no colors are repeated in the same row, column, or 3x3 grid.

**Strategic Planning.** These games require planning the optimal solution in optimization problems. For example, in Sokoban, the task is to plan the shortest path for pushing a box from the starting point to the target location.

Table 16: GameQA train and test set statistics summary. All lengths are calculated by words.

| Statistic Category | Train Set | Test Set |
|---|---:|---:|
| **Overall Counts** | | |
| Total Games | 20 | 30 |
| Total Tasks | 102 | 158 |
| Total Questions | 126,760 | 15,047 |
| **Image Statistics** | | |
| Unique Images | 74,620 | 8,620 |
| Avg. Image Width (px) | 511.00 | 504.10 |
| Avg. Image Height (px) | 475.73 | 468.98 |
| **Question Characteristics** | | |
| Avg. Question Length | 275.27 | 272.43 |
| Avg. Analysis Length | 106.85 | 144.89 |
|   - After Augmentation | 300.79 | - |
| Multiple Choice Questions | 86,520 | 10,518 |
| Avg. Choices for MCQs | 7.10 | 7.05 |
| Fill-in-the-Blank | 40,240 | 4,529 |

## F.3 DATA AUGMENTATION

To prevent model overfitting to specific reasoning patterns, we employed an LLM-based reasoning **paraphrase** strategy using InternVL2.5-78B. We only provide it with **textual information**, namely the question, original answer process and final answer, to rewrite the answer process. This approach enriches linguistic style and logical expression diversity while maintaining semantic consistency.

## F.4 DATA QUALITY ASSURANCE

To ensure the high quality and reliability of our synthetic dataset, we implemented a quality assurance process, consisting of four stages:

1. **Human Inspection:** STEM students inspected initial samples to ensure logical correctness, clarity and completeness of questions, images and reasoning steps.

2. **LLMs Check:** Fed samples to GPT-4o and Claude-3.5 Sonnet to ensure model comprehensibility, identifying necessary refinements in the samples.

3. **Post-Augmentation Verification:** Manually verified reasoning accuracy in a random subset of augmented data.

4. **Automated Data Filtering:** Removed samples based on length, high repetition (>70% 4-gram overlap), or wrong answers, reducing the total size from about 150k to 126,760.

## G DETAILS ON SOKOBAN TASK SYNTHESIS

### G.1 SOKOBAN QA TEMPLATES

Here we provide templates used for generating Sokoban puzzle samples, including the three problem types: Target Perception, State Prediction, and Strategic Optimization.

**Target perception QA template.** The Target Perception template (Table 17) is used to generate questions about identifying the current position of game elements.

Table 17: Target perception QA template

```
"question": "This is a Sokoban puzzle where Cartoon people is player
    , green X is target, brown box with X is box to push, brown
    tiles are walls, and light brown areas are movable spaces. The
    coordinates (x, y) in this puzzle represent the matrix format.
    What is the current position of the <object> (row, column)?\
    nOptions:\n[1] <option_1>\n[2] <option_2>\n[3] <option_3>\n[4] <
    option_4>\n[5] <option_5>\n[6] <option_6>\n[7] <option_7>\n[8] <
    option_8>",

"answer": <number>,

"analysis": "Player position: <pos>\nBoxes positions: <pos>\nTarget
    positions: <pos>\nThe player is currently at position <pos>.\n\
    nSo the answer is <answer>. The option number is <number>.",
```

**State prediction QA template.** The State Prediction template (Table 18) is used to generate questions about predicting the final position of the player after a sequence of moves.

Table 18: State prediction QA template

```
"question": "This is a Sokoban puzzle where Cartoon people is player
    , green X is target, brown box with X is box to push, brown
    tiles are walls, and light brown areas are movable spaces. The
    coordinates (x, y) in this puzzle represent the matrix format.
    If the player makes these moves: <mov_seq>, where will player
    end up?\n\nOptions:\n[1] <option_1>\n[2] <option_2>\n[3] <
    option_3>\n[4] <option_4>\n[5] <option_5>\n[6] <option_6>\n[7] <
    option_7>\n[8] <option_8>",
"answer": <number>,
"analysis": "Player position: <pos>\nMove 1 - <dir>: Player moves
    from <pos> to <pos>\nMove 2 - <dir>: Player moves from <pos> to
    <pos>\n...\nFinal position: <pos>\nThe option number is <number
    >",
```

**Strategic optimization QA template.** The Strategic Optimization template (Table 19) is used to generate questions about finding the optimal sequence of moves between positions.

### G.2 SUPPLEMENTARY PROMPT

As mentioned in Section 2.2. LLMs can not only refine the human-designed templates, but also design new questions and QA templates with the example prompt provided below. We will conduct a careful manual review of the quality of tasks generated by the LLM and make selections, even if we have already prompted to generate diverse and meaningful QA pairs.

Table 19: Strategic optimization QA template

```
"question": "This is a Sokoban puzzle where Cartoon people is player
    , green X is target, brown box with X is box to push, brown
    tiles are walls, and light brown areas are movable spaces. The
    coordinates (x, y) in this puzzle represent the matrix format.
    Treat the boxes as walls, What is the shortest sequence of moves
     for human to move himself from position <pos> to position <pos
    >?\n\nOptions:\n[1] <option_1>\n[2] <option_2>\n[3] <option_3>\n
    [4] <option_4>\n[5] <option_5>\n[6] <option_6>\n[7] <option_7>\n
    [8] <option_8>",

"answer": <answer_number>,

"analysis": "Player position: <pos>\nBoxes positions: <pos>\nTarget
    positions: <pos>\nStart position: <pos>\nEnd position: <pos>\
    nOptimal move sequence: <mov_seq>\nMove 1 - <dir>: Player moves
    from <pos> to <pos>\nMove 2 - <dir>: Player moves from <pos> to
    <pos>\n...\nFinal position: <pos>\n\nSo the answer is <answer>.
    The option number is <number>.",
```

---

**Prompt for designing new questions and the corresponding QA templates**

Generate Game QA Derivative Templates Based on the provided basic QA template for the Sokoban game, please design more question-answering template variations. The reference file already includes three basic template categories:

1. State Prediction - Predict the player's position after a move.

2. Target Perception - Identify the current positions of game elements.

3. Strategy Optimization - Find the optimal movement path.

Please design 3-5 innovative derivative templates for each category, ensuring the new templates:

- Maintain consistency with the original JSON format.

- Cover different reasoning difficulties and complexities.

- Test different cognitive and reasoning abilities.

When designing, please follow this reasoning hierarchy:

- Level 1: Target Perception QA - Focus on basic visual recognition and state understanding (e.g., "Where is the box?").

- Level 2: State Prediction QA - Focus on state changes and transition reasoning (e.g., "After performing these moves, where will the player be?").

- Level 3: Strategy Optimization QA - Focus on finding the optimal solution (e.g., "What is the minimum number of moves to push the box to the target?").

For each new template, please provide:

1. Template name and type classification.

2. Complete JSON template structure (including all necessary placeholders).

3. A brief description explaining the specific abilities tested by the template.

4. Placeholder filling examples (how to generate specific question instances).

Please ensure your template designs can generate diverse and meaningful Q&A pairs based on the game state and maintain consistency with the original template structure.

# H PROMPTS

## H.1 DATA AUGMENTATION

Below is the prompt used to perform the LLM-based reasoning paraphrase strategy detailed in Appendix F.3, with visual information not provided for the model.

---

**Prompt for data augmentation**

## Question: {query} *(the question generated by the data engine)*

## Ground Truth: {response} *(the answer with reasoning process generated by the data engine)*

Based on the above question and the provided ground truth, the current process of providing the answer is overly mechanical and simplistic. Please provide detailed reasoning steps based on the content of the question and the reasoning steps in the Ground Truth. The reasoning steps should be detailed, logical, and consistent with the Ground Truth.

Additionally, before starting the reasoning, emphasize: "I will carefully analyze the question and the image and provide detailed reasoning steps." Do not include statements such as "This matches the provided Ground Truth" or similar expressions in your response.

Please follow the above requirements to provide a detailed analysis, reasoning, and answer.

---

## H.2 TRAINING

---

**System prompt for the policy model trained in GRPO**

Please carefully observe the image, thoroughly understand the conditions provided in the question, use logical reasoning to arrive at the result, and reflect on and verify the reasoning process to ensure the accuracy of the answer. Finally, provide the correct answer.

---

We extract the final answer of the policy model's response using a rule-based parser (e.g., by finding phrases like "The answer is..."). The evaluator model (Qwen2.5-32B-Instruct-AWQ) solely determines if this final answer is semantically equivalent to the ground truth, with prompt shown below.

---

**Prompt for the LLM-based evaluator in GRPO**

**System prompt:**

Compare the ground truth with the prediction from AI model and determine if the prediction is correct. The question is about an image, which we have not given here. You need to determine whether the model's prediction is consistent with the ground truth. No points will be awarded for wrong answers, over answers or under answers. There are times when the answer may have a different form of expression and some variation is acceptable.

**User instruction prompt:**

## Ground Truth: The correct answer is {answer}.

*(For multiple choice question: The correct option is {answer}: {option_content}.)*

## Prediction: {final_answer}

Correctness: (Yes or No)

---

## H.3 INFERENCE

---
**Prompt for inference**

{query} Let's think step by step. Please analyze the question carefully and follow these requirements:

Provide detailed step-by-step reasoning,

Show all your work and calculations,

End your response with one of these formats:

1. For choice questions: 'The answer is [option]'

2. For other questions: 'The answer is [final answer]'

The final answer line must be on its own line at the very end of your response.

---

## H.4 EVALUATION

---
**Prompt for evaluation**

**System prompt:**

Compare the ground truth with the prediction from AI model and determine if the prediction is correct.

The question is about an image, which we have not given here. You need to determine whether the model's prediction is consistent with the ground truth. No points will be awarded for wrong answers, over answers or under answers.

The reasoning process in the prediction does not need to be considered too much, you only need to determine if the final answer is consistent. There are times when the answer may have a different form of expression and some variation is acceptable.

**User instruction prompt:**

## Question: {question}

## Ground Truth: The correct answer is {answer}.

*(For multiple-choice question: The correct option is {answer}: {option_content}.)*

## Prediction: {final_answer}

You need to determine whether the model's prediction is consistent with the ground truth. Output only:

Correctness: (Yes or No)

---

# I  CASE STUDY ON MODEL PERFORMANCE ON GAMEQA

We present GPT-4o case studies on different GameQA games below, to showcase deficiencies of GPT-4o on visual perception and reasoning. Our qualitative analysis of model performance across four game types reveals common behaviors and challenges.

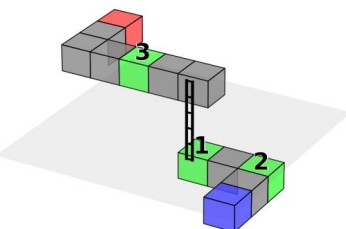

**Question:** What is the correct height relationship between the three numbered points?
**Answer:** 1 = 2 < 3
**GPT-4o Response:** Point 1 is on top of a cube that is one level above the blue starting cube. Point 2 is on the same level as the blue starting cube ... Therefore, the height relationship is: 2 < 1 < 3.
**Error Analysis:** Point 1 is at same height as point 2.

Figure 12: GPT-4o 3D Maze case study

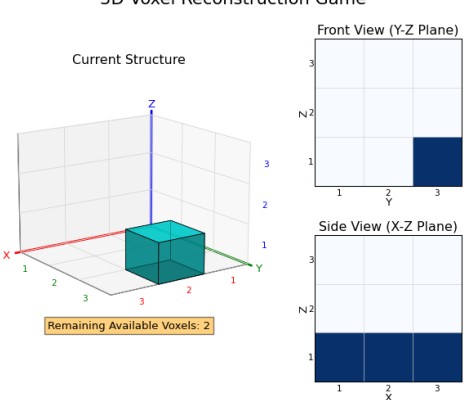

**Question**: Goal: Reconstruct a 3D structure by adding voxels to match given projections. Position (x,y,z) ranges from 1 to 3, with (1,1,1) at front-left-bottom. All voxels must be connected face-to-face. New voxels can only be placed adjacent to existing ones. Choose the correct sequence from the options below.
**Options**:
1: Add voxels at positions: [(1, 3, 1), (3, 3, 1)]
**Answer**: 1
**GPT-4o Response**: To solve this puzzle, we need to add voxels to the current structure such that the projections match the given Front View (Y-Z) and Side View (X-Z).
Let's analyze the projections:
1.Front View (Y-Z Plane):
  1. The projection shows a filled cell at (2, 1), indicating there is at least one voxel along the line of sight at y=2 and z=1.
  2. The projection shows a filled cell at (3, 1), indicating there is at least one voxel along the line of sight at y=3 and z=1.
2.Side View (X-Z Plane):
  1. The projection shows filled cells at (1, 1), (2, 1), and (3, 1), indicating there is at least one voxel along the line of sight at x=1, x=2, and x=3, all at z=1.
Given the current structure, we have a voxel at (2, 2, 1). We need to add two more voxels to match both projections …
**Error Analysis**: GPT-4o gives wrong position of the existing voxel.

Figure 13: GPT-4o 3D Reconstruction case study

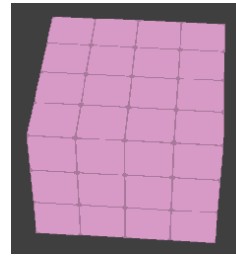

**Question**: How many cubes are there in total in the scene?
**Answer:** 48
**GPT-4o Response:** The scene shows a cube composed of smaller cubes. By counting the cubes along one edge, we can determine the dimensions of the larger cube. There are 5 cubes along each edge. Therefore, the total number of smaller cubes in the scene is calculated by cubing the number of cubes along one edge: 5 x 5 x 5 = 125. So, there are 125 cubes in total in the scene.
**Ground Truth**: Looking at the top of it, we know that the length and the width are 4 and 4 respectively. Besides, we can see that the height is 3. Therefore, the total number of cubes is 4 x 4 x 3 = 48. The answer is 48.
**Error Analysis:** GPT-4o counts number of cubes in each edge wrong.

Figure 14: GPT-4o Minecraft case study

**3D Spatial Perception and Understanding.**   Visual language models exhibit significant limitations in 3D spatial reasoning games and score lowest among the four game categories. For example, in Figure 12, GPT-4o struggled on ordering green cubes by their height (Z-coordinate), recognizing cube 1, that is closer to top of the image, as having higher Z-coordinate than cube 2. Moreover, in Figure 13, GPT-4o failed to identify the position of the only voxel in the image. In Figure 14, GPT-4o appeared to be unable to count how many cubes are on each edge of the cuboid, which is generally easy for humans.

**Pattern Recognition and Matching.** In this game category, we find that models faced difficulties in identifying patterns and locating objects. This was particularly challenging with non-grid layouts or images without row and column indicators. As seen in Figure 15, GPT-4o has poor performance in circular Zuma grid, claiming to find 3 pairs of same color marbles, but none is correct. GPT-4o also meets problems in the card games. In Figure 16, GPT-4o fails to understand "top of a pile" appears lower in the image. In Figure 17, GPT-4o misreads the rank of the card.

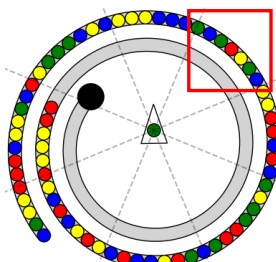

**Question:** How many marble groups of two or more same-colored marbles are there at the up-right side of the frog?
**Answer:** 0
**GPT-4o Response:** Starting from the top, the first group in the 'up-right' region is a group of two yellow marbles. Moving clockwise, the next group is a group of two blue marbles. Continuing further, there is a group of two red marbles … there are a total of 3 groups.
**Error Analysis:** None of the three groups GPT-4o claims is correct.

Figure 15: GPT-4o Zuma case study. The red rectangle is added to highlight the area referred to in the question and does not exist in original image.

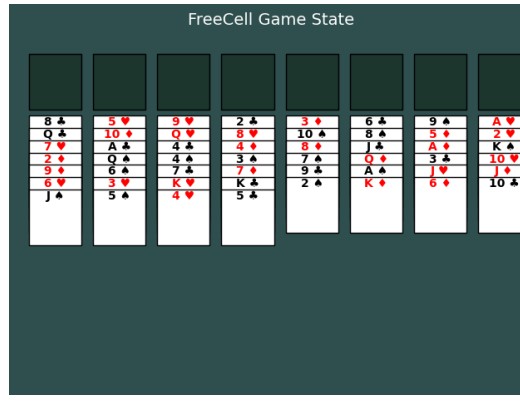

**Question:** In this FreeCell game, we have 8 cascade piles at sum, and their indexes are [0, 1, 2, 3, 4, 5, 6, 7]. In FreeCell, cards can be moved according to specific rules: A card can be moved to a free cell if available, stacked in descending order alternating colors in cascade piles, or placed in foundation piles starting from Ace. Now, find the 3rd card from the top of cascade pile 1. Options are as follows:
1.(Diamond, 5) 2.(Club, 1) 3.(Spade, 6) 4.(Heart, 2) 5.(Club, 4) 6.(Heart, K) 7.(Heart, A) 8.(Spade, 9)
**Answer:** 3
**GPT-4o Response:** To find the 3rd card from the top of cascade pile 1, we need to look at the cards in that pile. The cards in cascade pile 1, from top to bottom, are: 1. 5 of Hearts 2. 10 of Hearts 3. Ace of Clubs. Therefore, the 3rd card from the top of cascade pile 1 is the Ace of Clubs. The correct option is: 2. (Club, 1)
**Ground Truth:** The selected pile1 contains (from top to bottom) the following cards: (Spade,5), (Heart,3), (Spade,6), (Spade,Q), (Club,A), (Diamond,10), (Heart,5). The 3rd card from the top is (Spade, 6).
**Error Analysis:** GPT-4o does not understand top of a pile appears lowest in image.

Figure 16: GPT-4o Freecell case study

**Multi-step Reasoning.** Models have substantial deficiencies in multi-step reasoning. When a rule needs to be applied multiple times recursively, models are prone to repeat the initial application instead of performing subsequent reasoning steps. In Figure 19, GPT-4o repeats it finds prefixes of the target word "JUNKMAN" in options 2, 7 and 8, but actually the three options are far from the word "JUNKMAN". Another common flaw is failure to track game state changes. For example, in 2D Turing Machine game (Appendix J.3.5), models frequently forgot the symbol on a cell that had been modified in a previous step, still using the unchanged symbol to deduce the next move. A similar phenomenon was also observed in Langton's Ant game (Appendix J.3.3), where the model might not follow the new color of a certain cell that it had correctly recognized, still following the original cell color in the input image.

**Strategic Planning.** These games show that models lack certain abilities to find the best strategy. They lack the human-like insight to prune unpromising choices, and are also unable to conduct large-scale search or traversal, resulting in poor performance. In the Sokoban game case (Figure 20), for example, the number of moves needed is 27, a relatively big number. GPT-4o directly states the answer is 29, without conducting any effective analysis. Irrational mistakes that seems more unreasonable also exist. As shown in Figure 21, GPT-4o states that after the first step, body of snake includes (0, 3) and new head position is (0, 3), not realizing that this overlap means the snake has hit

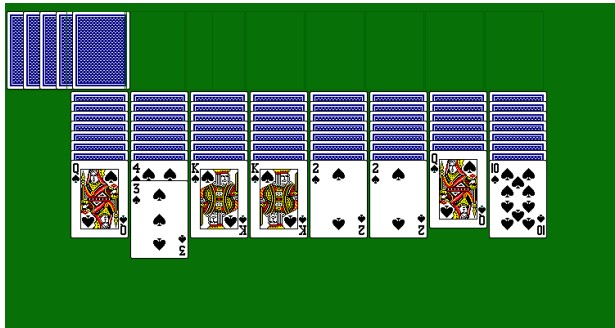

**Question**: Piles are numbered from left to right starting with `0`. Cards in a pile are numbered from on-screen top to bottom starting with `0`. Cards can be moved to another pile as long as they form a **descending sequence of the same suit**. The cards within each waste pile are also numbered starting from the bottom card. What will happen if I want to move the number 6 card of pile 6 to pile 3? **Options:** A. The move will be successful, and the cards will be in descending order, following the rules of movement. B. The move cannot be made because this card is face-down and its value is unknown. C. The move cannot be made because …
**Answer**: A

**GPT-4o Response**: In the image provided: Waste pile 6 has a visible card, which is a Jack (J) of spades. Waste pile 3 has a visible card, which is a King (K) of spades. The number 6 card in pile 6 would be the sixth card from the bottom of the pile. However, in the image, pile 6 only has one face-up card, the Jack of spades, and the rest are face-down. Therefore, the sixth card from the bottom is not visible and is face-down. Given this information, the correct option is: B. The move cannot be made because this card is face-down and its value is unknown.

**Ground Truth**: Source pile 6 is the 7th pile from the left, and destination pile 3 is the 4th pile from the left. The number 6 card in pile 6 is the 7th card, which is Queen of Spade. The top card of pile 3 is King of Spade. So moving the Queen of Spade from pile 6 to pile 3 will be successful as it forms a valid descending sequence.

**Error Analysis**: GPT-4o recognizes the visible card in pile 6 incorrectly, and misread the meaning of number 6 card.

Figure 17: GPT-4o Spider Solitaire case study

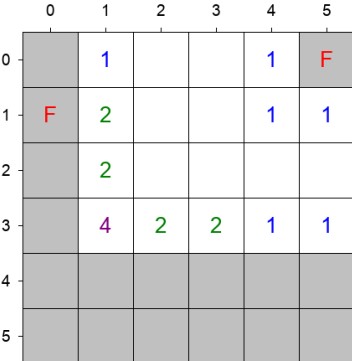

**Question:** This is a Minesweeper game. The size of the chessboard is 6x6, and there are a total of 8 mines hidden on the board. The numbers on the board indicate how many mines are adjacent to that cell, including diagonals. Cells marked with "F" (flagged) are identified as potential locations of mines based on logical deduction or prior knowledge.
**Question:** How many mines are left to be found?
**Answer**: 6

**GPT-4o Response**: To determine how many mines are left to be found, we need to analyze the board and the flagged cells. 1. **Count the flagged cells:** There are three flagged cells on the board: (0,5), (1,0), and (0,0). 2. **Total mines:** The problem states there are a total of 8 mines on the board. Total mines = 8  Flagged cells = 3   Mines left to be found = Total mines - Flagged cells = 8 - 3 = 5

**Ground Truth**: There are a total of 8 mines hidden on the board. By counting the number of F-marked cells (a total of 2), we can determine the remaining mines: 6

**Error Analysis**: GPT-4o incorrectly states (0,0) is flagged.

Figure 18: GPT-4o Minesweeper case study

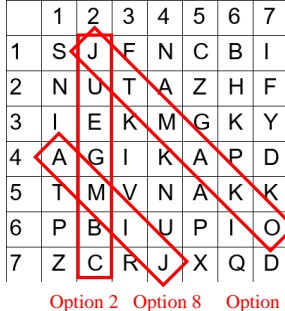

**Question**: Find the word "JUNKMAN" in the grid. Where does it start and in which direction does it go? Options: …
**Answer:** Row 7, Column 4, Direction: up
**GPT-4o Response:** … Option 2: Row 1, Column 2, Direction: down. Starting at (1, 2) = 'J', (2, 2) = 'U', (3, 2) = 'N', (4, 2) = 'K', (5, 2) = 'M', (6, 2) = 'A', (7, 2) = 'N'  - This spells 'JUNKMAN', so this option is correct.
… Option 7: Row 1, Column 2, Direction: diagonal-right-down.  Starting at (1, 2) = 'J', (2, 3) = 'U', (3, 4) = 'N', (4, 5) = 'K', (5, 6) = 'M', (6, 7) = 'A' …
Option 8: Row 7, Column 4, Direction: diagonal-left-up - Starting at (7, 4) = 'J' - Diagonal-left-up: (6, 3) = 'U', (5, 2) = 'N', (4, 1) = 'K' …
**Error Analysis:** GPT-4o is hallucinating that the three options listed have letters similar to "JUNKMAN".

Figure 19: GPT-4o Word Search case study. Red rectangles and "Option X" texts are added to highlight the options and do not exist in original image.

its head. This reveals certain deficiencies in models' fundamental reasoning and spatial imagination skills.

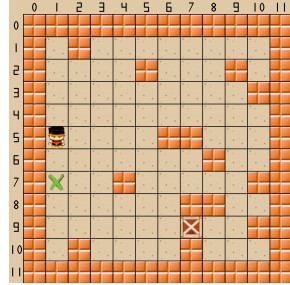

**Question:** What is the minimum number of moves needed to solve this puzzle?
**Answer:** 27
**GPT-4o Response:** To solve this Sokoban puzzle, the player needs to push the box onto the target. The player must navigate around the walls and push the box in a way that it reaches the target with the minimum number of moves.
After analyzing the puzzle, the minimum number of moves needed to solve it is 29.
**Error Analysis:** GPT-4o states an answer without analyzing.

Figure 20: GPT-4o Sokoban case study

In addition, it's worth mentioning that **a bit counter-intuitively, models actually face major difficulties in visual perception under various game scenes**, as shown in Figure 22, 18, 23 and 24, constraining their final performances.

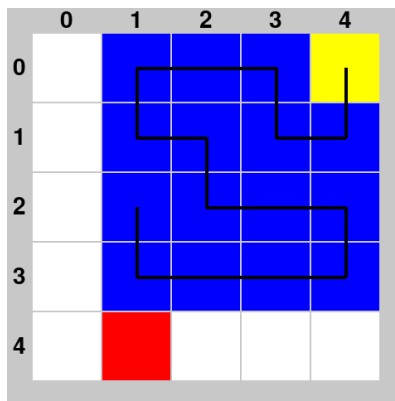

**Question**: This is a Snake game. The yellow block is the head of the snake. The blue block is the body of the snake. The red block is the food. The coordinates (x, y) in the grid represent the matrix format, where x is the row index and y is the column index. The origin (0,0) is in the the upper left of the grid. Which will happen if the snake moves like this each step: step 1: left … Options: 0: The snake hits the bound of the grid. 1: The snake hits its body. 2: The snake reaches the food. 3: Nothing happens.
**Answer:** 1
**GPT-4o Response:** To determine what happens when the snake moves according to the given steps, let's track the position of the snake's head and body through each step: Initial position: - Head: (0, 4) - Body: (0, 3), (0, 2), (0, 1), (1, 1), (2, 1), (2, 2), (2, 3), (1, 3), (1, 2) Step 1: Move left - Head moves to: (0, 3) New body: (0, 4), (0, 3) … The snake hits the bound of the grid.
**Ground Truth:** At first the snake head(yellow block) is at (0, 4). The snake body(blue blocks) is at [(1, 4) … (2, 1)]. Then it moves like this: step 1: move left Before moving, the snake is at [(0, 4) … (2, 1)]. After moving left, the snake head will move left directly, which will be at (0, 3). Each block of the snake body will move to the position of the block in front of it, so the snake body will be at [(0, 4) … (3, 1)]. Now the snake hits its body.
**Error Analysis**: GPT-4o does not correctly identify all cells of the snake body, and does not realize the snake hits its body, though it states after step 1 head moves to (0, 3) and new body includes (0, 3).

Figure 21: GPT-4o Snake case study

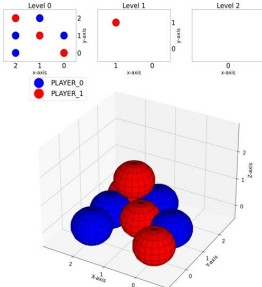

**Question**: On an 3x3 board, there are 3 levels (0 to 2). At each level k, the x and y coordinates range from 0 to 2-k, resulting in (2-k)**2 slots per level. Two players are designated as PLAYER_0 and PLAYER_1, with blue balls for PLAYER_0 and red balls for PLAYER_1 … What is the status of the ball on Level 1, which has coordinate ([1, 1])? Options: 1. PLAYER_0 2. PLAYER_1 3. Empty 4. Index out of bound
**Answer:** 2
**GPT-4o Response:** The image shows that there is no ball placed at coordinate (1, 1) on Level 1. Therefore, the correct answer is: 4. Index out of bound
**Ground Truth**: Based on level 1's grid (specifically at coordinate [1, 1]), the ball is red, which corresponds to PLAYER_1.
**Error Analysis:** (1, 1) on Level 1 is marked as red, but GPT-4o reads it as empty, and wrongly thinks that empty fits option 4.

Figure 22: GPT-4o Pyramid Chess case study

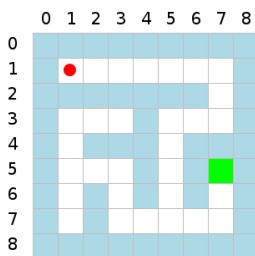

**Question**: Which directions are available to move now? Options: A. up B. left C. right D. up, down E. left, right F. up, left, right G. down, left, right H. up, down, left, right
**Answer**: C
**GPT-4o Response**: Up (0, 1): This is a blue block, so the player cannot move up. Down (2, 1): This is a white block, so the player can move down. Left (1, 0): This is a blue block, so the player cannot move left. Right (1, 2): This is a white block, so the player can move right. Therefore, the available directions for the player to move are down and right. The correct option is: D. up, down
**Ground Truth**:
The player is on (1, 1), and for cells adjacent, only (1, 2) is empty. The player can move right. Therefore, The option is C.
**Error Analysis:** GPT-4o incorrectly recognizes (2, 1) as white.

Figure 23: GPT-4o Maze case study

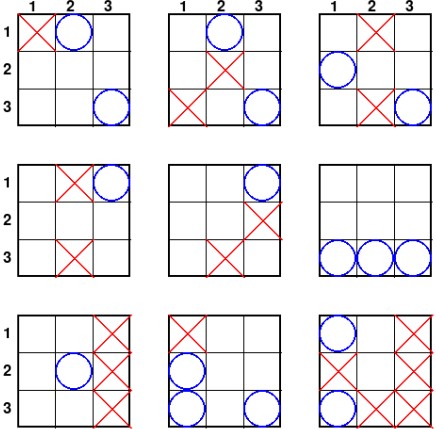

**Question**: In this game, the board is divided into 9 3*3 squares (called Nine-grids). We use (i, j, row, col) to represent the coordinates of a cell: (i, j) represents the coordinates of the Nine-grid; (row, col) represents the coordinate of the cell within the Nine-grid; i, j, row, col all range from 1 to 3. Two players take turns placing pieces on the board to mark the cells on the board, with the first player using "X" and the second player using "O" (this is the same as traditional TicTacToe). After the game starts, the first player places a chess piece in any cell in the Nine-grid in the middle (i.e., the Nine-grid (2, 2)). How many middle cells in the image are marked? Options: 1. 0 2. 1 3. 2 4. 3 5. 4 6. 5 7. 6 8. 7 9. 8 10. 9
**Answer:** 3
**GPT-4o Response:**
Top-left grid: (2,2) is unmarked. Top-middle grid: (2,2) is unmarked. Top-right grid: (2,2) is **X** (marked). Middle-left grid: (2,2) is **X** (marked). Center grid: (2,2) is unmarked. Middle-right grid: (2,2) is unmarked. Bottom-left grid: (2,2) is **O** (marked). Bottom-middle grid: (2,2) is unmarked. Bottom-right grid: (2,2) is **X** (marked). Correct answer: 5. 4
**Ground Truth**: By analyzing the blocks in the picture, we can see that the middle cells of nine-grid (1, 2) (Top-middle) and (3, 1) (Bottom-left) are marked, while the rest are empty. So, there are 2 middle cells marked, which means the answer is 3.
**Error Analysis:** GPT-4o misjudges the middle cells of Top-middle, Top-right, Middle-left, Bottom-right, and Bottom-right.

Figure 24: GPT-4o Ultra Tic-Tac-Toe case study

## J DETAILS ON THE 30 GAMES AND MORE EXAMPLE DATA SAMPLES IN THE GAMEQA DATASET

**A total of 10 games are introduced below in detail, with detailed question information and specific example QAs provided, while the others 20 games in brief.**

Typically, each game provides example images for three Plot Levels (Easy, Medium, Hard) representing different image complexities, along with their grading criteria.

For demonstration purposes here, the images have been uniformly scaled. Refer to the dataset repository for the actual relative sizes and resolutions of the images. The average height and width of the images in our dataset are presented in Section 3.

For the 10 games introduced in detail:

1. The specific questions uses these images as visual input.

2. Labels such as "E1", "M2", and "H1" are used to denote specific images. For example, "Q1 (E1)" indicates that the corresponding image for this Q1 question sample is the "E1" image (i.e., the first image of the "Easy" Plot Level).

3. For each game, its *Introduction* text is a common component prepended to the beginning of every associated question.

4. Due to space limitations, we reasonably simplify the *Introduction* for some games, and we also omit some information in some of the analyses. However, most analyses remain detailed and clearly demonstrate the reasoning processes.

### J.1 3D SPATIAL PERCEPTION AND UNDERSTANDING

#### J.1.1 3D RECONSTRUCTION

The game takes place in a 3x3x3 three-dimensional space with randomly initialized small cubes (voxels). Players reference two target side views (projections) and continue placing voxels in the 3D space to make the structure match these views (not considered in some tasks), with a maximum limit on placed voxels, all of which must be connected (not placed in midair). Question types include counting voxels in the current structure, identifying if given coordinates contain a voxel, checking if the current structure matches the side views, predicting side views after voxel additions, selecting the addition sequence that results in the structure matching side views, and calculating the minimum voxels needed to be added from the current structure to meet the two side views. The difficulty (Plot Level) is primarily determined by the number of voxels in the target three-dimensional structure.

**Images and Plot Level division**

| Easy | Medium | Hard |
|------|--------|------|
| Final voxel count $\in [3, 5]$ | Final voxel count $\in [6, 10]$ | Final voxel count $\in [11, 15]$ |

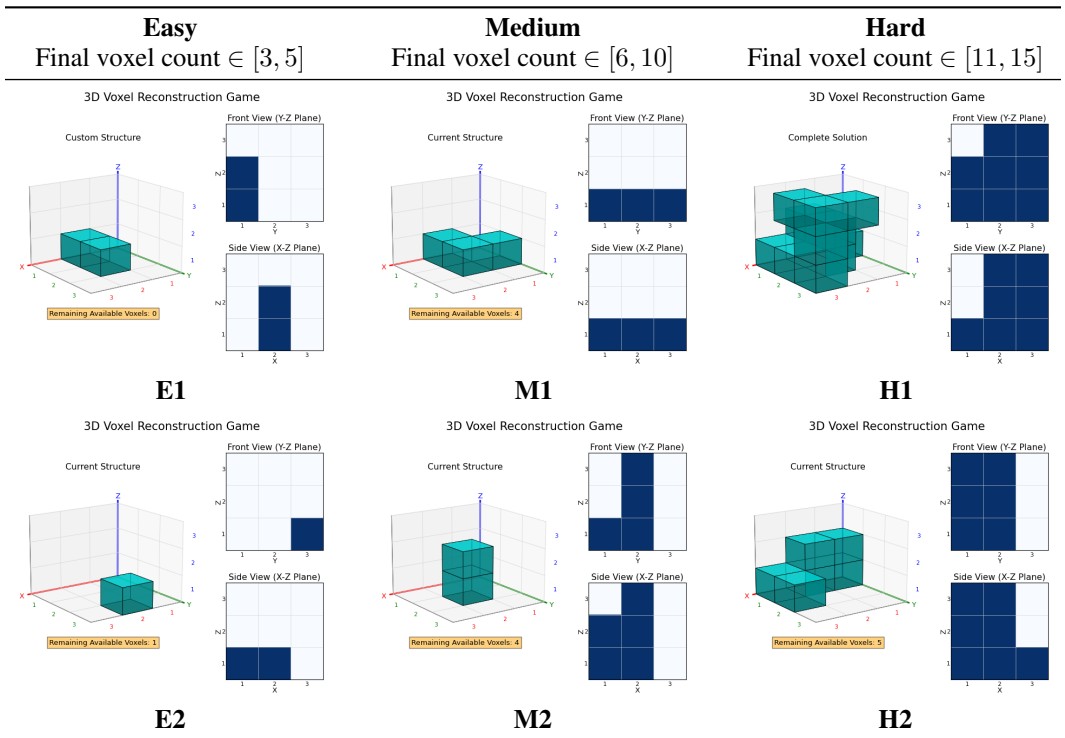

## Question information

|    | QA type | QA Level | Description |
|----|---------|----------|-------------|
| Q1 | Target Perception | Easy | Count voxels in the 3D structure. |
| Q2 | Target Perception | Easy | From options, select the position containing a voxel. |
| Q3 | Target Perception | Medium | Select the option describing how the structures projections match target projections. |
| Q4 | State Prediction | Medium | Predict projections after adding specified voxels. |
| Q5 | State Prediction | Hard | Choose the correct voxel addition sequence to match target projection(s), adhering to game rules. |
| Q5 | Strategy Optimization | Hard | Calculate the minimum additional voxels required to match both target projections. |

## Specific questions and analysis

*Introduction:* The current structure has some initial voxels, and your goal is to complete it. Game Rules: 1. Goal: Reconstruct a 3D structure by adding voxels to match given projections.

2. Grid Space: The game is played on a 3x3x3 cube grid.

3. Coordinates: Position (x,y,z) ranges from 1 to 3, with (1,1,1) at front-left-bottom.

4. Position Rule: Each position can contain at most one voxel.

5. Connectivity: All voxels must be connected face-to-face.

6. Voxel Limit: You have a maximum of n additional voxels available.

7. Placement Rule: New voxels can only be placed adjacent to existing ones.

8. Front View (Y-Z): Shows structure when viewed along the negative X-axis direction (front to back), with Y as horizontal axis and Z as vertical axis. Projection coordinates are in (y,z) format.

9. Side View (X-Z): Shows structure when viewed along the positive Y-axis direction (left to right), with X as horizontal axis and Z as vertical axis. Projection coordinates are in (x,z) format.

**Q1 (E1):** How many voxels are there in the given structure?

**Analysis:** The structure contains voxels at the following positions: (2,1,1), (2,2,1). By counting these positions, we can see there are 2 voxels in total. Therefore the answer is 2.

---

**Q2 (M1):** Which of the following positions contains a voxel? Choose the correct position from the options below.

Options: 1: (3,2,2); 2: (3,2,1); 3: (2,3,1); 4: (2,2,3); 5: (2,1,1); 6: (1,3,3)

**Analysis:** Let's analyze each option:

Option 1 - Position (3,2,2): This position is empty.

... *(omitted)*

Option 5 - Position (2,1,1): This position contains a voxel. This is the correct answer.

Option 6 - Position (1,3,3): This position is empty.

Therefore, the correct answer is option 5.

---

**Q3 (H1):** How does the voxel structure's projections match with the target projections?

Choose the correct description from the options below.

Options:

1: Neither Y-Z projection nor X-Z projection matches the target;

2: Only Y-Z projection matches the target; 3: Only X-Z projection matches the target;

4: Both Y-Z and X-Z projections match the target

**Analysis:** Let's analyze the projections:

1. Looking along the negative X-axis direction (Front View, using (y,z) coordinates): - We can see voxels at positions [(2, 1, 2), ... *(omitted)*, (3, 3, 3)], forming a Y-Z projection of [(1, 1), ... *(omitted)*, (3, 3)] - This matches the target Y-Z projection exactly.

2. Looking along the positive Y-axis direction (Side View, using (x,z) coordinates): - We can see voxels at positions [(1, 1, 1), ... *(omitted)*, (3, 3, 2)], forming a X-Z projection of [(1, 1), ... *(omitted)*, (3, 3)] - This matches the target X-Z projection exactly.

Based on the above analysis, both projections match the target. Therefore, the correct answer is option 4.

---

**Q4 (E2):** Action: Add 1 voxels at positions: [(2, 2, 1)]

Question: After adding these voxels, what will be the X-Z projection of the new structure?

Answer Format:

1. Write the answer as a list of three lists: [[row1], [row2], [row3]] 2. Each row should contain three numbers (0 or 1) 3. Rows are ordered from top to bottom of the projection 4. Numbers in each row are ordered from left to right 5. Use 1 to indicate presence of a voxel in the projection, 0 for empty space 6. Example format: [[0, 1, 0], [1, 1, 0], [0, 1, 1]]

**Analysis:** Let's analyze the projection:

Looking along the positive Y-axis direction (Side View, using (x,z) coordinates):

- We can see voxels at positions [(2, 2, 1)], which in X-Z projection appear at positions [(2, 1)]. Therefore, the answer is: [[0, 0, 0], [0, 0, 0], [0, 1, 0]]

**Q5 (M2):** Which sequence of voxel additions will make the structure match the both target projections?
Choose the correct sequence from the options below.
Options:
1: Add voxels at positions: [(1, 1, 1), (1, 1, 2), (1, 2, 1), (3, 2, 1)]; ... *(omitted)*
6: Add voxels at positions: [(1, 2, 1), (1, 2, 2), (2, 1, 1), (2, 2, 3)]; ... *(omitted)*
8: Add voxels at positions: [(2, 1, 2), (2, 3, 1), (3, 3, 3)]

**Analysis:** Let's analyze each option:
Current structure: [(2, 2, 1), (2, 2, 2)]
Option 1: - The added voxels maintain connectivity - Does not match both target projections - Uses 4 voxels, which is within the limit of 4
... *(omitted)*
Option 6: - The added voxels maintain connectivity - Matches both target projections - Uses 4 voxels, which is within the limit of 4
... *(omitted)*
Option 8: - The added voxels are not all connected to the existing structure - Does not match both target projections - Uses 3 voxels, which is within the limit of 4
Therefore, the correct answer is option 6.

**Q6 (H2):** What is the minimum number of voxels needed to add to the current structure to make it match both target projections?

**Analysis:** Let's solve this optimization problem through systematic reasoning:
1. Basic Information: - Current structure: 6 voxels at positions [(1, 1, 1), (1, 1, 2), (2, 1, 1), (2, 1, 2), (3, 1, 1), (3, 2, 1)] - Remaining available voxels: 3
2. Analysis of Y-Z Projection (Front View):
a) Current Y-Z projection: [0, 0, 0] (top) [1, 0, 0] (middle) [1, 1, 0] (bottom)
b) Target Y-Z projection: [1, 1, 0] (top) [1, 1, 0] (middle) [1, 1, 0] (bottom)
c) Candidate positions from Y-Z view: (?, 1, 3), (?, 2, 2), (?, 2, 3) where ? can be any value from 1 to 3 for x-coordinate
d) Note: At positions where projection already shows 1, we can add more voxels without affecting the projection. For example, if (2, y0, z0) exists (where y0 and z0 are specific fixed values), we can add (1, y0, z0) or (3, y0, z0) at the same projection position.
3. Analysis of X-Z Projection (Side View): ... *(omitted)*
4. Finding Required Positions:
By matching candidates from both projections:
- When (?, y, z) from Y-Z view matches (x, ?, z) from X-Z view, position (x, y, z) can be filled.
- Example: if we have (?, 2, 3) and (1, ?, 3), then (1, 2, 3) is required
- To ensure connectivity, we can add voxels at positions where projections already show 1
* This strategy is optimal because it doesn't create new projections
* Use these positions as 'bridges' to connect required positions Required positions from projection matching: [(1, 1, 3), (2, 2, 2), (2, 2, 3)]
5. Connectivity Analysis and Completion: ... *(omitted)*
6. Verifying Optimality: ... *(omitted)*
Therefore, the minimum number of voxels needed to complete the reconstruction is 3.

### J.1.2 3D MAZE

This game involves pathfinding within a three-dimensional maze constructed from unit cubes arranged in a 3D grid space (voxel-based). Traversal is subject to specific rules: horizontal movement (along X or Y axes) is permitted between adjacent cubes only if they reside at the same height (Z-coordinate). Vertical movement (ascending/descending along the Z-axis) is permitted between vertically aligned cubes (sharing X and Y coordinates) only if a ladder explicitly connects them. Key locations are color-coded: a blue cube designates the starting position, and a red cube marks the goal destination. Additionally, green cubes, often labeled with numbers, serve as waypoints, decision junctions, or specific points of interest referenced in the questions. Question types assess spatial navigation and path analysis: (1) determining the correct direction of travel required at each green waypoint to follow a path towards the destination; (2) ordering a set of specified green cubes

based on their height (Z-coordinate); (3) identifying the sequence of green cubes visited along the shortest path from the start to the end; (4) reporting the exact sequence in which green cubes are encountered when traversing from start to end following a defined path. Path generation often utilizes concatenation of randomized 'atomic' path segments (e.g., move +2X, move +2Y, move +2/3Z) to create a primary route, with branching paths potentially added similarly to introduce choices, aiming to minimize visual occlusion between path segments.

**Images and Plot Level division**

| Easy | Medium |
|---|---|
| Simple, a single path | Complex, side road exists |

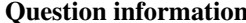

| E1 | M1 |
| E2 | M2 |

**Question information**

| | QA type | QA Level | Description |
|---|---|---|---|
| Q1 | Target Perception | Easy | Height Comparison |
| Q2 | State Prediction | Medium | Sequence Finding |
| Q3 | State Prediction | Medium | Main Path |
| Q4 | State Prediction | Hard | Path Finding |

**Specific questions and analysis**

*Introduction:* Rules: 1. Player can only walk on top of cubes

2. Player can climb ladders if they can reach the cube under the ladder

3. From a ladder, player can reach the top of the last cube with the ladder

4. Blue cube is start position, red cube is goal position

5. Green cubes are numbered points (1, 2, and 3)

**Q1 (E1):** What is the correct height relationship between the three numbered points? Use '<' for 'lower than' and '=' for 'same height as'.
Options:
1: 2 = 3 < 1 2: 1 < 3 < 2 3: 3 < 1 < 2 4: 1 < 2 = 3
5: 3 < 2 < 1 6: 2 < 1 = 3 7: 1 = 2 = 3 8: 3 < 1 = 2

**Analysis:** Analyzing the heights of each point:
Comparing points 1 and 2: Found a path from 1 to 2:
* Move left-forward * Move left-forward
- Point 2 is same height as point 1
Comparing points 1 and 3: Found a path from 3 to 1:
* Go up 3 blocks * Go up 3 blocks
- Point 3 is lower than point 1
Comparing points 2 and 3: Found a path from 3 to 2:
* Go up 3 blocks * Go up 3 blocks * Move left-forward * Move left-forward
- Point 3 is lower than point 2
Therefore, the correct height relationship is 3 < 1 = 2, making the answer Option 8.

**Q2 (E2):** What is the correct sequence of numbered checkpoints when following the path from start to goal?
Options:
1: Start -> 2 -> 3 -> 1 -> 4 -> Goal; 2: Start -> 2 -> 3 -> 4 -> 1 -> Goal;
3: Start -> 4 -> 3 -> 1 -> 2 -> Goal; 4: Start -> 4 -> 2 -> 3 -> 1 -> Goal;
5: Start -> 3 -> 2 -> 4 -> 1 -> Goal; 6: Start -> 2 -> 4 -> 3 -> 1 -> Goal

**Analysis:** Following the path from start to goal:
- Step 1: Move right-forward - Step 2: At checkpoint 2 - Step 3: Move up
- Step 4: Move left-forward - Step 5: At checkpoint 3 - Step 6: Move left-forward
- Step 7: Move up - Step 8: At checkpoint 4 - Step 9: Move right-forward
- Step 10: At checkpoint 1 - Step 11: Move right-forward
Therefore, the correct sequence is Start -> 2 -> 3 -> 4 -> 1 -> Goal, making the answer Option 2.

**Q3 (M1):** Which numbered blocks are passed through when following the most direct path from start to goal?
Options:
1: 1, 2; 2: 2, 3; 3: 3; 4: 2; 5: 1; 6: None; 7: 1, 2, 3; 8: 1, 3

**Analysis:** Following the main path from start to goal:
- Step 1: Move up - Step 3: Move up - Step 4: Move right-forward
- Step 5: Move left-forward - Step 6: Move right-forward - Step 7: Move right-forward
Blocks not on main path: 1, 2. Therefore, the blocks passed through on the main path are: 3, making the answer Option 3.

**Q4 (M2):** Which combination of path choices leads to the goal?
Options:
1: 1-right-forward, 2-right-forward, 3-up;
2: 1-left-forward, 2-right-forward, 3-left-forward;
3: 1-left-forward, 2-up, 3-left-forward; 4: 1-left-forward, 2-up, 3-up;
5: 1-left-forward, 2-right-forward, 3-up; 6: 1-right-forward, 2-up, 3-up;
7: 1-right-forward, 2-right-forward, 3-left-forward;
8: 1-right-forward, 2-up, 3-left-forward

**Analysis:** From the start point, you first meet branch 1, then branch 2, then branch 3, before finally reaching the goal.
Analyzing each branch point:
- At branch 1, going right-forward leads to branch 2, while going left-forward leads to a dead end
- At branch 2, going up leads to branch 3, while going right-forward leads to a dead end
- At branch 3, going up leads toward the goal, while going left-forward leads to a dead end
Therefore, the correct sequence is 1-right-forward, 2-up, 3-up, that is 1-right-forward, 2-up, 3-up, making the answer Option 6.

### J.1.3  RUBIK'S CUBE

This game is based on the classic Rubik's Cube puzzle. The game interface presents both 3D views and an unfolded view of the cube. The 3D views display the cube from two different angles: left-tilted 30 degrees looking down, and right-tilted 30 degrees looking up. The cube features six faces with distinct colors (yellow, white, orange, red, blue, and green), and players can manipulate the cube according to standard rotation rules (where F, B, L, R, U, D represent Front, Back, Left, Right, Upper, and Down faces, with a prime symbol denoting counterclockwise rotation).

Question types, assessing spatial reasoning and pattern recognition, include identifying the color at a specific position on a face, counting a color's occurrences on a face, and predicting a position's color after a move sequence. Further questions ask for the minimum moves to solve a single face or the entire cube. The difficulty level (Plot Level) is determined by the number of random moves used to scramble the cube: 1 move for Easy, 2 moves for Medium, and 3 moves for Hard.

**Images and Plot Level division**

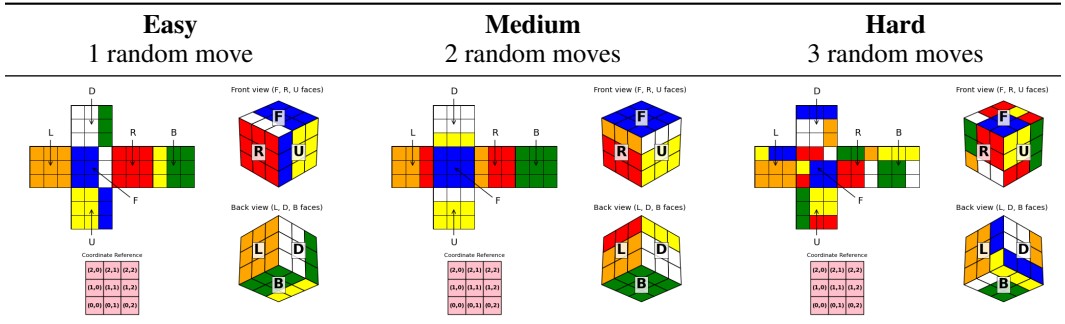

### J.1.4  PYRAMID CHESS

This is a 3D two-player competitive game. Players take turns placing balls on a board, building a pyramid structure layer by layer. The player whose ball occupies the pyramid's top wins.

Question types challenge players to assess the board by determining which player's ball occupies a given position, the specific state of any board position, and the total count of balls. Additionally, questions involve predicting the result of a player placing a ball, calculating the minimum number of moves required to place a ball at a certain position, and identifying a player's optimal placement in the current state. Plot Level is determined by the board's base size, with larger base dimensions increasing the challenge.

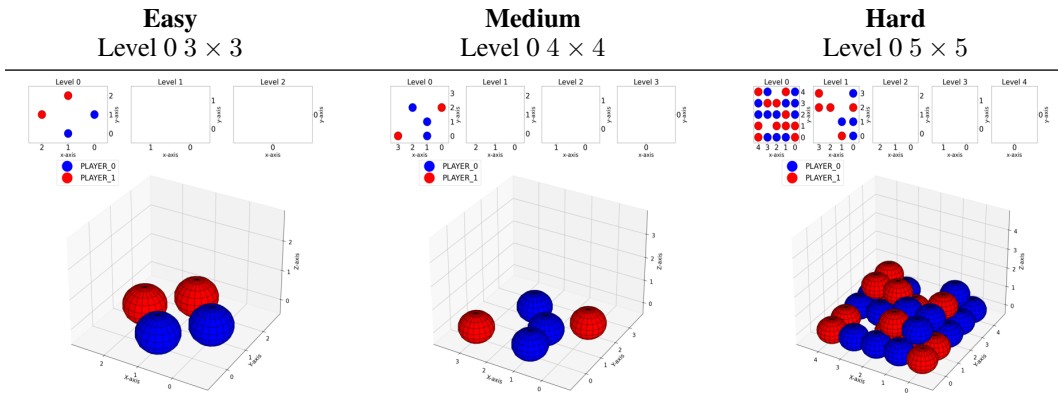

### J.1.5  MINECRAFT

This Minecraft QA generator is designed to produce a series of questions that test 3D perception and understanding within a simulated "Minecraft" environment. Given the open-ended nature of Minecraft, the tasks are custom-designed to probe specific cognitive abilities. The generated questions aim to evaluate how well an agent can interpret and reason about 3D scenes.

The question set begins by assessing precise 3D perception. Q1 requires recognizing various sceneries present in the image, such as different ores, TNT, pumpkins, or environmental features like rivers and lava. Q2 tests the ability to accurately count the total number of blocks in a given structure. These foundational perceptual skills are prerequisites for the subsequent three tasks, which demand reasoning based on both visual input and provided rules. These more complex questions involve planning: determining the minimum blocks to cross a river (Q3), calculating the blocks needed to reach a target block at a certain height, possibly using ladders (Q4), and a combined scenario requiring both river crossing and climbing to access a target block, again considering ladders (Q5). Plot Level is determined by the number of sceneries (Q1), the cuboid size (Q2), the width of the river (Q3, Q5) and the height of the target block (Q4, Q5).

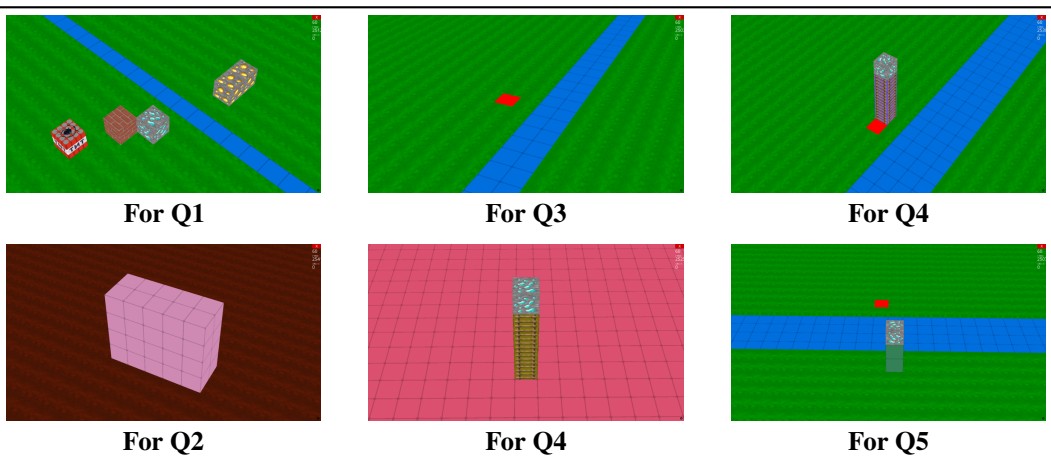

## J.2 Pattern Recognition and Matching

### J.2.1 Color Hue

This game involves reasoning about color gradients within a grid structure. Certain rows and/or columns within the grid display smooth color transitions. Cells that are intentionally left blank or empty are visually marked with a gray crosshatch pattern. Color information may be conveyed using standard color names (e.g., "purple"), derived programmatically from their Hue-Saturation-Value (HSV) properties.

Question types focus on understanding and interpolating these color gradients: (1) identifying the specific color present at a given row and column index; (2) determining the starting and ending colors of a specified gradient row or column; (3) selecting the correct color from a provided set of options (e.g., six color patches) that should logically fill a designated blank cell (marked with a letter) based on the surrounding gradient(s). The complexity ('plot level') scales with the dimensions of the grid.

**Images and Plot Level division**

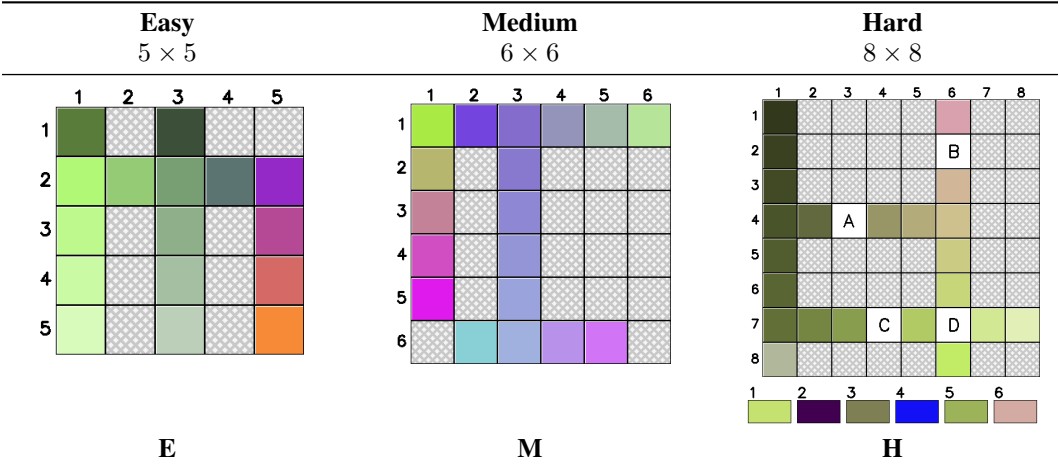

| | Easy $5 \times 5$ | Medium $6 \times 6$ | Hard $8 \times 8$ |
|---|---|---|---|
| | E | M | H |

## Question information

| | QA type | QA Level | Description |
|---|---|---|---|
| Q1 | Target Perception | Easy | Color Description |
| Q2 | Target Perception | Medium | Gradient Pattern |
| Q3 | State Prediction | Hard | Color Matching |

## Specific questions and analysis

*Introduction:* Rules:

1. Each numbered region represents a piece on the board.

2. Pieces are considered adjacent if they share at least one edge.

3. Pieces that only touch at corners are not considered adjacent.

4. Some pieces have been removed and are shown below the main board.

**Q1 (M):** What color is the cell at row 1, column 6?
Options:
1: green; 2: white; 3: vivid red; 4: pale bright green; 5: bright cyan; 6: cyan; 7: bright orange; 8: dark green.

**Analysis:** The cell at position (1, 6) is pale bright green. So the answer is Option 4.

**Q2 (E):** What is the gradient pattern in column 5?
Options:
1: transitioning from bright purple to pale dark cyan;
2: transitioning from vivid green to pale bright purple;
3: transitioning from pale bright yellow to vivid dark blue;
4: transitioning from vivid bright blue to pale red;
5: transitioning from yellow to light gray;
6: transitioning from red to pale bright cyan;
7: transitioning from purple to bright red;
8: transitioning from black to pale bright cyan.

**Analysis:** The column 5 shows a pattern that is transitioning from purple to bright red. So the answer is Option 7.

**Q3 (H):** Which color should be put in cell B?
Options: Colors are numbered from 1 to 6 in the palette below.

**Analysis:** We need to find the correct color for cell B at position (2, 6). Let's analyze the color patterns around this cell:
Looking vertically, we see a pattern transitioning from pale bright red to bright yellow. Let's look at our color options:
Option 1 is bright yellow; Option 2 is vivid dark purple; Option 3 is pale yellow; Option 4 is vivid bright indigo; Option 5 is yellow; Option 6 is pale bright red. Based on the pattern, we should use pale bright red (Option 6).

### J.2.2 TANGRAM

This game presents a spatial reasoning puzzle inspired by Tangram, involving the manipulation and fitting of polygonal shapes within a grid. The grid is partitioned into several distinct regions or "pieces", each identified by a unique integer ID. Cells belonging to a specific piece display that piece's ID number; cells not part of any displayed piece are left blank, representing empty space. One or more pieces are removed from the main board. Questions test pattern recognition and spatial matching skills across various dimensions: identifying piece area and adjacency, determining correct rotations to fit removed pieces back into empty spaces, and strategically positioning multiple pieces to fill available gaps. The puzzle complexity scales with the grid size.

**Images and Plot Level division**

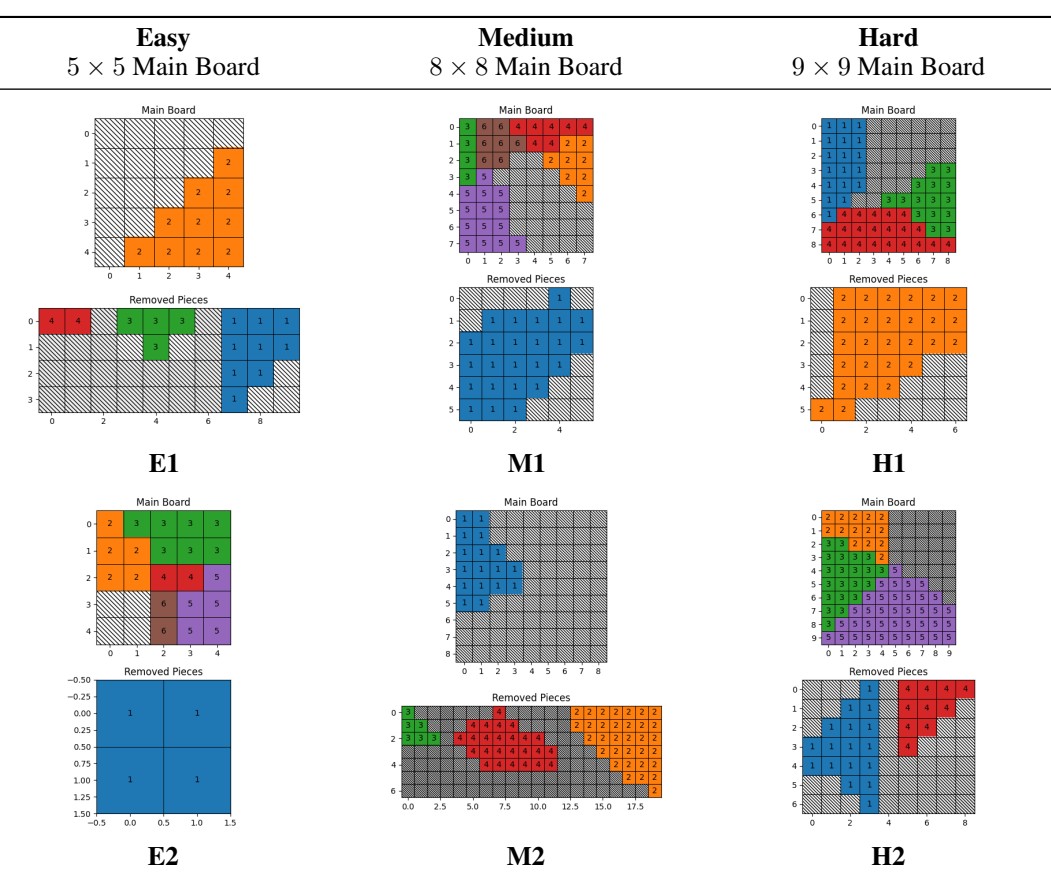

| **Easy** | **Medium** | **Hard** |
| 5 × 5 Main Board | 8 × 8 Main Board | 9 × 9 Main Board |

E1     M1     H1

E2     M2     H2

**Question information**

| | QA type | QA Level | Description |
|---|---|---|---|
| Q1 | Target Perception | Easy | Main board piece |
| Q2 | State Prediction | Medium | Removed piece rotation feasibility |
| Q3 | Target Perception | Medium | Target piece area calculation |
| Q4 | Target Perception | Medium | Adjacent piece type count |
| Q5 | State Prediction | Hard | Piece Placement |

**Specific questions and analysis**

*Introduction:* Rules:

1. Each numbered region represents a piece on the board.

2. Pieces are considered adjacent if they share at least one edge.

3. Pieces that only touch at corners are not considered adjacent.

4. Some pieces have been removed and are shown below the main board.

**Q1 (E1):** How many pieces are currently on the main board?
Options: 1: 4 2: 1 3: 0 4: 3 5: 2 6: 5 7: 7 8: 6
**Analysis:** Let's analyze the puzzle state:
Pieces currently on the board: Piece 2 (vivid bright orange) around position (3, 3)
Removed pieces: Piece 4; Piece 3; Piece 1
By counting the unique non-zero numbers on the main board, we can see there are 1 pieces remaining. Therefore, the answer is Option 2.

**Q2 (M1):** Can the removed piece fit back into the main board by only rotation? If yes, what rotation(s) would work?
Options: 1: rotate 0 degrees 2: no matter what degrees rotated, it always can fit 3: both rotate 0 and 180 degrees 4: rotate 90 degrees clockwise 5: can't put inside (flipped) 6: rotate 180 degrees 7: rotate 90 degrees by both direction 8: rotate 90 degrees counterclockwise
**Analysis:** Let's analyze how piece 1 can be rotated to fit the hole:
1. Dimension Analysis: - Hole dimensions: 6x6 - Piece dimensions: 6x6
- Based on dimensions, these rotations (clockwise) might work: 0, 90, 180, 270
2. Testing Each Rotation:
0ř rotation: - Failed: First mismatch at row 0, column 1 (mapped to (0,1) of removed piece and (2,3) of board) - At this position, the hole was empty but the piece was absent
*... (omitted: 90ř and 180ř rotation)*
270ř rotation: - Success! Piece fits perfectly
3. Summary: - Valid rotations found: 270 - Some rotations work
So, the correct answer is: rotate 90 degrees counterclockwise which is Option 8.

**Q3 (M2):** Question: What is the area (number of cells) of Piece 1?
Options: 1: 11 2: 12 3: 14 4: 17 5: 18 6: 19 7: 20 8: 21
**Analysis:** Let's analyze Piece 1 (vivid blue) row by row:
The piece spans from row 0 to 5 (height of 6):
Row 0: 2 cells from column 0 to 1; ... *(omitted)*
Adding up all the cells: 2 + 2 + 3 + 4 + 4 + 2 = 17 cells. Therefore, the answer is Option 4.

**Q4 (H1):** Question: How many different pieces are adjacent to Piece 3?
Options: 1: 5 2: 6 3: 0 4: 2 5: 3 6: 4 7: 7 8: 1
**Analysis:** Let's analyze Piece 3 (green):
Piece Boundaries: - Spans rows 3 to 7 (height: 5) - Spans columns 4 to 8 (width: 5)
1. Cell-by-cell examination: Cell (3,7): No adjacent pieces; ... *(omitted)* Cell (5,4): -
down: Piece 4 (vivid bright red) at (6,4) ... *(omitted)* Cell (7,8): - down: Piece 4 (vivid
bright red) at (8,8)
2. Adjacent Pieces Summary: - Piece 4 (vivid bright red): 7 contact sides
Total number of unique adjacent pieces: 1. Therefore, the answer is Option 8.

**Q5 (H2):** Question: At which position should Piece 1 be placed? Each option shows
(top_row,left_col) to (bottom_row,right_col).
Options: 1: (0,3) to (6,6) 2: (0,6) to (6,9) 3: (0,4) to (6,7) 4: (0,5) to (6,8)
**Analysis:** Let's analyze the placement of Piece 1 and Piece 4:
1. Hole dimensions: 7x5 2. Piece 1 dimensions: 7x4 3. Piece 4 dimensions: 4x4
We know that if Piece 1 fits, then it must be placed at one of the four corners.
Testing each corner: - upper-left: Attempting to place Piece 1 at (0,5) to (6,8) Failed:
Cell (4,0) on Removed Pieces plot maps to board position (4,5) which isn't empty -
upper-right: Attempting to place Piece 1 at (0,6) to (6,9) Success! Remaining hole
dimensions: 4x4 Then placing Piece 4 at (0,5) to (3,8) Both pieces fit perfectly! -
bottom-left: Since Piece 1 has the same height as the hole, bottom-left corner is same as
upper-left corner. Skipped. ... *(omitted: bottom-right, same as upper-right corner)*
Therefore, Piece 1 should be placed at position (0,6) to (6,9) as shown in Option 2.

### J.2.3 FREECELL

This scene presents a solitaire card game whose goal is to move all cards to foundation piles, fol-
lowing specific rules. This game is played with a standard deck of 52 cards, arranged in n tableau
columns, four open cells, and four foundation piles. Cards can be moved between tableau columns
according to descending order and alternating colors, while empty tableau spaces can only be filled
by kings. The four open cells act as temporary storage, allowing players to temporarily hold cards
for strategic moves. Complexity is controlled by adjusting the number of tableau columns, testing
the model's ability to search for a efficient operation list to complete the game.

| **Easy** | **Medium** | **Hard** |
|---|---|---|
| 8 cascade piles | 6 cascade piles | 4 cascade piles |

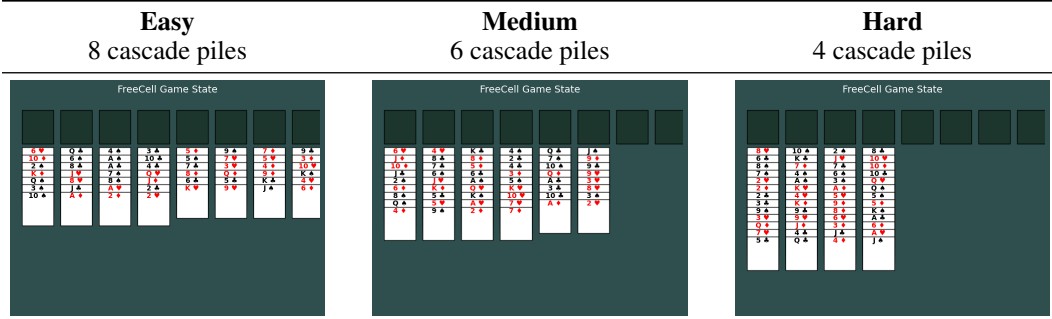

### J.2.4 TETRIS

This Tetris-derived game maintains the original objectives while simplifying visuals to highlight core
information. Players arrange falling blocks to eliminate rows by: moving/rotating pieces during
descent until they land at the bottom or on other blocks, clearing complete horizontal rows. The
game ends when blocks reach the grid's top. The simplified interface shows a white grid with gray
squares representing placed blocks and red squares indicating the current falling piece (with grid
coordinates). While actual games use color-coding for different block batches, this visual distinction
is omitted as irrelevant to gameplay logic. Advanced Tetris variants are excluded.

Questions cover: 1) Empty squares in a specified row 2) Identifying the current red block's shape 3)
Timestamps until the falling block lands after given moves 4) Maximum eliminable rows from the
current block's optimal placement.

| **Easy** | **Medium** | **Hard** |
|:---:|:---:|:---:|
| $8 \times 8$ | $12 \times 12$ | $16 \times 16$ |

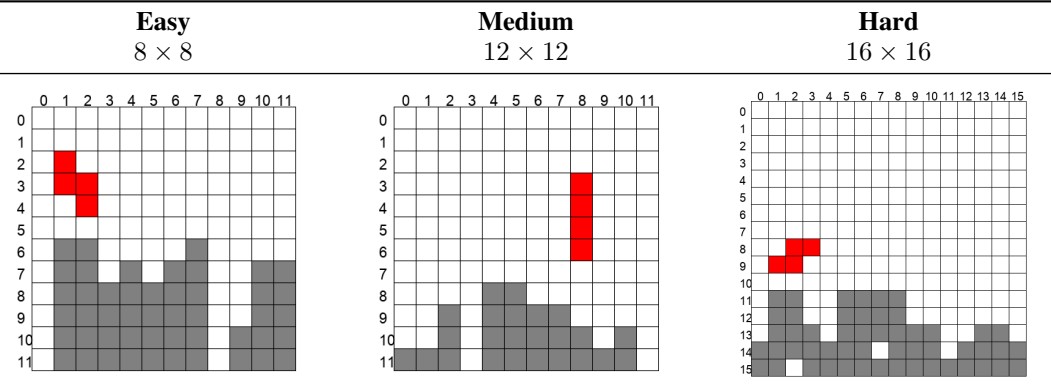

### J.2.5 ZUMA

This game is a classic marble-shooting puzzle game where players control a frog that shoots colored marbles toward a chain of rolling marbles on a track. The objective is to clear all marbles before they reach the black hole at the end. Players must create groups of three or more same-colored marbles, which will disappear from the track. The frog's marbles travel in a straight line until they hit marbles already in their path.

The game tests spatial reasoning, color recognition, and strategic planning through various question types: identifying the color of the next marble to be shot, counting marbles of specific colors, determining the number of same-colored marble groups in certain directions, predicting which marble will be hit at specific angles, analyzing the outcome of shots, and evaluating optimal elimination strategies. Plot difficulty levels are determined by track length and marble count.

| **Easy** | **Medium** | **Hard** |
|:---:|:---:|:---:|
| Short track with a few marbles | Medium-length track with more marbles | Long track with many marbles |

### J.2.6 SPIDER SOLITAIRE

The game is based on Microsoft's classic Spider Solitaire, with the original four suits simplified to just one suit. The objective of the game is to move all 13 cards of the same suit, arranged in descending order from King to Ace, from the waste piles to the foundation piles. The cards in the waste piles must be arranged in descending order. The foundation piles serve as the final destination for complete sequences. The game screen includes several waste piles, a stock pile, and foundation piles, with each pile containing several stacked cards. Some cards are face down, indicating that their rank is unknown, while others are face up, revealing their rank. The dataset includes tasks such as identifying the card on top of a pile, moving cards from the waste piles, and determining the optimal move. The dataset is divided into three difficulty levels based on the number of waste piles.

| Easy | Medium | Hard |
|------|--------|------|
| 8 waste piles | 9 waste piles | 10 waste piles |

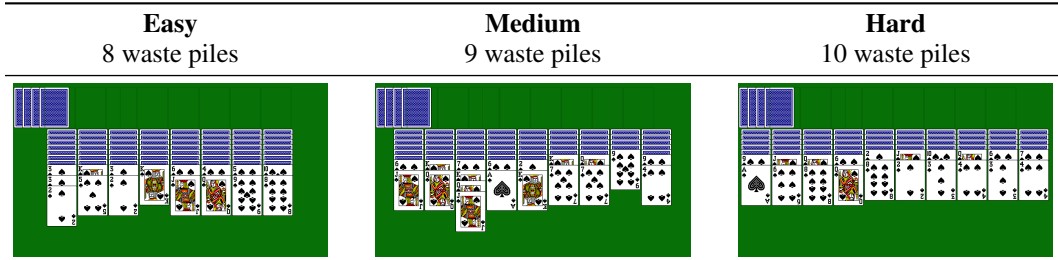

### J.2.7   JEWEL2

Jewel2 is a grid-based strategic puzzle game. It is inspired by Microsoft's classic game Bejeweled 2, with certain modifications made to the original game. The game board is square-shaped and consists of five basic elements and seven special elements. The basic elements are five gems of the same shape but different colors, while the special elements are seven gems with different shapes from the basic ones, designed to test the model's pattern recognition ability. The main objective of the game is to eliminate elements by forming horizontal or vertical lines of three or more identical items. Successfully eliminating elements increases your score and clears space for new elements to appear. The game tasks include recognizing elements on the board, executing elimination operations, and maximizing the score. Plot Level is determined by the size of the board

| Easy | Medium | Hard |
|------|--------|------|
| $4 \times 4$ | $5 \times 5$ | $6 \times 6$ |

| Jewel2 Game | Jewel2 Game | Jewel2 Game |
|-------------|-------------|-------------|
| Total Cleared: 44 | Total Cleared: 95 | Total Cleared: 1 |

### J.2.8   KLONDIKE

This Klondike Solitaire-based strategy game challenges players to analyze card layouts and apply rules for optimal decisions. It uses a standard interface with Stock, Waste, Foundation, and Tableau piles. The goal is to move all 52 cards, by suit and in ascending order (Ace to King), to the four Foundation Piles. Key mechanics include building Tableau piles down in alternating colors and descending order, building Foundations up, and strategically moving cards to reveal face-down ones, utilize the Waste Pile, and advance cards to Foundations.

Questions, generated from the current card layout, cover diverse Klondike decision-making and analysis scenarios. Types include identifying valid moves, determining the most effective move strategy (e.g., one that reveals a card or helps build Foundations), and analyzing for deadlocks. Players must apply logical reasoning based on on-screen card information and Klondike rules to select correct answers. Difficulty is dynamically set by the number of face-up cards.

| Easy
#face-up cards $\leq 19$ | Medium
#face-up cards$\in [20, 23]$ | Hard
#face-up cards $\geq 24$ |
|---|---|---|

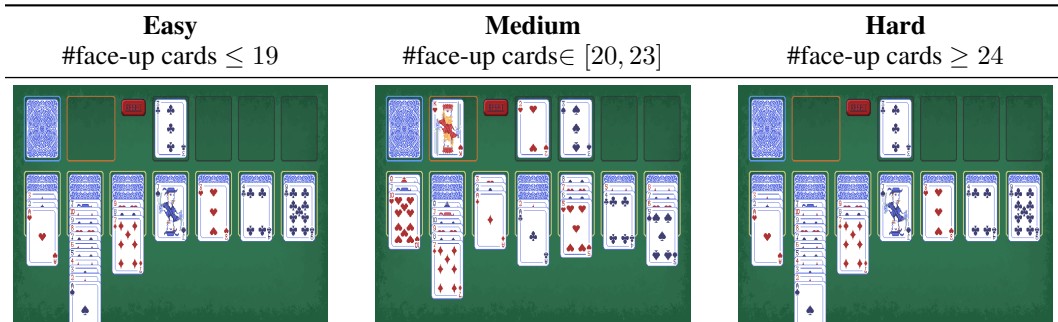

## J.3 MULTI-STEP REASONING

### J.3.1 STAR BATTLE

This scene presents a 2D n Œ n matrix which are divided into n regions. Each region has a specified color and is connective. The goal is to place stars in the matrix to make sure each row, col, region has only one star and the stars must not be adjacent to each other on rows, columns and diagonals. Complexity is controlled by adjusting the matrix size, testing the model's ability to reason according to the known rules.

**Images and Plot Level division**

| Easy
$5 \times 5$ (5 colors) | Medium
$6 \times 6$ (6 colors) | Hard
$8 \times 8$ (8 colors) |
|---|---|---|

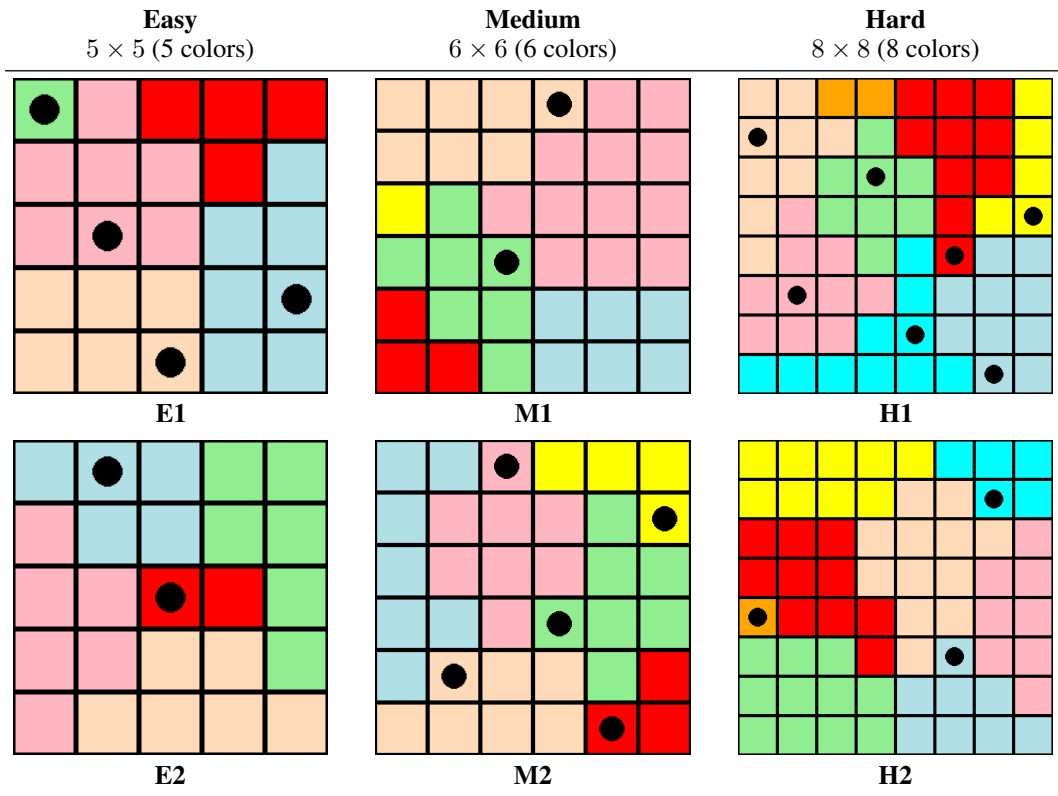

**Question information**

| | QA type | QA Level | Description |
|---|---|---|---|
| Q1 | Target Perception | Easy | Identify the cell belonging to the given region |
| Q2 | Target Perception | Easy | Identify the cell belonging to the given region and containing a star |
| Q3 | State Prediction | Medium | Identify the cell where a star can be placed |
| Q4 | State Prediction | Hard | Find the position of the final star needed to complete the puzzle |

**Specific questions and analysis**

*Introduction:*

We have a 5*5 grid. The grid is divided into 5 regions. Cells with the same color belong to the same region.

Colors: Region0 (light pink), Region1 (powder blue), Region2 (light green), Region3 (peach), Region4 (red), Region5 (yellow), Region6 (cyan), Region7 (orange).

In the image, a star is represented by a black dot. If a cell has been placed a star, a black dot will be shown on this cell. We should place the star in this Star Battle Puzzle according to the following rules:

Each row must contain exactly 1 star(s). Each column must contain 1 star(s). Each region must contain exactly 1 star(s). Stars cannot be adjacent to each other, including diagonally.

The cells in the grid are labeled with row and column numbers starting from 0. The top-left corner of the grid is (0, 0). (x,y) means a cell at row x and column y. Now we have placed some stars in the grid.

**Q1 (E2):** The region with index 1 is represented by the color powder blue in the grid. Given the current state, which cell in the following options belong to region 1?
Options:
1. (2, 3); 2. (0, 2); 3. (3, 4); 4. (3, 1);
5. (2, 2); 6. (3, 2); 7. (2, 1); 8. (1, 0)

**Analysis:** The region with index 1 is represented by the color powder blue in the grid. In this puzzle, we need to identify which cell in the following options belongs to this region. The region 1 contains the following cells: (0, 0), (0, 1), (0, 2), (1, 1), (1, 2). So (0, 2) belongs to region 1. The answer is Option 2.

**Q2 (E2):** In the current puzzle state, region 1 is associated with color powder blue. Please identify which of the following cells in this region that contains a star?
Options:
1. (1, 1); 2. (2, 0); 3. (0, 1); 4. (0, 0);
5. (0, 2); 6. (4, 4); 7. (1, 2); 8. (1, 4)

**Analysis:** In this task, we need to find all the stars in the region with index 1. The region with index 1 corresponds to the color powder blue. This region contains the following cells: (0, 0), (0, 1), (0, 2), (1, 1), (1, 2). Note that a star is represented by a black dot. Now scan the cells of the region 1 on the image. The cell with a black dot is: (0, 1). So the answer is Option 3.

**Q3 (M1):** Now we have placed some stars in the grid. Based on the current puzzle state, which of the following cells can a star be placed in?
Options:
1. (4, 2); 2. (5, 3); 3. (1, 1); 4. (2, 2);
5. (1, 0); 6. (3, 4); 7. (3, 3); 8. (4, 0)

**Analysis:** Cell (3, 3) cannot hold a star because: It is adjacent to a star, so it cannot hold a star. Cell (4, 2) cannot hold a star because: It is adjacent to a star, so it cannot hold a star. Besides, this cell is in region 2, which already contains one star, so it cannot hold a star. Cell (1, 0) cannot hold a star because: It is not adjacent to any star. However, this cell is in region 3, which already contains one star, so it cannot hold a star. Cell (2, 2) cannot hold a star because: It is adjacent to a star, so it cannot hold a star. Cell (5, 3) cannot hold a star because: It is not adjacent to any star. This cell is in region 1, which contains no stars. However, Column 3 has already been placed a star. Therefore, it cannot hold a star. Cell (1, 1) cannot hold a star because: It is not adjacent to any star. However, this cell is in region 3, which already contains one star, so it cannot hold a star. Cell (3, 4) cannot hold a star because: It is not adjacent to any star. This cell is in region 0, which contains no stars. However, Row 3 has already been placed a star. Therefore, it cannot hold a star. Cell (4, 0) can hold a star because: It is not adjacent to any star. This cell is in region 4, which contains no stars. Both row 4 and column 0 now have no stars. Thus, the correct answer is Option 8.

**Q4 (H1):** Now the puzzle has only one star left to be placed. The left star should be placed in which cell?

**Analysis:** **Step-by-step reasoning to solve the puzzle:**
1. **Preplaced stars and their positions:**
- The following stars are already placed: Row 1, Column 0, Row 2, Column 3, Row 3, Column 7, Row 4, Column 5, Row 5, Column 1, Row 6, Column 4, Row 7, Column 6.
- These positions fulfill the requirement of placing one star per row, column, and region.
2. **Identify rows and columns with and without stars:**
- **Rows with stars:** Row 1, Row 2, Row 3, Row 4, Row 5, Row 6, Row 7.
- **Rows without stars:** Row 0.
- **Columns with stars:** Column 0, Column 1, Column 3, Column 4, Column 5, Column 6, Column 7.
- **Columns without stars:** Column 2.
3. **Determine remaining valid cell:**
- The final star must be placed in a row and column that are both missing stars.
- Based on the information above, the row without a star is Row 0 and the column without a star is Column 2.
- The only available intersection is cell (0, 2), which satisfies the row and column constraints.
4. **Region check:**
- The preplaced stars occupy the following regions: 0, 1, 2, 3, 4, 5, 6.
- The remaining region that requires a star is: Region 7.
5. **Final validation:**
- The cell (0, 2) belongs to the remaining region without a star. - Placing the star here satisfies all row, column, region, and adjacency constraints.
Thus, the final star must be placed at **Row 0, Column 2**.

### J.3.2 SUDOKU

Sudoku is a puzzle that requires filling a grid such that each row, column, and subgrid contains all digits from 1 to 9 without repetition.Our Sudoku-like puzzle game can be adapted to serve as a multi-modal dataset by replacing the numbers 1-9 with nine different colors. In this game, players are provided with a grid, where each row, column, and subgrid must contain all nine colors without repetition.

The types of questions in the game are as follows: 1.The color of a specific cell. 2.The number of cells of a certain color on the board.3.The number of rows, columns, or blocks with more blank cells than a specified number.4.The number of possible color options for a specific cell under the current

board conditions.5.The color for a third cell after two other cells are filled with specific colors. The difficulty level of the game is determined by the size of the grid and the number of filled cells.

**Images and Plot Level division**

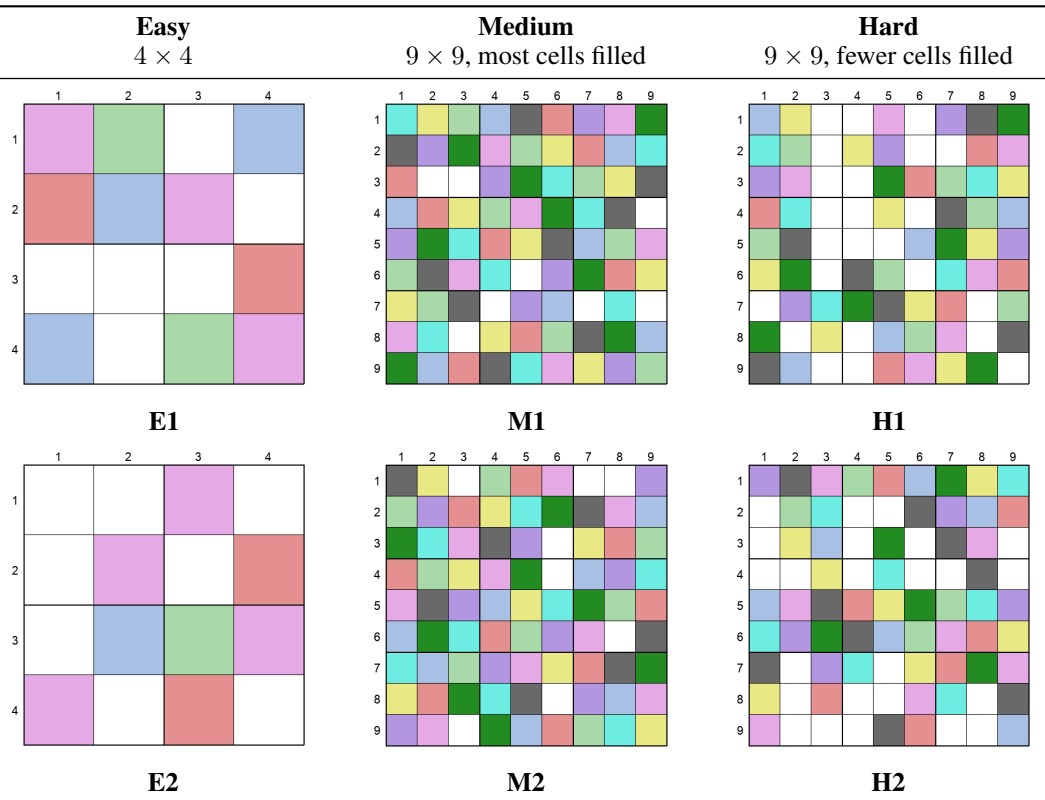

| Easy | Medium | Hard |
|------|--------|------|
| $4 \times 4$ | $9 \times 9$, most cells filled | $9 \times 9$, fewer cells filled |
| E1 | M1 | H1 |
| E2 | M2 | H2 |

**Question information**

| | QA type | QA Level | Description |
|---|---------|----------|-------------|
| Q1 | Target Perception | Easy | Position color identification |
| Q2 | Target Perception | Easy | Color occurrence count |
| Q3 | Target Perception | Medium | Sparse unit count |
| Q4 | State Prediction | Medium | Valid color candidated inference |
| Q5 | State Prediction | Hard | Guided-position color deduction |

**Specific questions and analysis**

**Q1 (E1):** What color is at position (2,1) (note that on the board the position (2,1) has already been filled with a certain color)? Choose from the following options: A.red, B.green, C.blue, D.magenta

**Analysis:** From the image, we can see the color at Position (2,1) is red.
So the answer is A.

**Q2 (M1):** How many times does aqua appear on the board?

**Analysis:** Color aqua appears at: (1,1), (2,9), (3,6), (4,7), (5,3), (6,4), (7,8), (8,2), (9,5), total 9 times.
So the answer is 9.

**Q3 (E2):** How many columns have more than 1 empty cell?

**Analysis:** Col analysis:

col 1 has 3 empty cells in positions 1, 2, 3;

col 2 has 2 empty cells in positions 1, 4;

col 3 has 1 empty cells in positions 2;

col 4 has 2 empty cells in positions 1, 4.

In total, 3 col(s) have more than 1 empty cell.

So the answer is 3.

---

**Q4 (H1):** How many colors can be filled in position (7,1)? Infer based on the current situation focusing only on the colour of the position.

**Analysis:** Constraint anlysis for position (7,1):

Existing colors in row: purple, aqua, forest green, gray, yellow, red, green

Existing colors in column: blue, aqua, purple, red, green, yellow, forest green, purple

Existing colors in box: purple, aqua, green, yellow, gray, blue

Therefore, possible colors are: magenta. So the answer is 1.

---

**Q5 (H2):** After determining colors at positions (2,1), (2,5), what color should be at position (2,4)? Choose from following options: A.red, B.green, C.blue, D.magenta, E.yellow, F.aqua, G.gray, H.purple, I.forest green

**Analysis:** Deductive reasoning process:

Step 1: Position (2,1): Existing colors in the row: green, aqua, gray, purple, blue, red. Existing colors in the column: purple, blue, aqua, gray, yellow, magenta. Existing colors in the 3x3 box: purple, gray, magenta, green, aqua, yellow, blue

Therefore, the only possible color for this position is forest green.

Step 2: ... Therefore, the only possible color for this position is magenta.

Final analysis for position (2,4): ... After previous deductions, possible color reduced to: yellow

So the answer is E.

### J.3.3  LANGTON'S ANT

This game simulates the behavior of Langton's Ant in a cellular automaton. The ant is represented by a red arrow indicating its initial position and direction. It moves on a randomly generated grid composed of black and white squares, following a fixed set of rules: If the ant is on a black square, it turns 90 degrees to the right, flips the square to white, and moves forward one step; if the ant is on a white square, it turns 90 degrees to the left, flips the square to black, and moves forward one step.

There are three types of questions in the game: 1. Identify the ant's initial position and direction. 2. Predict the ant's position and direction after a given number of steps. 3. Given a specific square, infer how many times its color has changed after the ant has moved a certain number of steps.

The difficulty of the game is determined by the question type and the size of the grid: the three question types increase in complexity respectively, and the grid size defines the level of difficultyn = 5 indicates an easy level, while n = 13 indicates a hard level.

**Images and Plot Level division**

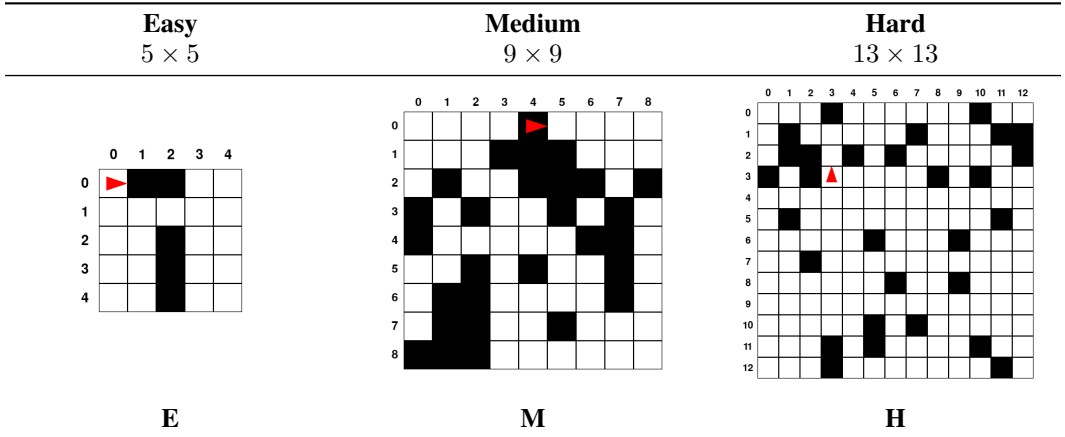

| | Easy | Medium | Hard |
|---|---|---|---|
| | 5 × 5 | 9 × 9 | 13 × 13 |
| | E | M | H |

## Question information

| | QA type | QA Level | Description |
|---|---|---|---|
| Q1 | Target Perception | Easy | Identify the current position and direction of the ant. |
| Q2 | State Prediction | Medium | Predict the ant's position and direction after several steps. |
| Q3 | State Prediction | Hard | Count how many times a specific cell changes its color. |

## Specific questions and analysis

*Introduction:*

In Langton's Ant, we have a grid where each cell is either white or black. A red arrow represents an ant, showing its current position and direction. The ant follows these simple rules:

1. If the ant is on a white cell, it turns right 90 degrees, changes the cell to black, and moves forward one step

2. If the ant is on a black cell, it turns left 90 degrees, changes the cell to white, and moves forward one step

3. If the ant would move off the grid, it wraps around to the opposite side (using modulo with grid size)

**Q1 (E):**   What is the current position and direction of the ant in the image?
Answer using one of the following options with its corresponding letter:
A: Position (1, 3), facing up; B: Position (0, 4), facing left
C: Position (2, 3), facing down; D: Position (4, 3), facing up
E: Position (4, 3), facing down; F: Position (0, 0), facing up
G: Position (0, 0), facing right; H: Position (4, 3), facing right

**Analysis:**   Step-by-step analysis:
1. Look at the red arrow in the image which represents the ant. 2. The arrow's position indicates the ant is at coordinates (0, 0). 3. The arrow's direction shows the ant is facing right.
Therefore, the ant's current position is (0, 0) and it's facing right. The answer is G.

---

**Q2 (E):**   After 6 steps, what will be the ant's position and direction?
Answer using one of the following options with its corresponding letter:
A: Position (2, 0), facing down; B: Position (2, 3), facing right
C: Position (0, 0), facing right; D: Position (4, 1), facing right
E: Position (4, 4), facing up; F: Position (0, 3), facing down
G: Position (2, 1), facing left; H: Position (4, 4), facing left

**Analysis:**   Initial state: The ant is at (0, 0) facing right.
Let's follow the ant's movement step by step:
- Step 1: Ant is on a white cell at (0, 0), facing right. It turns right, changes the cell to black, moves forward to (1, 0), now facing down.
- Step 2: Ant is on a white cell at (1, 0), facing down. It turns right, changes the cell to black, moves forward to (1, 4), now facing left.
- Step 3: Ant is on a white cell at (1, 4), facing left. It turns right, changes the cell to black, moves forward to (0, 4), now facing up.
- Step 4: Ant is on a white cell at (0, 4), facing up. It turns right, changes the cell to black, moves forward to (0, 0), now facing right.
- Step 5: Ant is on a black cell at (0, 0), facing right. It turns left, changes the cell to white, moves forward to (4, 0), now facing up.
- Step 6: Ant is on a white cell at (4, 0), facing up. It turns right, changes the cell to black, moves forward to (4, 1), now facing right.
Final state: The ant is at (4, 1) facing right. The answer is D.

---

**Q3 (E):**   After 12 steps, how many times did the cell at position (0, 0) change its color? Answer with a number.

**Analysis:**   Initial state: The ant is at (0, 0) facing right. Target cell (0, 0) starts as white.
Let's follow the ant's movement step by step:
- Step 1: Ant is on a white cell at (0, 0), facing right. It turns right, changes the cell to black, moves forward to (1, 0), now facing down. Target cell (0, 0) is visited and changes from white to black (change #1).
- Step 2: Ant is on a white cell at (1, 0), facing down. It turns right, changes the cell to black, moves forward to (1, 4), now facing left. Target cell (0, 0) remains black.
- Step 3: Ant is on a white cell at (1, 4), facing left. It turns right, changes the cell to black, moves forward to (0, 4), now facing up. ...
Step 4: Ant is on a white cell at (0, 4), facing up. It turns right, changes the cell to black, moves forward to (0, 0), now facing right. ...
Step 5: Ant is on a black cell at (0, 0), facing right. It turns left, changes the cell to white, moves forward to (4, 0), now facing up. Target cell (0, 0) changes from black to white (change #2).
- Step 6: Ant is on a white cell at (4, 0), facing up. It turns right, changes the cell to black, moves forward to (4, 1), now facing right. ...
- Step 7: Ant is on a white cell at (4, 1), facing right. It turns right, changes the cell to black, moves forward to (0, 1), now facing down. ...
... (Omitted: Step 8-11. Ant continues moving, flipping cells, but (0, 0) remains white.)
- Step 12: Ant is on a white cell at (0, 1), facing down. It turns right, changes the cell to black, moves forward to (0, 0), now facing left. Target cell (0, 0) remains white.
Final state: The ant is at (0, 0) facing left. Target cell (0, 0) changed color 2 times. The answer is 2.

### J.3.4 WORD SEARCH

This game is a visual search task based on the classic Word Search puzzle paradigm. It features a grid where each cell contains a single letter. Target words are embedded within this grid, oriented horizontally, vertically, or diagonally (spanning eight possible directions).

Question types assess visual parsing and pattern recognition within the grid, including: (1) identifying the letter located at a specific row and column index; (2) counting the total occurrences of a given letter across the entire grid; (3) determining the direction (out of eight possibilities) in which a specified word extends, given its starting cell coordinates; and (4) locating both the starting cell coordinates and the correct direction for a given target word within the grid. The complexity ('plot level') is influenced by the grid size.

| **Easy** $5 \times 5$ | | | | | | **Medium** $7 \times 7$ | | | | | | | | **Hard** $8 \times 8$ | | | | | | | | |
|---|---|---|---|---|---|---|---|---|---|---|---|---|---|---|---|---|---|---|---|---|---|---|
| | 1 | 2 | 3 | 4 | 5 | | 1 | 2 | 3 | 4 | 5 | 6 | 7 | | 1 | 2 | 3 | 4 | 5 | 6 | 7 | 8 |
| 1 | B | M | H | J | X | 1 | T | O | Z | X | D | T | N | 1 | D | T | O | D | C | J | C | C |
| 2 | Z | C | C | P | H | 2 | B | R | U | I | J | P | Z | 2 | U | L | V | G | H | G | A | G |
| 3 | A | I | X | T | N | 3 | Y | K | S | G | Z | Q | O | 3 | Z | A | F | U | A | J | M | I |
| 4 | F | J | G | O | O | 4 | O | Z | B | X | B | A | S | 4 | P | K | X | K | L | R | W | T |
| 5 | W | O | Q | M | P | 5 | X | X | M | V | T | R | S | 5 | O | E | P | V | V | C | D | K |
| | | | | | | 6 | R | L | W | P | Q | R | P | 6 | G | P | G | Y | A | M | C | J |
| | | | | | | 7 | S | C | D | W | C | X | T | 7 | F | N | S | E | H | J | J | Z |
| | | | | | | | | | | | | | | 8 | T | R | Y | O | S | K | C | X |

### J.3.5 2D TURING MACHINE

This game presents a simulation of a two-dimensional Turing machine. The state of the machine's tape, represented as a grid, is visualized using distinct colors for different symbols within each cell. The initial position of the read/write head is indicated visually, typically by a black dot, and is also specified in the accompanying text description. The core task requires simulating the step-by-step operation of the defined Turing machine. Question types focus on tracking the machine's execution, including: (1) determining the head's coordinates after a specified number of steps; (2) identifying the symbol (color) under the head after a specified number of steps; (3) describing the sequence of symbol changes within a particular cell over a given number of steps; and (4) identifying the step number at which the machine first enters a specific state. The complexity ("Plot Level") of the task is primarily determined by the dimensions of the grid.

| **Easy** $3 \times 3$ | **Medium** $4 \times 4$ | **Hard** $5 \times 5$ |
|---|---|---|

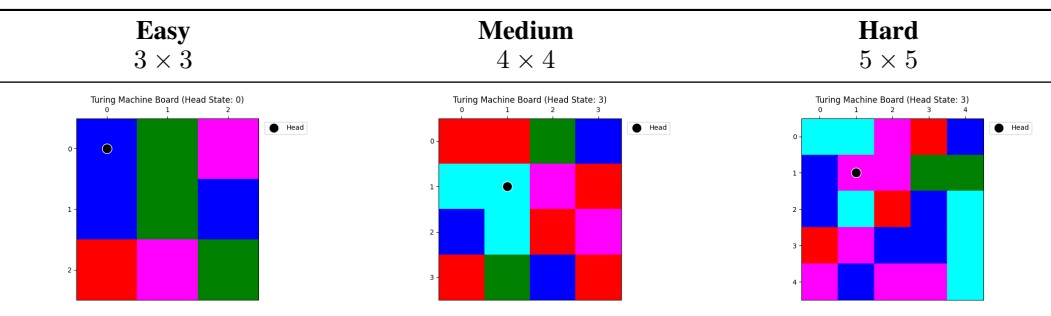

### J.3.6 TENTS

Tents is a logic puzzle played on a grid with predefined tree positions and row/column tent counts. The objective is to place tents adjacent to trees while following these rules: each cell holds either a tree, a tent, or remains empty; the number of tents matches the number of trees; every tent must be horizontally or vertically adjacent to at least one tree; no two tents can be adjacent in any direction (including diagonally); and row/column tent totals must match the given numbers.

Questions involve analyzing partially filled grids, such as determining the current number of tents in a row, remaining tents to place, identifying tree locations among given positions, available spots for new tents without immediate rule violations, and selecting rule-compliant tent placements. Puzzle difficulty scales with grid size.

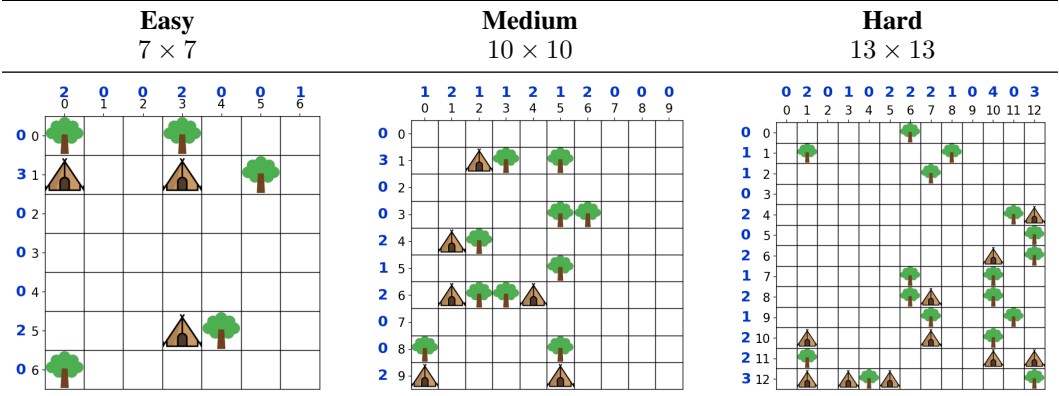

### J.3.7 RHYTHM GAME

This is a rhythm game featuring dynamic falling blocks. Players are tasked with selecting a column to place their finger and clicking on the operation blocks that fall to the first row of the selected column to score points. Alternatively, players may choose not to click any column, which will not affect the falling of the blocks. The blocks in the game are divided into three types: Click blocks, Reverse blocks, and Snake blocks, each with different scores and click effects, prompting players to make choices while playing to get the highest score.

Questions will be asked based on the current game situation, involving issues such as block type identification, grid ratio calculation, and score calculation. Players need to reason and answer based on the information on the screen in the game. In addition, the game difficulty is divided into three levels according to the complexity of the scene, including Easy (15Œ4), Medium (15Œ6), and Hard (20Œ6).

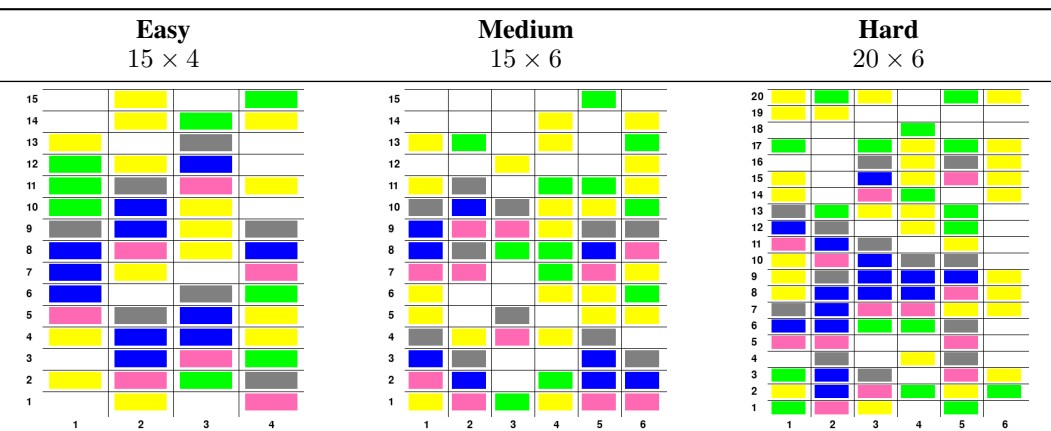

### J.3.8 LIFEGAME

This is a cellular automaton simulation on an nŒn 2D grid, where cells are either "alive" (black squares) or "dead" (white/empty squares) and evolve over generations. A cell's next state is determined by its current state and its eight neighbors: a dead cell with exactly three live neighbors becomes alive (simulating reproduction); an alive cell dies with fewer than two (simulating underpopulation) or more than three live neighbors (simulating overpopulation), but survives with two or three.

Game tasks involve counting current alive cells, predicting alive cells after one iteration, determining a specific cell's state change over N iterations, and calculating iterations for a given region to reach a stable state (static or oscillating). The "Plot Level" (difficulty) is determined by the grid size, with larger grids indicating higher difficulty.

| Easy | Medium | Hard |
|------|--------|------|
| $3 \times 3$ | $4 \times 4$ | $5 \times 5$ |

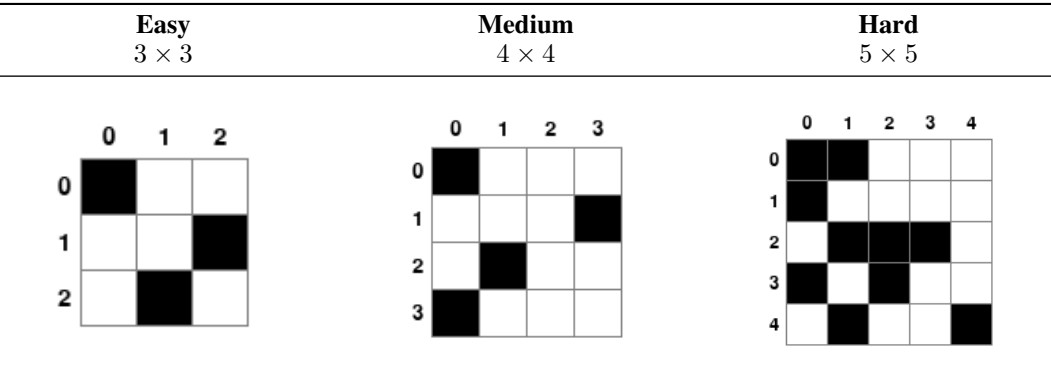

### J.3.9 MINESWEEPER

The game is inspired by Microsoft's classic game Minesweeper. The objective is to reveal all cells that do not contain mines while correctly flagging the mines. If a player accidentally reveals a cell containing a mine, the game ends immediately. The Minesweeper game board consists of cells marked with numbers (indicating the number of mines in the surrounding 3x3 grid), white revealed cells, gray hidden cells, flagged cells (marked with the letter "F"), and cells containing mines, which are unknown to the player. The game tasks include determining the status of cells, inferring the locations of mines, predicting the outcome of actions, and deciding on optimal reveal strategies. The difficulty levels are determined by the board size, with 4x4 being easy, 5x5 being medium, and 6x6 being hard. The board size and the number of mines change based on the difficulty level.

| Easy | Medium | Hard |
|------|--------|------|
| $4 \times 4$ | $5 \times 5$ | $6 \times 6$ |

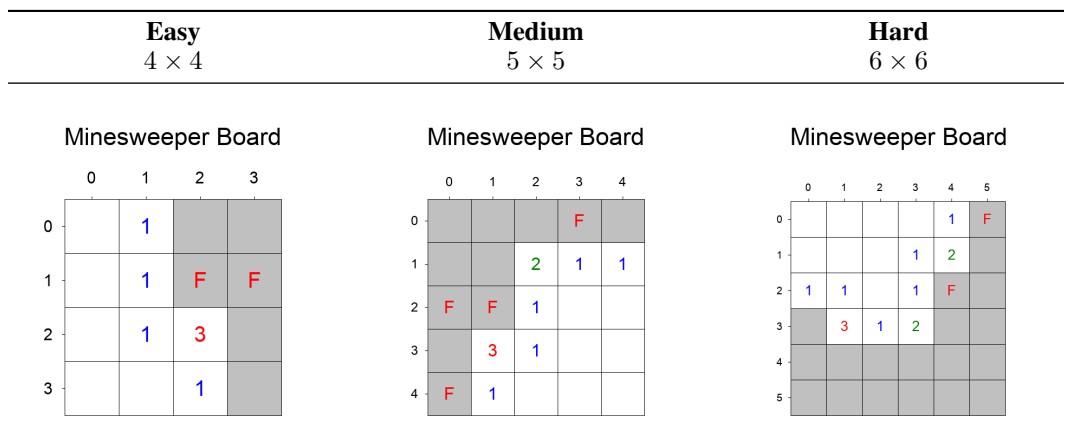

## J.4 STRATEGIC PLANNING

### J.4.1 SOKOBAN

This game is based on the classic Sokoban puzzle game. The game scene consists of a grid-based area featuring a player (represented by a black humanoid figure), boxes (brown squares with X texture), target points (green X marks), walls (brick-textured barriers), and movable areas (light brown floor). Players can move in four directions (up, down, left, right), push boxes forward, but cannot pull boxes or move through walls. The objective is to push all boxes onto target points. Question types evaluate spatial planning and logical reasoning: (1) predicting the player's final position after a sequence of moves; (2) predicting a box's final position after a sequence of movements; (3) determining the minimum number of moves required to solve the puzzle; (4) identifying the current position of the player; (5) calculating the Manhattan distance between a box and its target point; and (6) finding the optimal sequence of moves to reach a specific position. The game difficulty is determined by the board size.

**Problem information**

|    | QA type | QA Level | Description |
|----|---------|----------|-------------|
| Q1 | Target Perception | Easy | Identify the current position of the player on the board |
| Q2 | Target Perception | Easy | Calculate the Manhattan distance between a box and its target |
| Q3 | State Prediction | Medium | Given a sequence of player moves, predict the final position of the player |
| Q4 | State Prediction | Medium | Given a sequence of moves, predict the final position of the box |
| Q5 | Strategy Optimization | Hard | Find the optimal sequence of moves to reach a specific position |
| Q6 | Strategy Optimization | Hard | Determine the minimum number of moves needed to solve the puzzle |

**Images and Plot Level division**

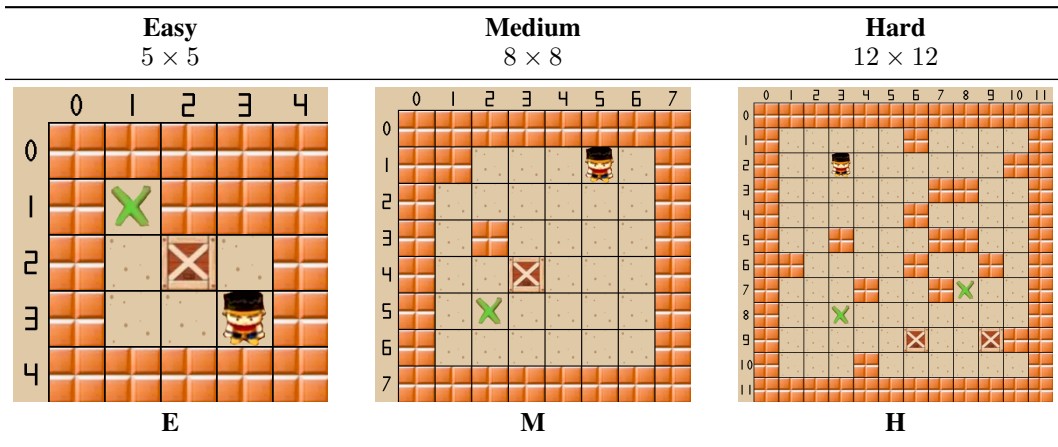

| **Easy** | **Medium** | **Hard** |
|----------|------------|----------|
| $5 \times 5$ | $8 \times 8$ | $12 \times 12$ |
| E | M | H |

**Specific questions and analysis**

*Introduction:* This is a Sokoban puzzle where cartoon person is player, green X is target, brown box with X is box to push, brown tiles are walls, and light brown areas are movable spaces. The coordinates (x, y) in this puzzle represent the matrix format.

**Q1 (M):** What is the current position of the player (row, column)?
Options:
[1] (6, 4) [2] (1, 4) [3] (3, 4) [4] (1, 5)
[5] (4, 3) [6] (4, 1) [7] (5, 1) [8] (6, 3)

**Analysis:** - Player position: (1, 5) - Boxes positions: (4, 3) - Target positions: (5, 2) The player is currently at position (1, 5). So the answer is (1, 5). The option number is 4.

**Q2 (M):** What is the Manhattan distance between the box and the target?
Options:
[1] 15 [2] 16 [3] 6 [4] 2
[5] 12 [6] 1 [7] 14 [8] 5

**Analysis:** - Player position: (1, 5) - Boxes positions: (4, 3) - Target positions: (5, 2) Box position: (4, 3) Target position: (5, 2) Manhattan distance = |4 - 5| + |3 - 2| = 2. So the answer is 2. The option number is 4.

**Q3 (E):** If the player makes these moves: Up Down Left Up Down Left Left Up, where will player end up?
Options:
[1] (6, 5) [2] (1, 6) [3] (6, 6) [4] (2, 3)
[5] (2, 6) [6] (1, 2) [7] (5, 2) [8] (2, 5)

**Analysis:** - Player position: (1, 5) - Boxes positions: (4, 3) - Target positions: (5, 2) Move sequence analysis: Initial position: (1, 5) Move 1 - Up: Failed - Wall in the way (Player stays at (1, 5)) Move 2 - Down: Player moves from (1, 5) to (2, 5) Move 3 - Left: Player moves from (2, 5) to (2, 4) Move 4 - Up: Player moves from (2, 4) to (1, 4) Move 5 - Down: Player moves from (1, 4) to (2, 4) Move 6 - Left: Player moves from (2, 4) to (2, 3) Move 7 - Left: Player moves from (2, 3) to (2, 2) Move 8 - Up: Player moves from (2, 2) to (1, 2) Final position: (1, 2). So the answer is (1, 2). The option number is 6.

**Q4 (M):** Treat boxes as objects that can move by themselves, and treat people as floor (movable areas). After the moves up, right, down, up, left, right, up, left, where will the box that started at position (4, 3) end up?
Options:
[1] (2, 3) [2] (3, 6) [3] (3, 1) [4] (1, 5)
[5] (6, 2) [6] (4, 6) [7] (3, 5) [8] (6, 4)

**Analysis:** - Player position: (1, 5) - Boxes positions: (4, 3) - Target positions: (5, 2) Move sequence: Move up: Box moved from (4, 3) to (3, 3) Move right: Box moved from (3, 3) to (3, 4) Move down: Box moved from (3, 4) to (4, 4) Move up: Box moved from (4, 4) to (3, 4) Move left: Box moved from (3, 4) to (3, 3) Move right: Box moved from (3, 3) to (3, 4) Move up: Box moved from (3, 4) to (2, 4) Move left: Box moved from (2, 4) to (2, 3) Box moves from (4, 3) to (2, 3). So the answer is (2, 3). The option number is 1.

**Q5 (M):** Treat the boxes as walls. What is the shortest sequence of moves for human to move himself from position (1, 5) to position (1, 6)?
Options:
[1] Down [2] Left [3] Down Right Down [4] Right
[5] Right Down Left [6] Down Right Left
[7] Down Left Up [8] Left Down

**Analysis:** - Player position: (1, 5) - Boxes positions: (4, 3) - Target positions: (5, 2) Start position: (1, 5) End position: (1, 6) Optimal move sequence: Right. So the answer is Right. The option number is 4.

**Q6 (M):** What is the minimum number of moves needed to solve this puzzle?
Options:
[1] 5 [2] 15 [3] 10 [4] 7
[5] 11 [6] 9 [7] 6 [8] 8

**Analysis:** - Player position: (1, 5) - Boxes positions: (4, 3) - Target positions: (5, 2) Solution analysis: Step-by-step solution: Player moves from (1, 5) to (2, 5) Player moves from (2, 5) to (3, 5) Player moves from (3, 5) to (3, 4) Player moves from (3, 4) to (3, 3) Player moves from (3, 3) to (4, 3) (box moves from (4, 3) to (5, 3)) Player moves from (4, 3) to (4, 4) Player moves from (4, 4) to (5, 4) Player moves from (5, 4) to (5, 3) (box moves from (5, 3) to (5, 2)) Total player moves: 8. So the answer is 8. The option number is 8.

### J.4.2   MAZE

This project focuses on generating question-and-answer datasets for a grid-based maze game. In this game, a player, represented by a red circle, must navigate a path of white blocks to reach a green goal block, while avoiding blue obstacle blocks. Movement is restricted to the four cardinal directions. The generated questions are designed to evaluate a range of cognitive abilities, primarily centered on spatial reasoning and pathfinding. These include tasks such as identifying the current locations of game elements, determining permissible moves, predicting the outcomes of specific actions, and deducing optimal routes to the goal. The complexity of the mazes and the associated questions scales, with mazes offered in Small, Medium, and Large sizes, and individual questions categorized by difficulty.

**Images and Plot Level division**

| Easy $9 \times 9$ | Medium $11 \times 11$ | Hard $13 \times 13$ |
|:---:|:---:|:---:|

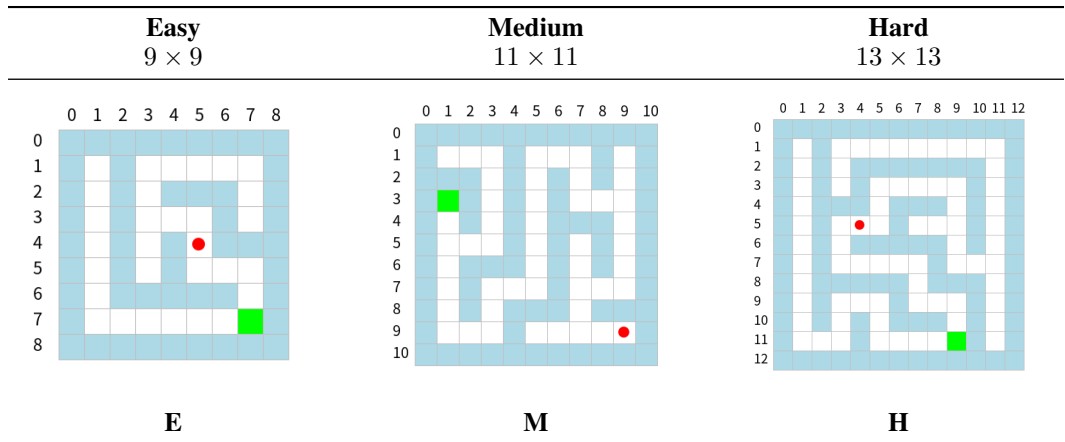

| E | M | H |
|:---:|:---:|:---:|

**Problem information**

|  | QA type | QA Level | Description |
|---|---|---|---|
| Q1 | Target Perception | Easy | Ask the position of player |
| Q2 | Target Perception | Easy | Ask the position of goal within the maze |
| Q3 | Target Perception | Easy | Ask the available directions to move are currently |
| Q4 | State Prediction | Medium | The position after moving |
| Q5 | Strategy Optimization | Hard | Find the path to the goal |
| Q6 | Strategy Optimization | Hard | Count how many turns it takes to reach the finish |

**Specific questions and analysis**

*Introduction:*

1. This is a maze mini-game. The player needs to navigate around obstacles to reach the destination and achieve victory.

2. The red circle represents the player, the green block is the goal and the blue blocks are obstacles.

3. The player can only move within the white blocks.

4. The coordinates are given in the format (row, col), where row represents the vertical position and col represents the horizontal position.

| | |
|---|---|
| **Q1 (E):** | Which of the following are the coordinates of the player?
Options:
A. (4, 6); B. (5, 5); C. (3, 5); D. (4, 4); E. (4, 5) |
| **Analysis:** | Take a look at the game screen, the red circle represents the player. The coordinates of player are (4, 5), so the right option is E. |
| **Q2 (E):** | Which of the following are the coordinates of the goal?
Optoins:
A. (7, 7); B. (7, 8); C. (6, 7); D. (7, 6); E. (8, 7) |
| **Analysis:** | Take a look at the game screen, the green block represents the goal. The coordinates of goal are (7, 7), so the right option is A. |
| **Q3 (E):** | Which directions are available to move now?
Options:
A. up; B. down; C. up, down; D. up, right;
E. left, right; F. up, down, right;
G. down, left, right; H. up, down, left, right |
| **Analysis:** | The player is on (4, 5), and (3, 5) (5, 5) is empty. The player can move up, down. Therefore, the option is C. |
| **Q4 (E):** | What are the coordinates of player after moving down?
Options:
A. (4, 6)
B. (5, 5)
C. (3, 5)
D. (4, 4)
E. (4, 5) |
| **Analysis:** | Observe the screen, the position of player is (4,5). After moving down, the player is in (5, 5). Therefore, the right option is B. |
| **Q5 (E):** | Which sequence of movements will allow the player to reach the destination?
Options:
A. left, left, left, right, right, left
B. down, right, right, down, down
C. down, down, left, up, right, down
D. up, up, up, right, up, down
E. left, down, left, right, down, left |
| **Analysis:** | Lets figure out the path to the goal step by step: Step 1. Go down, from (4, 5) to (5, 5). Step 2. Go right, from (5, 5) to (5, 6). Step 3. Go right, from (5, 6) to (5, 7). Step 4. Go down, from (5, 7) to (6, 7). Step 5. Go down, from (6, 7) to (7, 7). Achieved the goal! Therefore, the right sequence of movements are: down, right, right, down, down. The right option is B. |
| **Q6 (E):** | Find the path to the finish and count the number of turns it takes to get there. Provide one number. |
| **Analysis:** | First, lets figure out the path to the goal step by step: Step 1. Go down, from (4, 5) to (5, 5). Step 2. Go right, from (5, 5) to (5, 6). Step 3. Go right, from (5, 6) to (5, 7). Step 4. Go down, from (5, 7) to (6, 7). Step 5. Go down, from (6, 7) to (7, 7). Achieved the goal! Therefore, the path is: (4, 5), (5, 5), (5, 6), (5, 7), (6, 7), (7, 7).
Then, let's count the number of turns step by step: Step 2. Turn detected: from down to right. Step 3. No turn detected. Step 4. Turn detected: from right to down. Step 5. No turn detected.
In summary, the total number of turns is 2. |

### J.4.3 TICTACTOE

This game is derived from the classic Tic-Tac-Toe game, featuring a 3Œ3 grid area with two players represented by red and blue grid markers respectively. The objective is to create a straight line of three same-colored markers either horizontally, vertically, or diagonally to win. Question types include: (1) determining the color of specific grid cells, (2) identifying the optimal move for the current player, and (3) predicting the opponent's best response after a given move. The difficulty scales across three levels based on scenario complexity, where higher difficulty requires evaluating progressively more decision-making conditions to answer the same question types, systematically testing the model's strategic reasoning and conditional judgment capabilities.

**Images and Plot Level division**

| Easy | Medium | Hard |
|------|--------|------|

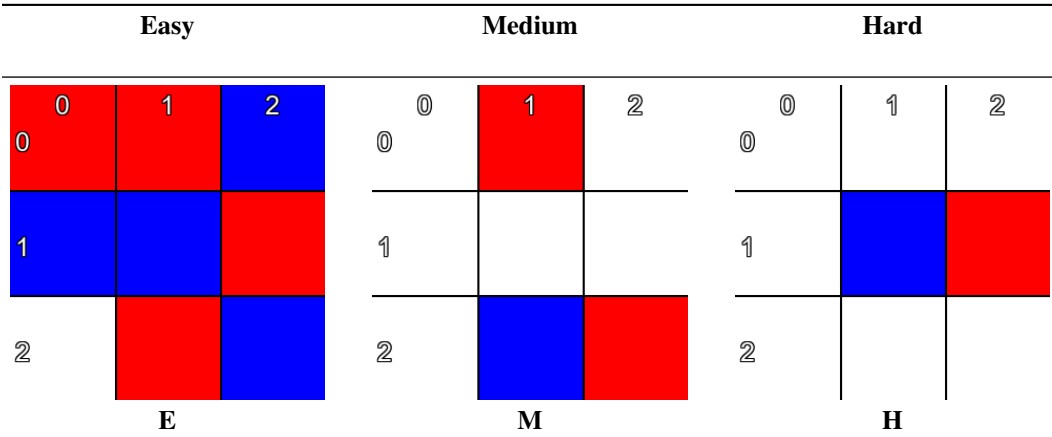

| E | M | H |
|---|---|---|

**Question information**

| | QA type | QA Level | Description |
|---|---|---|---|
| Q1 | Target Perception | Easy | Questions about the current state of a specific block of the board. |
| Q2 | Strategy Optimization | Medium | Questions about the optimal strategy to take a move of the current player of the board. |
| Q3 | Strategy Optimization | Hard | Questions about the outcome to take a specific move of the current player of the board, and the optimal strategy to take a move of the opponent player after the specific move. |

**Specific questions and analysis**

*Introduction:*

Tic-Tac-Toe is a classic two-player game played on a 3x3 grid, (row, col) from (0, 0) to (2, 2). Players take turns marking a space in the grid, one using **O** (the red block) and the other using **X** (the blue block). In each game, player **O** starts first. The objective is to be the first to get three of your marks in a row (horizontally, vertically, or diagonally). If all nine squares are filled without either player achieving this, the game ends in a draw. Notice: the current player to make a move should be inferred from the number of pieces for each players on the board. When inferring the optimal move, if optimal move can be inferred by some rules, choose the optimal move. Otherwise, choose the first move. (The order of choices is (0, 0), (0, 1), (0, 2), (1, 0), ..., (2, 2), choose the first move that is not occupied)

**Q1 (E):** Question: What is the color of the block at (0, 0)?
Options: A. red; B. blue; C. white

**Analysis:** The current board is [['O', 'O', 'X'], ['X', 'X', 'O'], [' ', 'O', 'X']]. The block at (0, 0) is "O", and the color matching "O" is red, so the block at (0, 0) is red. The answer is A.

**Q2 (M):** What is the optimal move for the current player? If no move exists, choose the answer "None".
Options: A. None; B. (0, 0); C. (0, 1); D. (0, 2); E. (1, 0); F. (1, 1); G. (1, 2); H. (2, 0) or (2, 1) or (2, 2)

**Analysis:** The current board is [[' ', 'O', ' '], [' ', ' ', ' '], [' ', 'X', 'O']]. Since the player "O" plays first in each game, if the count of "O" is the same as "X", the current player is "O". Otherwise, the current player is "X". The count of "O" is 2 and the count of "X" is 1, so the player now is X. Current player is X, opponent is O. Must block opponent O's potential double threat on Row 0 and Top-left to bottom-right diagonal, so player X should choose position (0, 0). The answer is B.

**Q3 (H):** If the current player moves to (0, 2), will this move be successful? If not, choose the answer "None". If successful, will the current player win immediately? If yes, choose the answer "None". Otherwise, what is the opponent's optimal move following this step?
Options: A. None; B. (0, 0); C. (0, 1); D. (0, 2); E. (1, 0); F. (1, 1); G. (1, 2); H. (2, 0) or (2, 1) or (2, 2)

**Analysis:** Yes, this move will be successful. The current board is [[' ', ' ', ' '], [' ', 'X', 'O'], [' ', ' ', ' ']]. Since the player "O" plays first in each game, if the count of "O" is the same as "X", the current player is "O". Otherwise, the current player is "X". The count of "O" is 1 and the count of "X" is 1, so the player now is O. Since the current player O moves to (0, 2), the current player won't win immediately. After that, current player is X, opponent is O. Must block opponent O's winning threat on Column 2, so player X should choose position (2, 2). The answer is H.

### J.4.4 ULTRA TICTACTOE

Ultra TicTacToe is an advanced variant of TicTacToe played on a 3x3 grid of 3x3 subgrids (Nine-grids). Players alternate placing "X" (first player) and "O" (second player) markers using a four-coordinate system (i,j,row,col), where (i,j) denotes the subgrid position and (row,col) specifies the cell within that subgrid. The initial move must be made in the central Nine-grid (2,2), with subsequent moves constrained to the subgrid determined by the opponent's previous move position. Scoring occurs when three identical markers form a line within any subgrid (each such line counts as 1 point). The game concludes when all nine central cells of the subgrids are occupied. Question types involve analyzing board states (identifying marker ownership at coordinates), calculating available move options, quantifying marked cells, evaluating scoring patterns within subgrids, and determining optimal strategic placements. Game complexity tiers are defined by move count ranges: Easy (10-34 steps), Medium (35-59 steps), and Hard (60-81 steps).

### J.4.5 SPACE INVADERS

Adapted from the classic arcade game, Space Invaders is a simplified space warfare game. Players control a ship at a grid's bottom, moving it by column to fire lasers upward. Lasers destroy the nearest alien invader in that column (different colors worth 10, 20, or 30 points), earning points and potentially exposing others. Collectively moving enemies add dynamic challenge. The game uses visually intuitive images for ship and aliens instead of text symbols.

Players analyze game scene images to answer questions covering: game state perception (e.g., enemy counts by location or color); single-shot outcome prediction (points from current or post-move shots); effects of consecutive shots in dynamic scenarios; and strategic planning for maximum points. These questions range from simple recognition to complex reasoning. Three difficulty levels are based on scene complexity; higher levels feature larger grids with more numerous and complexly arranged enemies, demanding greater player skill.

| Easy | Medium | Hard |
|------|--------|------|

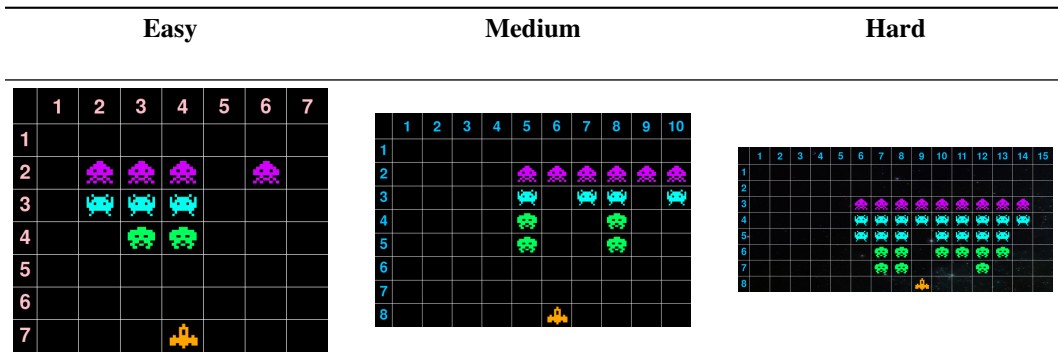

### J.4.6 SNAKE

This game is derived from the classic game Snake, which involves a square white grid scene with coordinates, with snakes and food represented by colored squares. The snake head is represented by a yellow square, the snake body by blue squares, and the food by a red square. Each step the snake can move in four directions: up and down, left and right. The game ends if the snake head hits the bound of the grid or its own body.

The questions include 1. The coordinate of the snake head. 2. The coordinate of the food. 3. The length of the snake. 4. Which will happen until this process ends if following a specific sequence of moves (hitting its own body, hitting the wall, reaching the food, or nothing happens)? 5. The length of the shortest path to reach the food. Plot Level is determined by the grid size.

| Easy
$5 \times 5$ | Medium
$10 \times 10$ | Hard
$15 \times 15$ |
|------|--------|------|

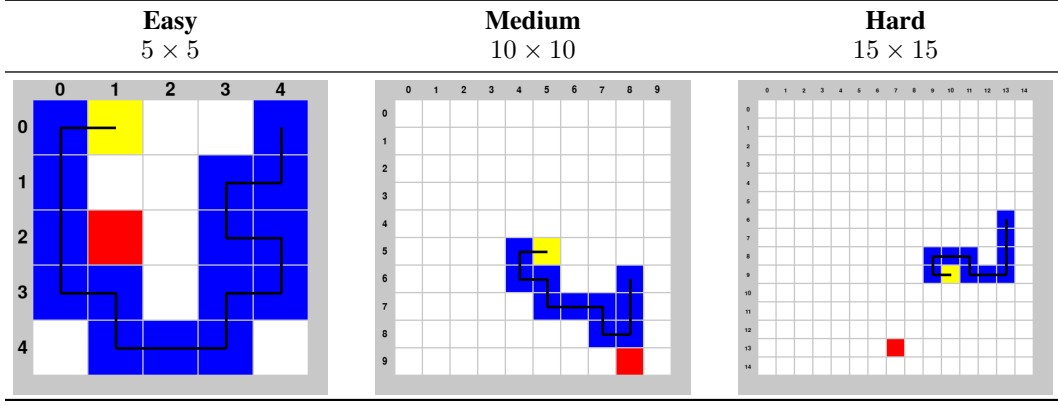

### J.4.7 CHESS RANGER

Chess ranger is derived from chess. The game presents a problem with an 8×8 chessboard image containing 6 pieces, where the possible types of pieces are King, Queen, Rook, Bishop, Knight, and

Pawn. The goal of the game is to use the movement and capture rules of chess pieces to ensure that only one piece remains on the board at the end.

The types of questions in the game are as follows:1.The number of pieces of a certain type on the board.2.The identity of the piece located in a specific square on the board.3.The location of a particular type of piece on the board.4.The required number of steps to solve the current chessboard configuration.5.The moves that can solve the puzzle among several possible options. The difficulty level of the game is determined by the number of pieces on the board: 4, 5, 6 pieces corresponding to easy, medium and hard.

| **Easy** | **Medium** | **Hard** |
|:---:|:---:|:---:|
| 4 pieces | 5 pieces | 6 pieces |

### J.4.8 PACMAN

The game is inspired by the classic maze game Pac-Man, with the original four ghosts simplified to just two ghosts. The objective of the game is for Pac-Man to eat as many beans as possible while avoiding being caught by the ghosts. The game scene includes Pac-Man, beans, walls, and ghosts (Pinky and Blinky), with Pac-Man, Pinky, and Blinky represented by special images. The beans are represented as small yellow circles, and the walls are dark blue squares. Pac-Man, Pinky, and Blinky cannot move through walls. The dataset includes tasks such as determining Pac-Man's current position and direction, counting the number of beans in a specific area, predicting the paths of the ghosts, forecasting the outcome of Pac-Man's movements, and analyzing strategies to maximize the score while avoiding ghosts. The dataset is divided into three difficulty levels based on grid size: Easy (16x16), Medium (18x18), and Hard (20x20).

| **Easy** | **Medium** | **Hard** |
|:---:|:---:|:---:|
| $16 \times 16$ | $18 \times 18$ | $18 \times 18$ |

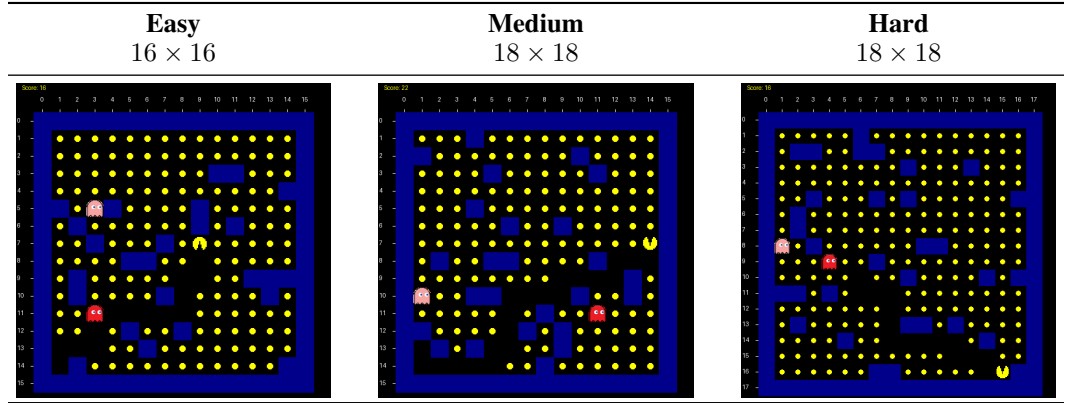

