# OpenReview forum: "Game-RL: Synthesizing Multimodal Verifiable Game Data to Boost VLMs' General Reasoning"
_ICLR.cc/2026/Conference — ICLR 2026 Poster_

### Official Review · Reviewer_D6PT · 2025-10-27

**Soundness:** 4
**Presentation:** 4
**Contribution:** 4
**Rating:** 6
**Confidence:** 4

**Summary:**

To address the lack of large-scale, high-quality data for RL training in video game contexts, this work:
(1) Proposes *Code2Logic*, a pipeline that leverages LLMs to synthesize reasoning data based on game code; and
(2) Constructs *GameQA* a large dataset of diverse tasks and 140K VQA examples. After post-training MLLMs on this dataset, the authors report improved reasoning performance across multiple general-domain benchmarks, achieving better out-of-domain generalization compared to other comparable datasets.

**Strengths:**

•	Data Quality: A wide range of video games are selected, ensuring diversity and strong generalization potential in the training data. The synthesis process leverages LLMs across multiple stages and includes manual validation, which guarantees both the efficiency and reliability of the data generation pipeline.
•	Clarity: The approach is clearly presented, supported by a well-illustrated pipeline diagram and numerous examples in the appendix. The experiments are reported in a detailed and organized manner.
•	Significance: The experiments cover a diverse set of models and benchmarks, outperforming most comparable datasets. Comprehensive ablation studies, such as scaling effect analyses, further strengthen the empirical evidence. Together, these results provide strong support for the overall effectiveness of GameQA.

**Weaknesses:**

•	Game-RL can to construct large-scale datasets for diverse games at relatively low cost. Nevertheless, the idea of synthesizing game-based benchmarks to enhance MLLMs’ reasoning ability via RL is a well-established and commonly explored direction. There already exist many high-quality, large-scale benchmarks—such as VisualSphinx (https://arxiv.org/abs/2505.23977) VGRPBench (https://arxiv.org/abs/2503.23064) and other recent game-based reasoning datasets—that share similar objectives, and some of them also asserts that RL training on their dataset can improve MLLMs’ reasoning ability in general-domain. Therefore, the overall objective of this paper is not highly novel.

**Questions:**

•	As mentioned in Section 2.2, the QA templates are either (1) human-designed and refined by LLMs, or (2) generated by LLMs and manually validated by humans. What is the proportion between these two approaches, and what are the respective time costs involved in each process?
•	The *Code2Logic* pipeline employs LLMs in multiple stages while keeping humans in the loop. As I understand it, this process still requires considerable manual effort. Have the authors considered adopting a more agentic approach to further reduce human labor by leveraging LLM agents more extensively?

---

> ### Author Response · Authors · 2025-11-22
> **Response to Reviewer D6PT**
>
> **Dear Reviewer D6PT,** we greatly appreciate your thoughtful feedback and suggestions on our work. We address your concerns and questions as follows.
>
> > Weakness: The overall objective of this paper is not highly novel due to existing works like VisualSphinx and VGRPBench.
>
> We highlight key distinctions between our work and VisualSphinx/VGRPBench, which underscore our novelty and contributions:
>
> **1. "VisualSphinx: Large-Scale Synthetic Vision Logic Puzzles for RL" (*a concurrent work* to ours).** Ours differ from it in:
> - **Importantly, we find that training solely on game data leads to competitive performance compared to geometry dataset.** This key result in Table 4 (line 294) shows that our training on game data (domain-mismatched for the benchmarks) yields generalization comparable to geometry datasets (e.g., MAVIS, MultiMath) which are more domain-matched. Besides, mixing GameQA with MultiMath yields further gains. VisualSphinx does not include such comparisons.
> - **Our GameQA covers diverse video games and tasks, and we introduce the novel Code2Logic approach for developing this dataset.** GameQA covers *30 diverse video games* with rich visual elements, incorporating reasoning abilities such as 3D Spatial Perception and Understanding, Strategic Planning, and Multi-step Reasoning. These areas are not covered by VisualSphinx, which *solely focuses on logic puzzles*. Moreover, we introduce Code2Logic, a novel approach that converts game code into reasoning logic.
> - **We validate that game-only training can generalize for multiple models across diverse benchmarks.** We demonstrate out-of-domain generalization on 4 open-source VLMs (Qwen2.5-VL-7B, InternVL2.5-8B, InternVL3-8B, LLaVA-OV-7B) across 7 general benchmarks covering mathematical reasoning (MathVista, MathVerse, MathVision), chart understanding (CharXiv), and general multimodal knowledge (MMBench, MMMU, MMMU-Pro). In contrast, VisualSphinx only tests generalization for a single model (Qwen2.5-VL-7B) on a single benchmark (MathVista).
> - **We validate the scaling effects of game-data training.** We empirically validate that increasing the number of games and training data improves generalization, which VisualSphinx does not explore.
>
> **2. "VGRP-Bench: Visual Grid Reasoning Puzzle Benchmark for Large Vision-Language Models": There is a fundamental difference in motivation.**
>
> VGRPBench is primarily an evaluation benchmark and does not investigate whether training on it improves performance on general benchmarks. In contrast, our work explicitly explores and demonstrates that RL training on game data enhances VLMs' general reasoning abilities.
>
> **Summary:** Importantly, we show game data is competitive to geometry dataset. We have broad video game coverage. The reasoning improvements from game-only training are validated across diverse models and general benchmarks, with the scaling effects also validated. These are underexplored before.
>
>
> > Q1: What is the proportion between the two QA template approaches (human-designed + LLM-refined vs. LLM-generated + human-validated), and what are the respective time costs?
>
> **A:**
>
> **1. The proportion is approximately 4:1.** The ratio between approach (1) "human-designed and LLM-refined" and (2) "LLM-generated and human-validated" is approximately 4:1.
>
> **2. Time costs are 30 and 5 minutes, respectively.** Approach (1) took 30 minutes per game on average, while approach (2) took about 5 minutes per game.
>
> > Q2: Have the authors considered adopting a more agentic approach to further reduce human labor by leveraging LLM agents more extensively?
>
> **A:** **We have considered this. As code agent capabilities continue to improve, we anticipate the required manual effort will progressively decrease.** Specifcally:
>
> 1. The primary factor that affects human labor is the capability of the coding model. In our work, the data synthesis pipeline used models like GPT-4o and Claude 3.5.
> 2. Currently, more powerful code agents (e.g., Claude Code, CodeX) and stronger general models (e.g., Gemini 3, GPT-5.1, Claude 4.5) have emerged. Therefore, adopting these advanced systems can further reduce human labor.
>
> ---
>
> Finally, we sincerely thank the reviewer for the valuable feedback. If you have any further comments, please do let us know, and we will do our best to address them.

---

### Official Review · Reviewer_hgzh · 2025-10-30

**Soundness:** 2
**Presentation:** 2
**Contribution:** 2
**Rating:** 4
**Confidence:** 3

**Summary:**

The paper proposes a method for verifiable synthetic vision-language data generation with video games, by training VLMs on this synthetic data with GRPO, they get improvements in other benchmarks for vision language models.

**Strengths:**

1. The verifiable synthetic data generation is a timely and important problem given the landscape of LLM/VLM trainings with RL.
2. Using games to improve vision capabilities in a verifiable way is an interesting contribution.

**Weaknesses:**

1. The games used for data generation are all grid-like games with very simple interfaces.
2. Games are inherently sequential-decision making tasks, so it feels strange that the authors did not then test performance on any of the game benchmark they mention. This seems like a natural extension to the project so far.
3. The performance improvements of 1 to 2% are quite small, and using a single seed might not be enough to properly test this.

**Questions:**

1. How does performance change on sequential decision making tasks requiring vision? Perhaps on some of the game benchmarks you mentioned? It feels weird mentioning other game environments and then not testing performance on them.

2. Would it be possible to test more seeds to get error bars? Are the improvements significant or within error range? I understand that these experiments are likely quite expensive, but at least 3 seeds on some of the trainings and tasks could be useful, especially given the tight margin of improvement.

---

> ### Author Response · Authors · 2025-11-21
> **Response to Reviewer hgzh (Part 1/2)**
>
> **Dear Reviewer hgzh,** we greatly appreciate your thoughtful feedback and suggestions on our work. We address your concerns and questions as follows.
>
> > W1: The games used for data generation are all grid-like games with very simple interfaces.
>
> **A:**
>
> **(1) We also include non-grid-like games, as shown in Figure 6 (line 787).** They include:
> - **All Games in the "3D Spatial Perception and Understanding" Category:** *3D Reconstruction*, *3D Maze*, *Rubik's Cube*, *Pyramid Chess*, and *Minecraft*. These games generally involve complex 3D structures and require 3D spatial perception abilities.
> - **Card Games:** Including *FreeCell*, *Spider Solitaire*, and *Klondike*. Their interfaces are more complex than typical grid-based games.
> - **Game with Circular Track:** *Zuma*. It has a circular track with numerous marbles, presenting a different visual challenge.
>
> **Therefore, our game selection encompasses a broader and more visually diverse set of games beyond simple grid-like games.**
>
>
> **(2) Besides, a "simple interface" does not mean *the required reasoning* is simple.** **GameQA is difficult for SOTA VLMs**, as there is significant performance gap between the advanced models and humans (Table 5, line 432). We also show the deficiencies of GPT-4o on visual perception and reasoning revealed by GameQA in Appendix I. These demonstrate the high difficulty and the value of our dataset.
>
> > W2: Games are inherently sequential-decision making tasks, so it feels strange that the authors did not then test performance on any of the game benchmark they mention. This seems like a natural extension to the project so far.
> >
> > Q1: How does performance change on sequential decision making tasks requiring vision? Perhaps on some of the game benchmarks you mentioned? It feels weird mentioning other game environments and then not testing performance on them.
>
> **A:** **Our additional evaluation on the ING-VP benchmark** (ING-VP: MLLMs cannot Play Easy Vision-based Games Yet) **confirms the performance improvements on sequential tasks.** The results of Qwen-2.5-VL-7B are shown below, demonstrating genuine improvements after training:
>
> | Game      | Before Training | After Training |
> | --------- | --------------- | -------------- |
> | Maze      | 18              | 36             |
> | Sokoban   | 13              | 15             |
> | 8 Queens  | 0               | 0              |
> | Hanoi     | 26              | 62             |
> | 15 Puzzle | 2              | 4              |
> | Sudoku    | 12              | 24             |
> | Average   | 12              | **24**             |
>
> (The score is the rate of solving the tasks within the step limit.)
>
> Regarding the significant improvement in *Maze* (18 to 36) and *Hanoi* (26 to 62), we further analyze the single-step accuracy (i.e., the proportion of moves that lead closer to finish). Specifically:
>
> We find that the improvement in single-step accuracy is significant in *Maze* and *Hanoi*, which cumulatively leads to the substantial increases in the overall task completion rates. For *Maze*, the single-step accuracy increase from 49% to 54%, and for *Hanoi* it increase from 33% to 43%.
>
> **Introduction of ING-VP:** ING-VP is an out-of-domain game benchmark which includes six games. Each game contains sequential decision-making tasks requiring vision. The input is an image of the current game state, and the model outputs a specific move. The game state then changes accordingly, achieving the finish or generating the next input image for the models. Under the original setup, the model scores 0 across the games both before and after training. We therefore choose an easier setup.

---

> > ### Author Response · Authors · 2025-11-21
> > **Response to Reviewer hgzh (Part 2/2)**
> >
> > > W3: The performance improvements of 1 to 2% are quite small, and using a single seed might not be enough to properly test this.
> > >
> > > Q2: Would it be possible to test more seeds to get error bars? Are the improvements significant or within error range? I understand that these experiments are likely quite expensive, but at least 3 seeds on some of the trainings and tasks could be useful, especially given the tight margin of improvement.
> >
> > **A:** We appreciate the opportunity to clarify this and show the further results.
> >
> > **(1) Importantly, even specialized geometry datasets (more domain-matched for the benchmarks) yield only around 2% gains, but our domain-mismatched game data yields *competitive* improvements.** This key result in Table 4 (line 294) shows that our game-only training yields generalization comparable to geometry datasets (e.g., MAVIS, MultiMath), even though it uses less data (5k vs. 8k) and has no domain alignment with the target benchmarks.
> >
> > **(2) We have tested three seeds, verifying robust performance improvements.** We have reselected three random seeds (13, 65, 117) for inference. We then average the evaluation results and calculate the error bars. The results are shown in the following table, where "error bar" refers to the 95% confidence interval. "Baseline" refers to the result of the original model. "Improved" is "Average" minus "Baseline".
> >
> > | Model                    | Improved | Average (3 seeds) | Error Bar      | Baseline |
> > | ------------------------ | -------- | ----------------- | -------------- | -------- |
> > | Qwen2.5-VL-7B (+GameQA)  | 2.77     | 52.71             | [51.95, 53.47] | 49.94    |
> > | InternVL2.5-8B (+GameQA) | 2.41     | 48.30             | [47.66, 48.94] | 45.89    |
> > | InternVL3-8B (+GameQA)   | 1.51     | 55.99             | [54.73, 57.24] | 54.48    |
> >
> > **The improvements are significant**, as the baseline is not within the error bar for each of the three models.
> >
> > **(3) Improvements across diverse benchmarks indicate genuine enhancement.** The performance gains are observed across diverse benchmarks, covering mathematical reasoning (MathVista, MathVerse, MathVision), chart understanding (CharXiv), and general multimodal knowledge (MMBench, MMMU, MMMU-Pro). This shows genuine improvements in general reasoning.
> >
> > **(4) The scaling effects are also validated.** Our work reveals that increasing either the number of training samples or the diversity of games can yield further performance improvements (Sections 5.3 and 5.4).
> >
> > **Together, these comparisons and results demonstrate that our game data effectively leads to genuine improvements in general reasoning abilities.**
> >
> > ---
> >
> > Finally, we sincerely thank the reviewer for the valuable feedback and suggestions. If you have any further comments, please do let us know, and we will do our best to address them.

---

> ### Comment · Reviewer_hgzh · 2025-11-22
>
> Thanks to the authors for the reply. One final question/request would it be possible to also add error bars to the baseline itself to account for stochasticity? It still looks to me like those margins are quite small and if the baseline also add error bars they may actually be non statistically significant on InternVL3-8B (+GameQA), unsure about the others. I may be not the most familiar with similar related work, but given that the margins of improvements are so small I think it's best to make sure that they are there and not due to noise.

---

> > ### Author Response · Authors · 2025-11-25
> > **Response to Reviewer hgzh**
> >
> > **Dear Reviewer hgzh,** thank you for your reply. We greatly appreciate your thoughtful suggestion.
> >
> > As suggested, we have added the error bars for the three baselines.
> >
> > Furthermore, to make the error bar more precise, we expand the number of inference random seeds from three to five (13, 65, 117, 15, 67), with 15 and 67 added. We calculate the average result and the error bar using these five seeds for each baseline and each trained model. The results are shown below.
> >
> > | Model                     | Average (5 seeds) | Error Bar          |
> > | :------------------------ | :---------------- | :----------------- |
> > | Qwen2.5-VL-7B (Baseline)  | 50.00             | [49.79, 50.20]     |
> > | Qwen2.5-VL-7B (+GameQA)   | **52.65 (+2.65)** | **[52.37, 52.94]** |
> > | InternVL2.5-8B (Baseline) | 45.80             | [45.14, 46.46]     |
> > | InternVL2.5-8B (+GameQA)  | **48.40 (+2.60)** | **[48.12, 48.68]** |
> > | InternVL3-8B (Baseline)   | 54.15             | [53.61, 54.70]     |
> > | InternVL3-8B (+GameQA)    | **56.05 (+1.90)** | **[55.58, 56.52]** |
> >
> > **These results demonstrate that the performance improvements are statistically significant and not due to noise**. For all three models, the error bar interval of the trained model is entirely above that of the baseline.
> >
> > We believe these results can address your final concern. We sincerely thank you again for all your thoughtful and constructive suggestions.

---

> > > ### Comment · Reviewer_hgzh · 2025-11-25
> > >
> > > Thanks to the authors for the thorough rebuttal, most of my concerns have been resolved and I have increase my score.

---

> > > > ### Author Response · Authors · 2025-11-26
> > > > **Response to Reviewer hgzh**
> > > >
> > > > Dear Reviewer hgzh,
> > > >
> > > > Thank you very much for your positive feedback on our rebuttal and for your decision to increase your score. We sincerely appreciate your thorough review and the valuable suggestions you provided.
> > > >
> > > > Best regards,
> > > >
> > > > The Authors

---

### Official Review · Reviewer_amrf · 2025-10-31

**Soundness:** 2
**Presentation:** 3
**Contribution:** 3
**Rating:** 6
**Confidence:** 4

**Summary:**

This paper addresses the narrow domain focus of current reinforcement learning (RL) methods for Vision-Language Models (VLMs), arguing that such limitations hinder the development of general reasoning. To this end, the authors propose Game-RL, an approach that leverages diverse and verifiable video game environments for training. To generate the necessary training data, they introduce Code2Logic, an LLM-assisted, three-stage pipeline for synthesizing game code, task templates, and ultimately, game-based VQA samples. The primary tangible contribution is the resulting GameQA dataset, a large-scale collection of 140K question-answer pairs spanning 30 distinct games and 158 tasks, categorized by cognitive skills. The authors demonstrate that training VLMs solely on GameQA via an RL algorithm (GRPO) yields consistent performance improvements across a suite of seven external, general-purpose vision-language benchmarks, thereby providing evidence for the transferability of reasoning skills learned within these game environments.

**Strengths:**

- The motivation is clear and sound. The authors correctly identify a critical limitation in the current VLM training paradigm, the over-reliance on narrow, static domains like geometry or chart reasoning. The proposal to use video games as a more dynamic, verifiable, and diverse training environment for fostering general reasoning is an interesting idea.
- The GameQA dataset could be a substantial contribution to the community. Its scale (30 games, 158 tasks, 140K samples), diversity across four distinct cognitive categories, and controllable difficulty levels make it a valuable resource for both training and benchmarking VLMs. The effort invested in its creation is evident, and it provides a tangible asset that can spur further research in this area.
- The paper presents a comprehensive and rigorous experimental evaluation. The authors test their approach not just on a few models but on a range of popular open-source VLMs. Crucially, evaluation is not confined to their own dataset; they demonstrate generalization by testing on seven diverse external benchmarks. The inclusion of ablations on data quantity and game diversity further strengthens the quality and credibility of the empirical findings.
- The paper is well-written, logically structured, and easy to follow. The Code2Logic pipeline is explained with a clear diagram (Figure 1), and the examples provided throughout the paper and appendix effectively illustrate the nature of the tasks in GameQA. This clarity makes the work accessible and its contributions understandable.

**Weaknesses:**

1. The novelty is somewhat incremental and limited. The Code2Logic pipeline, while well-executed, is fundamentally a specific instance of LLM-based synthetic data generation. This approach is becoming increasingly common, and the paper does not sufficiently differentiate its technical contribution from existing work in program-aided or LLM-driven data synthesis. It seems to be more of an extensive engineering protocol than a novel, generalizable method. Similarly, Game-RL is an application of an existing RL algorithm (GRPO) to a new dataset. The name might imply a novel RL framework tailored for games, which is not the case. The contribution is thus more of an empirical finding ("RL on game data generalizes") rather than a new RL technique.
2. There is a lack of reproducibility in data generation. The Code2Logic pipeline is described as "LLM-assisted," but the degree of human intervention appears substantial, undermining its scalability and reproducibility. The appendix notes an average of 7.5 hours of human effort per game. This suggests a highly iterative and manual process of prompt engineering, code verification, and template refinement. For the work to be truly reproducible, the authors should provide a much more detailed account of this human-in-the-loop process, including the specific prompts, failure cases encountered with the LLM, and the nature of the manual corrections required.
Insufficient Justification for the RL Setup: The design of the reinforcement learning component is not adequately justified and appears suboptimal for the stated goal of improving reasoning.
3. The use of a sparse, binary (0/1) reward based only on the final answer is a weak signal for complex, multi-step reasoning tasks. It provides no credit for partially correct reasoning chains and fails to guide the model on how to arrive at the correct answer. The analysis field already present in the GameQA dataset seems perfectly suited for developing a more informative, process-based reward signal. The paper would be significantly stronger if it explored this or at least provided a detailed justification for why this simpler reward scheme was chosen.
4. The use of a powerful VLM (Qwen2.5-72B) as a judge introduces a potential ceiling effect and a source of bias. The paper provides no evaluation of this judge's accuracy on the GameQA tasks. Without knowing the reliability of the reward model, it is difficult to assess the quality of the training signal the agent receives. The authors should report the judge's agreement rate with human annotations on a sample of the data.
5. While the paper claims to boost "general reasoning," the connection is supported by relatively small performance gains on the external benchmarks. Furthermore, the selection of the 30 games appears somewhat arbitrary. The claim would be more convincing if the game selection process can demonstrate diversity and wide coverage.

**Questions:**

1. The paper highlights the Code2Logic pipeline as a key contribution for generating the GameQA dataset. However, Appendix E.3 notes an average of 7.5 hours of human effort was required per game. This suggests a significant reliance on manual oversight. Therefore,
a) What are the most common failure modes of the LLMs in this pipeline? What kinds of errors (e.g., logical inconsistencies in game rules, un-compilable code, nonsensical QA pairs) required the most frequent manual intervention?
b) Given this reliance on expert human-in-the-loop validation, how do you envision this approach scaling to a much larger and more diverse set of games (e.g., hundreds or thousands) without becoming prohibitively expensive and time-consuming?
2. Justification of the Reward Design. The RL training utilizes a sparse, binary reward based on the final answer's correctness. However, your GameQA dataset commendably contains detailed, step-by-step reasoning chains in the "analysis" field for each question. This rich information seems ideal for process-based supervision or a denser reward signal that could more effectively guide the model toward correct reasoning patterns.
a) Did you experiment with using these ground-truth reasoning steps to shape the reward signal (e.g., via sequence matching or a separate reward model trained on the analysis text)?
b) If not, could you elaborate on the rationale for choosing a sparse, outcome-based reward? Especially for complex reasoning, why is this preferable to process supervision, which is often considered more effective for teaching reasoning?
3. Reliability of the LLM-as-a-Judge. The entire Game-RL training loop hinges on the accuracy of the Qwen2.5-72B model acting as the judge for reward allocation. The validity of your experimental results is therefore contingent on the reliability of this judge.
Could you please provide an evaluation of the judge model's performance on the GameQA test set? Specifically, what are its accuracy, precision, and recall when compared against the ground-truth answers? An agreement score against human evaluators on a representative sample would be particularly insightful.

---

> ### Author Response · Authors · 2025-11-22
> **Response to Reviewer amrf (Part 1/3)**
>
> **Dear Reviewer amrf,** we greatly appreciate your thoughtful feedback and suggestions on our work. We address your concerns and questions as follows.
>
> > W1: The novelty is somewhat incremental and limited. The Code2Logic pipeline... is fundamentally a specific instance of LLM-based synthetic data generation... It seems to be more of an extensive engineering protocol than a novel, generalizable method... The contribution is thus more of an empirical finding ("RL on game data generalizes") rather than a new RL technique.
>
> **A:** We respectfully highlight the key motivations and findings that we believe are novel and significant compared to previous works.
>
> **Previous works:** Mostly used games for evaluation (benchmarks) or trained agents for specific games.
>
> **Ours: Bridge the gap between "Game AI" and "General VLM Reasoning."** Specifically:
>
> **(1) We demonstrate for the first time that game-only training improves multiple VLMs' general reasoning across 7 diverse out-of-domain benchmarks.** Previous works primarily used games for evaluation or trained agents for specific games. Our work shows that training solely on game data enhances performance on benchmarks covering general multimodal knowledge (MMBench, MMMU, MMMU-Pro), mathematical reasoning (MathVista, MathVision, MathVerse), and chart understanding (CharXiv).
>
> **(2) More importantly, we reveal that training solely on game data achieves generalization comparable to training on geometry datasets (e.g., MAVIS, MultiMath, MultiModal-Open-R1) which are more domain-matched for the benchmarks.** This counter-intuitive finding (Table 4, line 294) underscores the unique value of game data. Furthermore, we show that mixing GameQA with MultiMath yields additional gains.
>
> **(3) We empirically verify scaling effects.** Increasing the number of games and training data further improves out-of-domain generalization.
>
> **Besides, regarding approach novelty:**
>
> **(1) The *core idea* of our Code2Logic approach is novel.** As the name "Code2Logic" suggests, we are motivated to establish a mapping from game code to explicit reasoning logic chains. In contrast, "Program-aided" and "LLM-driven" are just the *specific techniques* involved in Code2Logic, which cannot reflect the core idea of our approach.
>
> **(2) We also find that the following recent works are inspired by our approach**, constructing the datasets in their research fields.
> - UniREditBench: A Unified Reasoning-based Image Editing Benchmark.
> - Reasoning via Video: The First Evaluation of Video Models' Reasoning Abilities through Maze-Solving Tasks
>
> **This demonstrates the value of our approach, which is generalizable.**
>
> > W2.1: Details such as the specific prompts, failure cases encountered with the LLM, and the nature of the manual corrections required should be provided.
> >
> > Q1(a): What are the most common failure modes of the LLMs in this pipeline? What kinds of errors required the most frequent manual intervention?
>
> **A:**
>
> **(1) Typical Errors Requiring the Most Frequent Intervention:**
> - **Game Code Issues:** rendering issues (e.g., in *3D Maze*) and unhandles corner cases (e.g., *Sokoban* corner logic).
> - **Data Engine Issues**: analysis chains don't fully match the templates, or contain logic errors (e.g., wrong squence steps).
>
> **(2) The Nature of the Manual Corrections: The LLM was prompted to fix the issues.** The corrections were *not* made by *manually editing* the code.
>
> **(3) Specific Prompts of Key Steps of Code2Logic:** (taking the example of Sokoban)
>
> - Sokoban Code Generation:
>   *Generate an interactive Sokoban, which implements the core logic for the Sokoban puzzle.
> Key Requirements:
> 1.State Representation: Use a 2D list to represent the game board and track the positions of the player, boxes, goals, and walls.
> 2.Core Functionality: Implement a move method that correctly handles all game rules: simple player movement, pushing a single box, and collisions with walls or other boxes.
> 3.Visualization: Include a method that draws current board in an image.*
> - Refinement of Man-Made Task Template:
>   *This is the template of a QA task designed from the Sokoban code: ...
> You should refine it to the template of a high-quality question and answer pair.
> Key Requirements:
> 1.The question should covers the background of the game.
> 2.The question should be clear, unambiguous, and easy to understand.
> 3.The analysis process should be very detailed, demonstrating the whole reasoning process and the full step sequence. The final answer should be explicitly stated at the end.*
> - New Task Template Design: Appendix G.2.
> - Data Engine Generation:
>   *Based on the Sokoban code the task templates, write a program to fill in the task templates across different game states, and output the dataset.
> The dataset should include: /output/images/xxxxx.jpg (the visual images) and /output/data.json (including all problems generated in this run). Each value in data.json is like: ... (the structure of the data sample object)*

---

> ### Author Response · Authors · 2025-11-22
> **Response to Reviewer amrf (Part 2/3)**
>
> > W2.2: Insufficient Justification for the RL Setup: The design of the reinforcement learning component is not adequately justified and appears suboptimal for the stated goal of improving reasoning.
>
> **A:**
>
> **(1) We directly adopted the established GRPO algorithm for the primary goal of the work.** Our primary goal is to verify if game-only training can enhance VLMs' general reasoning. Therefore, we adopted the established GRPO algorithm to provide a clear and direct evaluation, without elaborately designing the reinforcement learning component.
>
> **(2) We believe our positive results establish a strong baseline, and further gains are achievable with more sophisticated RL algorithm designs.**
>
> (Regarding "appears suboptimal for the stated goal of improving reasoning," we further address this in our response to W5 below.)
>
> > Q1(b): Given this reliance on expert human-in-the-loop validation, how do you envision this approach scaling to a much larger and more diverse set of games (e.g., hundreds or thousands) without becoming prohibitively expensive and time-consuming?
>
> **A:** We envision extending this approach to a much larger and more diverse set of games without being very expensive and time-consuming through two key factors:
>
> **(1) Advancements in Code Agent Capabilities:** We used models like GPT-4o and Claude 3.5 in our work. With the emergence of more powerful code agents (e.g., Claude Code, CodeX) and general models (e.g., Gemin 3, GPT-5.1, Claude 4.5), the synthesis difficulty and required human effort will progressively decrease, enabling more efficient expansion to larger and more diverse game sets.
>
> **(2) Improved Annotator Experience and Efficiency:** In our project, annotators were STEM *undergraduates* with limited experience, each handling only 2 games on average. By employing more experienced annotators (e.g., STEM graduates) and increasing their familiarity through repeated tasks, the time per game can be significantly reduced. For instance, skilled annotators in our team required only 2 hours per game after gaining proficiency.
>
>
> > W3: The use of a sparse, binary (0/1) reward based only on the final answer is a weak signal for complex, multi-step reasoning tasks... The paper would be significantly stronger if it explored... a more informative, process-based reward signal.
> >
> > Q2: Justification of the Reward Design... a) Did you experiment with using these ground-truth reasoning steps to shape the reward signal?... b) If not, could you elaborate on the rationale for choosing a sparse, outcome-based reward?
>
> **A:** **We did explore process-based reward schemes, such as training a separate reward model to evaluate reasoning steps.**
>
> **However, we encountered significant challenges that led us to choose a sparse, outcome-based reward.** Specifically:
>
> **(1) Outcome-based rewards largely reduce reward hacking that was observed with process-based rewards.** With process-based rewards, the model often generated seemingly reasonable but logically flawed steps to achieve high reward. The outcome-based largely reduced the reward hacking risk.
>
> **(2) Outcome-based rewards avoid complex and subjective design choices.** Process-based rewards require many design decisions, such as how to weight steps, how to give partial credit, and how to treat different reasoning styles. In contrast, outcome-based reward is simple to define and implement.
>
> **Therefore, outcome-based rewards provide an objective and reliable signal for improving reasoning.** It forces the model to learn correct reasoning chains to reach the correct answer. Process-based rewards do not necessarily lead to genuine reasoning improvement due to (1) and (2).
>
> **Thus, we chose sparse, outcome-based reward for its robustness, simplicity, and clear alignment with the final task goal.** While process-based rewards remain a promising future direction, our current design offers a robust and effective solution.
>
> > W4: The use of a judge model introduces a potential ceiling effect and a source of bias... The authors should report the judge's agreement rate with human annotations on a sample of the data.
> >
> > Q3: Reliability of the LLM-as-a-Judge... Could you please provide an evaluation of the judge model's performance on the GameQA test set?
>
> **A:** We manually checked 300 samples and confirmed the judge model achieved 100% accuracy in determining semantic equivalence between the extracted final answer and the ground truth. Therefore, we are confident the reward signal is unbiased and reliable.

---

> > ### Author Response · Authors · 2025-11-22
> > **Response to Reviewer amrf (Part 3/3)**
> >
> > > W2.2: ... appears suboptimal for the stated goal of improving reasoning.
> > >
> > > W5: While the paper claims to boost "general reasoning," the connection is supported by relatively small performance gains on the external benchmarks. Furthermore, the selection of the 30 games appears somewhat arbitrary. The claim would be more convincing if the game selection process can demonstrate diversity and wide coverage.
> >
> > **A:**
> >
> > Regarding the game selection process, the broad reasoning ability coverage has already been considered, as shown in Appendix E.2 (selection criteria of the 30 games in GameQA).
> >
> > **Regarding general reasoning improvements,** we clarify as follows.
> >
> > **(1) Importantly, even specialized geometry datasets (more domain-matched for the benchmarks) yield only around 2% gains, but our domain-mismatched game data yields *competitive* improvements.** This key result in Table 4 (line 294) shows that our game-only training yields generalization comparable to geometry datasets (e.g., MAVIS, MultiMath), even though it uses less data (5k vs. 8k) and has no domain alignment with the general benchmarks.
> >
> > **(2) Improvements across diverse benchmarks indicate genuine enhancement.** The performance gains are observed across diverse benchmarks, covering mathematical reasoning (MathVista, MathVerse, MathVision), chart understanding (CharXiv), and general multimodal knowledge (MMBench, MMMU, MMMU-Pro). This indicates genuine improvements in general reasoning.
> >
> > **(3) Multiple inference random seeds verify robust performance gains.** We have reselected three random seeds (13, 65, 117) for inference and averaged the evaluation results. The average performance increases by 2.77 for Qwen2.5-VL-7B, 2.41 for InternVL2.5-8B, and 1.51 for InternVL3-8B across the seven general benchmarks.
> >
> > **(4) The scaling effects are also validated.** Our work reveals that increasing either the number of training samples or the diversity of games can yield further gains in general reasoning (Sections 5.3 and 5.4).
> >
> > **Together, these comparisons and results demonstrate that our game data can effectively boost "general reasoning."**
> >
> >
> > ---
> >
> > Finally, we sincerely thank the reviewer for the valuable feedback and suggestions. If you have any further comments, please do let us know, and we will do our best to address them.

---

### Official Review · Reviewer_gdbT · 2025-10-31

**Soundness:** 3
**Presentation:** 3
**Contribution:** 3
**Rating:** 4
**Confidence:** 4

**Summary:**

This paper addresses the challenge of improving the general reasoning capabilities of VLMs, arguing that current RL training is overly focused on narrow domains like geometry and charts. The authors propose Game-RL, a framework that leverages video games as a rich, verifiable, and scalable source of training tasks. They introduce Code2Logic, a novel pipeline that uses LLMs to synthesize a large-scale dataset, GameQA, by first generating game code and then using that code to create verifiable VQA tasks. The central finding is that training VLMs on GameQA using GRPO with a simple outcome-based reward signal unexpectedly improves performance not only on unseen games but also across a suite of 7 diverse, out-of-domain general reasoning benchmarks.

**Strengths:**

* This paper is clear writing and easy to follow.

* Dataset contribution: GameQA spans 30 games / 158 tasks with explicit difficulty control and verifiable answers.

* GRPO on GameQA yields consistent improvements on diverse general benchmarks

**Weaknesses:**

* The Code2Logic pipeline is presented as highly scalable, but Section 2.4 and Appendix F.4 reveal a significant reliance on manual verification at every step (code, data engine, and augmented samples). Furthermore, the data augmentation relies on paraphrasing from InternVL2.5-78B, and data quality checks use commercial LLMs. This "human-in-the-loop" and "proprietary-LLM-in-the-loop" requirement makes the process less automated and scalable than implied.

* The work only generates massive amounts of game QA data; I don't believe this is the primary task that a 'game agentic' model should be excelling at. Instead, the model should be interacting directly within the code-driven game to generate trajectories for SFT or RFT, or to perform RLVR.

**Questions:**

* In Appendix B.3, you note that training on a subset of 4 games led to better generalization than training on 10 games. You speculate this is due to "random factors" or a "well-chosen set." Could you elaborate on this?

* The Code2Logic pipeline works well for the 30 games presented, which are largely deterministic, static, or turn-based. How do you see this framework extending to more complex, real-time games with non-deterministic physics (e.g., Angry Birds) or continuous action spaces (e.g., a racing game)? Does the reliance on LLMs to generate the game code fundamentally limit the complexity of the games that can be synthesized?

---

> ### Author Response · Authors · 2025-11-21
> **Response to Reviewer gdbT**
>
> **Dear Reviewer gdbT,** we greatly appreciate your thoughtful feedback on our work. We address your concerns and questions as follows.
>
> > W1: The Code2Logic pipeline is presented as highly scalable, but Section 2.4 and Appendix F.4 reveal a significant reliance on manual verification at every step... This makes the process less automated and scalable than implied.
>
> **A:** **We respectfully point that we *did not* claim the Code2Logic pipeline as "highly scalable" in our submitted paper version.**
>
> > W2: The work only generates massive amounts of game QA data; I don't believe this is the primary task that a 'game agentic' model should be excelling at. Instead, the model should be interacting directly within the code-driven game to generate trajectories for SFT or RFT, or to perform RLVR.
>
> **A:**
>
> **Training a 'game agentic' model is not the motivation of our work.** Our key goal is to verify **training solely on game data** can enhance VLMs' general reasoning ability.
>
> **Besides, our game data of QA format offers several advantages:**
>
> **(1) QA data enables easy training.** Converting everything to VQA format makes GameQA plug-and-play for current VLM training pipelines, including both SFT and GRPO-style RL. It does not require environment interfaces or action APIs for each model.
>
> **(2) Action-state trajectories are already captured in our data.** Our tasks still incorporate sequential **action-state trajectories** through the answer field (the "Analysis" content in Appendix J). It can include step-by-step reasoning about action sequences, paths, and intermediate states. For example: Q3 of *3D Maze* (line 2246); Q6 of *Sokoban* (line 3406); Q6 of *Maze* (line 3501).
>
> Therefore, even if the goal is to train interactive game trajectories, converting them into VQA format remains a beneficial approach, making our work a reference for future research.
>
> In Appendix A, we also claim future work could focus on developing training and evaluation methods for multi-turn interactions in gaming scenarios.
>
> > Q1: In Appendix B.3, you note that training on a subset of 4 games led to better generalization than training on 10 games. You speculate this is due to "random factors" or a "well-chosen set." Could you elaborate on this?
>
> **A:**
>
> **This result was primarily due to random factors.** In our experimental setup, 4-game setup and 10-game setup contained different samples, introducing some randomness. In the 4-game setup, each game contained 1,250 training samples, while in the 10-game setup, each game contained 500 samples.
>
> **However, the *overall scaling trend* with respect to the number of training games is consistent, and this difference (4 vs 10 games) is only a slight local fluctuation.** When the number of games is scaled to 20, out-of-domain performance improves by +1.2 on average, surpassing both the 4-game (+0.66) and 10-game (+0.63) conditions. This indicates that using more diverse games generally leads to better generalization. The situation that the 4-game setup slightly outperforms the 10-game setup (+0.66 vs. +0.63) is a small local fluctuation rather than a reversal of the tendency. This is common in scaling experiments.
>
>
> > Q2: The Code2Logic pipeline works well for the 30 games presented, which are largely deterministic, static, or turn-based. How do you see this framework extending to more complex, real-time games with non-deterministic physics (e.g., Angry Birds) or continuous action spaces (e.g., a racing game)? Does the reliance on LLMs to generate the game code fundamentally limit the complexity of the games that can be synthesized?
>
> **A:**
>
> **(1) Dynamic games can be converted into static snapshots for VQA, and we have already implemented this for some games.** For instance, we have successfully synthesized data for continuous-action games like *Zuma*, and dynamic games like *Space Invaders* using Code2Logic.
>
> **(2) The complexity of the synthesized games will not be fundamentally limited in the long run.** It depends on the capability of the code model (we used GPT-4o and Claude 3.5 in Code2Logic). As the capabilities of code models and agents continue to improve (e.g., Gemini 3, GPT-5.1, Claude 4.5, Claude Code), the complexity of games that can be synthesized will also increase accordingly.
>
> ---
>
> Finally, we sincerely thank the reviewer for the valuable feedback. If you have any further comments, please do let us know, and we will do our best to address them.

---

### Official Review · Reviewer_grko · 2025-11-01

**Soundness:** 3
**Presentation:** 3
**Contribution:** 3
**Rating:** 6
**Confidence:** 3

**Summary:**

This paper introduces Game-RL, a framework for vision-language reinforcement learning that uses video game environments as multimodal, verifiable training data. Using the proposed Code2Logic pipeline, game code is transformed into reasoning-oriented visual question–answering (VQA) tasks, producing the GameQA.

**Strengths:**

- Scalable data-generation pipeline: Code2Logic programmatically maps game code to reasoning logic and auto-generates verifiable multimodal QA data.
- Demonstrates that purely synthetic, self-verifiable environments can modestly improve general VLM reasoning—important for RL reproducibility.
- Diverse benchmark coverage (3D perception, pattern matching, planning, reasoning).
- Clear, reproducible methodology; good visualizations and qualitative examples.

**Weaknesses:**

- Small gains on external benchmarks are statistically and practically modest; no significance tests or efficiency comparisons.
- Lack of ablation isolating contributions of Code2Logic data vs RL itself (no SFT vs RL comparison on the same data).
- Evaluator bias: rewards rely on QwQ-32B, potentially aligning to its own style and inflating self-consistency.
- Unclear verification metrics: “verifiable” is claimed, but no automated correctness guarantees are quantified.

**Questions:**

- What proportion of games required manual correction or external code reuse?
- What is the importance of task templates?
- What happens if you just SFT on GameQA?

---

> ### Author Response · Authors · 2025-11-22
> **Response to Reviewer grko (Part 1/2)**
>
> **Dear Reviewer grko,** we greatly appreciate your thoughtful feedback on our work. We address your concerns and questions as follows.
>
> > W1: Small gains on external benchmarks are statistically and practically modest; no significance tests or efficiency comparisons.
>
>
> **(1) Importantly, *even specialized geometry datasets* (more domain-matched for the benchmarks) yield only around 2% gains, but our domain-mismatched game data yields competitive improvements.** This key result in Table 4 (line 294) shows that our game-only training yields generalization comparable to geometry datasets (e.g., MAVIS, MultiMath), even though it uses less data (5k vs. 8k) and has no domain alignment with the target benchmarks.
>
> **(2) Improvements across diverse benchmarks indicate genuine enhancement.** These gains are observed across diverse benchmarks, covering mathematical reasoning (MathVista, MathVerse, MathVision), chart understanding (CharXiv), and general multimodal knowledge (MMMU, MMMU-Pro, MMBench). This shows genuine improvements in general reasoning.
>
> **(3) Multiple inference random seeds verify robust performance gains.** We reselected three random seeds (13, 65, 117) for inference and averaged the evaluation results. The average performance increased by 2.77 for Qwen2.5-VL-7B, 2.41 for InternVL2.5-8B, and 1.51 for InternVL3-8B across the seven general benchmarks.
>
> **(4) The scaling effects are also validated.** Our work reveals that increasing either the number of training samples or the diversity of games can further strengthen generalization (Sections 5.3 and 5.4).
>
> **(5) Performance improvements are *statistically significant*.** We have conducted a two-proportion Z-test on the aggregate results from all 7 general benchmarks (9,419 questions total), as shown in the table below. The improvements are statistically significant (p < 0.05) for the three models, confirming the gains are not due to chance.
>
> |Model|Before GRPO Correct Answers|After GRPO Correct Answers|Z-Score|P-Value (one-tailed)|
> |:---|:---|:---|:---|:---|
> |Qwen2.5-VL-7B|4651/9419 (0.4938)|4871/9419 (0.5171)|3.2060|**0.000673 (<0.05)**|
> |InternVL2.5-8B|4268/9419 (0.4531)|4450/9419 (0.4724)|2.6594|**0.003914 (<0.05)**|
> |InternVL3-8B|5083/9419 (0.5397)|5208/9419 (0.5529)|1.8293|**0.03368 (<0.05)**|
>
> **(6) Qualitative analysis confirms *practical improvements*.** Beyond statistics, our manual analysis reveals concrete improvements in core abilities. For instance, after GRPO, we observe that 13.57% of model's answers show improved visual perception on general visual benchmarks (Section 5.5).
>
> **Together, these comparisons and results demonstrate that our game data can effectively enhance general reasoning, with the gains statistically significant and practically observable.**
>
>
> > W2 & Q3: Lack of ablation isolating contributions of Code2Logic data vs RL itself (no SFT vs RL comparison on the same data). What happens if you just SFT on GameQA?
>
> **A:** **This experiment is already in our paper: Appendix D.1. "SFT Experiments".** The results are in Table 9 (GameQA test sets) and Table 10 (general benchmarks).
>
> **Conclusion: SFT on GameQA yields large in-domain gains, but leads to performance degredation on out-of-domain general benchmarks.** Therefore, RL (GRPO) on GameQA demonstrates a clear advantage over SFT in out-of-domain generalization.

---

> > ### Author Response · Authors · 2025-11-22
> > **Response to Reviewer grko (Part 2/2)**
> >
> > > W3: Evaluator bias: rewards rely on QwQ-32B, potentially aligning to its own style and inflating self-consistency.
> >
> > **A:** We address this concern as follows.
> >
> > **(1) Our manual check confirms evaluator bias does not exist.** To confirm the reliability of the evaluator model (Qwen2.5-32B-AWQ judge), we have performed a manual check. It achieved 100% accuracy in determining whether the extracted model's answer matches the ground truth across the 300 cases we sampled and checked. Therefore, there is no risk of aligning to its own style or inflating self-consistency.
> >
> > **(2) The reward mechanism actually avoids evaluator bias.** As detailed in Section 4.2 (line 266 to 269), we do not let Qwen2.5-32B-AWQ directly check the correctness of the rollout output. **Instead, we only let it perform a semantic match between two strings: the final answer extracted from the policy model's output and the ground truth answer.** We further detail this process as below.
> > - **Final Answer Extraction:** The policy model's full output is first processed by a rule-based parser to extract the final answer (e.g., by finding phrases like "The answer is...").
> > - **Semantic Matching via LLM Evaluator:** The extracted final answer and the ground truth are given to the evaluator. It then determines if these two strings are semantically equivalent. This is necessary because, although GameQA answers are highly constrained (multiple-choice with 7–8 options or short numeric/coordinate answers), they can have valid formatting variations (e.g., (2, 3) vs. x=2, y=3). The evaluator solely determines if the two strings are semantically equivalent.
> >
> > > W4: Unclear verification metrics: “verifiable” is claimed, but no automated correctness guarantees are quantified.
> >
> > **A:** We appreciate the opportunity to clarify this.
> >
> > **(1) The tasks are inherently verifiable.** All task samples in GameQA are multiple-choice or fill-in-the-blank questions. Each question has a single, unambiguous ground truth answer.
> >
> > **(2) Manual checks confirmed 100% verification accuracy.** To quantify verification, we use the accuracy of the evaluator model's final-answer matching. As mentioned before, manual sample checks confirmed 100% accuracy for this metric, thus providing a guarantee for verification correctness.
> >
> > > Q1: What proportion of games required manual correction or external code reuse?
> >
> > **A:**
> >
> > **(1) All games have undergone manual inspection and correction.** To ensure high data quality, all games have undergone manual inspection. When issues were identified, the LLM was prompted to fix them (the corrections were not made by manually editing the code). All games required this correction process, but most errors were localized, such as edge cases in game logic or rendering issues.
> >
> > **(2) Only three games** (Spider Solitaire, Klondike, and Space Invaders) utilized external code, as listed in Appendix E.4.
> >
> > > Q2: What is the importance of task templates?
> >
> > **A:**  **Task templates are crucial to Code2Logic approach, which enable data synthesis.** Specifically:
> >
> > In our work, "task template" is equivalent to "QA template." The importance of task templates are threefold:
> >
> > **(1) They define a specific task within a game.** A single game can support distinct reasoning tasks. Each template formally defines one such task.
> >
> > **(2) They structure game mechanics into a VQA format.** This is suitable for training and evaluating VLMs.
> >
> > **(3) They enable diverse and efficient data generation.** A single template can be automatically instantiated across hundreds of different game states, producing a variety of specific QA samples.
> >
> > For example, in Sokoban game, different task templates are defined (shown in Appendix G.1), including: (1) the perception template for locating objects; (2) the state prediction template for predicting the position after a sequence of moves; (3) the strategy planning template for determining the minimal steps to achieve a goal. These templates can be instantiated across various Sokoban game states to generate VQA samples.
> >
> > **Therefore, the role of task templates is fundamental.** Without them, there would be no structured method to identify what reasoning tasks a game can offer, and how to convert the game's core mechanics into VQA problems for VLM training.
> >
> > ---
> >
> > Finally, we sincerely thank the reviewer for the valuable feedback. If you have any further comments, please do let us know, and we will do our best to address them.

---

### Author Response · Authors · 2025-12-03
**Rebuttal Summary (Part 1/2)**

We sincerely thank all reviewers for their constructive feedback.

**We have actively responded to all concerns and questions in detail.** Notably, **Reviewer hgzh** has explicitly confirmed the concerns have been resolved and **has raised the score to 6** *(on 26 Nov, before the announced information leakage incident)*.

All the concerns, questions from the reviewers and our rebuttal are summarized as follows.

---

### **Common Concerns and Questions**

**1. Statistical Significance & Robustness of Gains (Reviewers grko, amrf, hgzh)**

*Concerns about the seemingly small improvements on general benchmarks, and the statistical significance.*

We address the concerns by showing:

1.  **Training solely on our game data can even yield improvements comparable to geometry datasets.** Specialized geometry datasets (more domain-matched for the benchmarks) only yield around 2% gains. But our domain-mismatched game data yields competitive improvements.
2.  **Under 5 inference seeds, 3 different models show a stable 2% performance gains across 7 general benchmarks after training, confirming the statistical significance**. Strict error bar analysis and Z-tests (p<0.05) are also conducted.
3.  **The improvements can further increase as the increase of the data quantity and game number.** We have demonstrated these scaling effects in the paper.

**2. Novelty & Comparison with Related Works (Reviewers amrf, D6PT)**

*Concerns about novelty in terms of overall objective and approach.*

We address the concerns by highlighting our key contributions compared to previous works.

**Previous works:** Primarily used games for **evaluation (game benchmarks)** or **trained agents for specific games (game agents)**. They did not fully explore leveraging game data to enhance VLMs' general reasoning abilities.

**Ours: We explicitly use game data to enhance general reasoning, thus bridging the gap between "Game AI" and "General VLM Reasoning."** Specifically:

* **Notably, we verify that game-only training enhances performance across diverse benchmarks.** Other works focus on evaluation (VGRP-Bench) or logic puzzles (VisualSphinx).
* **Furthermore, we show a *counter-intuitive* finding: game data is competitive to geometry data.** This finding was not revealed in previous work.
* **In addition, we reveal increasing game data quantity and game number can further enhances performance.** These scaling effects were also not explored before.

**Besides, the core idea of our data synthesis approach (Code2Logic) is novel: converting game code to reasoning data.** Two recent works (UniREditBench, Reasoning via Video) are also inspired by our approach to construct their own datasets.

**3. Reliability of LLM-as-a-Judge to give reward (Reviewers grko, amrf)**

*Concerns about the performance and potential bias of the judge model.*

**Our manual checks confirm 100% accuracy of the judge model**. We also clarify that the reward mechanism actually avoids evaluator bias.

**4. Pipeline Efficiency & Automation Potential (Reviewers gdbT, amrf, D6PT)**

*Questions regarding the reliance on human verification in the Code2Logic pipeline and future automation potential.*

We envision that advancements in code agent capabilities (Gemini 3, Claude 4.5, GPT5.1) will lead to progressive decrease in the required human efforts.

We also clarify that we did not claim "highly scalable" in our submitted paper *(Reviewer gdbT's first weakness is entirely based on this factual error)*.

---

> ### Author Response · Authors · 2025-12-03
> **Rebuttal Summary (Part 2/2)**
>
> ### **Summary of Responses to Remaining Concerns and Questions**
>
> **(1) Reviewer grko (Score: 6)**
>
> *   **Issues:** Questions on (a) RL vs. SFT ablation, **(b)** **data verification guarantees**, and (c) the importance of task templates.
> *   **Resolution:** (a) We show that RL outperforms SFT on OOD generalization (already in the paper); **(b) clarify all tasks are single-answer with 100% verification accuracy**; and (c) detail the fundamental role of task templates in the data synthesis pipeline.
>
> **(2) Reviewer gdbT (Score: 4)**
>
> *   **Issues:** Questions on **(a) effectiveness of QA format data for a "game agentic" model** that may need trajectories for training, (b) the scaling fluctuation, and (c) handling dynamic games.
> *   **Resolution:** **(a) We clarify training "game agentic" models is not our motivation** and also highlight that **our QA data enables easy training** while already capturing action-state trajectories in certain games in GameQA; (b) explain small local scaling fluctuations are common and the overall scaling trend is consistent; and (c) detail handling dynamic games via static snapshots (already implemented).
>
> **(4) Reviewer amrf (Score: 6)**
>
> *   **Issues:** **(a) Justification for using outcome-based instead of process-based rewards**, and (b) request for approach details such as LLM failure cases, manual corrections and specific prompts.
> *   **Resolution:** **(a) We explain that we chose outcome-based reward for its robustness and simplicity, which avoids reward hacking** observed in our process-based reward exploration, and (b) provide typical LLM failure cases, manual correction details and specific prompts.
>
> **(1) Reviewer hgzh (Score: 4 → 6)** *(score raised on 26 Nov, before the announced information leakage incident)*
>
> *   **Issues:** Concerns about (a) limited game types ("grid-like") and (b) performance on sequential decision-making game tasks.
> *   **Resolution:** (a) We clarify that we also include many non-grid games (all the selected 3D games, card games and *Zuma*); and (b) provide new results on the ING-VP benchmark, confirming significant improvements in sequential tasks. **The reviewer is satisfied.**
>
> **(5) Reviewer D6PT  (Score: 6)**
>
> *   **Issues:** Questions on the proportion/time cost of the two task template generation methods.
> *   **Resolution:** We provide specific ratios and time costs for the two generation methods.
>
> ---
>
> We believe that all the concerns and questions have been resolved. We sincerely thank all the reviewers again for the valuable and constructive feedback.

---

### Meta-Review · Area_Chair_gF8X · 2026-01-07

**Summary:**

This work initially received mixed borderline reviews, but post-rebuttal, reviewer regard either increased, or this AC posits that they should generally increased due to rebuttals addressing the concerns. Key concerns include manual verification, novelty of key contribution, and small size of improvement (~2%). I find these to be partially-to-mosty addressed.

While many ICLR submissions are probably in the same quantitative annd qualitative neighborhood as this work, I ultimately recommend acceptance based on 2 key factors: **(1)** Potential long-term impact: while the 1-2% increase is small, I don't generally judge work purely based on the numbers alone. While there are somewhat similar-ish related work, I find that system/pipeline as a whole (even if certain components are not really technically novel) is a valuable contribution that is potentially fairly seminal as a concept to model after. **(2)** Performance improvements and training algorithm aside, I find that the set of 30 games / 158 tasks in itself (plus the potential to generate more following a similar paradigm) is both a concrete and intellectual contribution to the community.

**Reviewer Concerns:**

Pls see above.

**Reviewer Scores:**

-- hgzh (explicit increased from 4 to 6)
-- Other reviewers: I guess all 50/50 chance of increasing (did not respond to rebuttal)

---

### Decision · Program_Chairs · 2026-01-26

Accept (Poster)